# RNAscope: Benchmarking RNA Language Models for RNA Sequence Understanding

## Abstract

Pre-trained language models (pLMs) have advanced our understanding of RNA biology. However, current evaluation frameworks remain limited in capturing the inherent complexity of RNA, leading to insufficient and biased assessments that hinder their practical applications. Here, we introduce RNAscope, a comprehensive benchmarking framework designed to gauge RNA pLMs via structure prediction, interaction classification, and function characterization. This framework includes 1,253 experiments spanning diverse subtasks of varying complexity and enables systematic model comparison with consistent architectural modules. Model assessment shows that generalization of sequence flexibility across RNA families, target contexts, and environmental features remains challenging for existing models. RNAscope provides a systematic, robust, and fair evaluation framework to accelerate RNA modeling.

## 1 Introduction

RNA is central to biology, modulating gene expression and protein synthesis [42]. Its inherent structural flexibility is often shaped by interaction with other biomolecules, influencing key cellular processes [27, 77]. While RNA functions are evolutionarily encoded in their sequence, decoding the underlying sequence-based features across various biological contexts remains essential.

Concurrently, RNA pLMs [1, 10, 76, 11], pre-trained on diverse RNA species, have been utilized to study a wide range of contextual RNA properties, including structure [12, 73, 58], interaction [75, 72, 35], and function [4, 36, 8]. However, existing evaluation frameworks [50, 54] have yet to fully progress in parallel with the rapid development of these models, limiting their ability to reflect the broader landscape of structural patterns [60, 31, 48], interaction strength [62, 30, 49], and functional characteristics [9, 22, 44]. For instance, predicting binary RNA-target interaction alone is insufficient to characterize the full spectrum of biological RNA intermolecular binding, where their affinity and specificity often span several orders of magnitude.

In this study, we present **RNAscope**, a systematic benchmarking framework to evaluate the performance of pLMs (Fig.1). It assessed their representational capacity by compiling three cohorts of subtasks. This framework offers the following main contributions to the evaluation of RNA pLMs:

- **Comprehensive RNA Benchmarking.** RNAscope comprises 15 core subtasks for evaluating RNA pLMs across three fundamental biological domains: structural prediction, molecular interaction, and functional characterization. It showcases a systematic comparison and critical assessment with 1,253 experiments in varying modules, addressing a broad spectrum of RNA biology landscape.

- **In-depth Comparative Analysis.** RNAscope provides an extensive evaluation of various state-of-the-art RNA and DNA pLMs, harnessing the complexity of RNA properties across RNA families, targets contexts, and environmental features. It further presents a detailed comparison of their

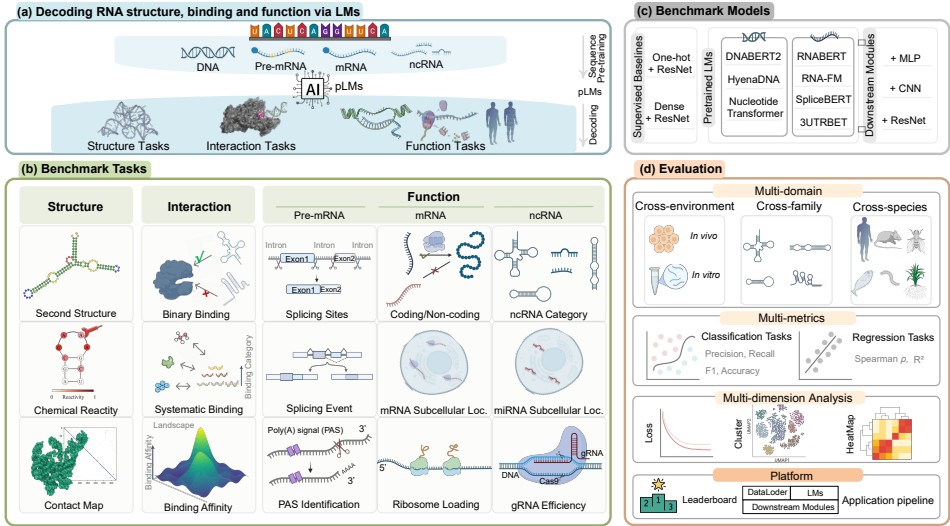

Figure 1: Overview of RNAscope. (a) RNAscope systematically evaluates pLMs in their abilities to capture RNA structure, interactions, and functions . (b-d) It comprises 15 core tasks (b), assessing supervised baselines, DNA and RNA pLMs with classifier modules (simple MLP, shallow CNN, and ResNet) (c) through diverse evaluation metrics and multi-dimensional analysis (d).

strengths and limitations related to un-pre-trained models from basic binary classification and more complex multiclass characterization.

• **Sustainable Framework Development.** RNAscope presents an open-source platform with public leaderboards (`https://rnascope-board.github.io/`), along with available datasets and code, promoting fair, transparent, and standardized model comparisons. By continually incorporating upcoming datasets, it will foster collaboration in the development of robust RNA pLMs, thereby engineering RNA biology.

## 2 Background

### 2.1 Structural, Interaction, and Functional Complexities of RNA Sequences

RNA molecules are sequences that fold into complex spatial structures, interact with diverse molecular partners, and carried out varied biological functions [70, 25]. Investigating RNA sequences to understand their structures, interactions, and functions is essential for advancing RNA biology and its applications [82, 13, 39].

In many cases, RNA molecules display a high degree of sequence-based evolutionary conservation across species, such as at splicing sites [3]. However, the complexity of the dynamic RNA regulation network extends far beyond these predominant RNA features, involving versatile structural conformations [64], intermolecular binding for both stable and transient events [21, 79], and coordination of both pervasive and cryptic functional patterns [29]. To address this complexity, it is essential to develop a more comprehensive framework that incorporates a wide range of RNA pattern propensities.

### 2.2 Advancements in RNA pLMs

In recent years, deep learning has substantially advanced bioinformatics, leading to the development of pLMs tailored for biological sequences. RNA pLMs—such as RNABERT[1], RNA-FM[10], SpliceBERT[11], 3UTRBERT[76], UTR-LM[14] and RiNALMo[46]—leverage large-scale RNA sequence data to learn sophisticated representations that encapsulate both sequence patterns and broader contextual information. These models, typically based on transformer encoder architectures, are pre-trained on specific RNA species to address various aspects of RNA biology.

Despite their successful applications, the scope of evaluation for these models is often limited to their specific tasks. This highlights the need for a comprehensive assessment of their capabilities across a broader spectrum of RNA biology tasks. Moreover, RNA transcripts are dynamically synthesized and

processed from the same DNA template into multiple isoforms, reflecting intricate layers of cellular regulation [81, 59]. Recent advancements in DNA pLMs, such as DNABERT2[80], HyenaDNA[43], and Nucleotide Transformer[16], have also enhanced the modeling of DNA sequences, helping to identify genome-wide functional patterns. Exploring the potential of these DNA pLMs in RNA-specific tasks represents another promising avenue for future research.

## 2.3 Existing RNA benchmarking frameworks

High-throughput biotechniques have greatly expanded the availability of RNA data in multiple domains. However, RNA benchmarking frameworks remain limited in scope and depth. Existing benchmarks, such as RnaBench [54], primarily focus on RNA secondary structure prediction and their molecular design task. Although BEACON [50] extends the evaluation to include function and engineering tasks, it still lacks effective assessment that account for RNA contextual properties, such as RNA interaction in the presence of other molecules. Moreover, these benchmarks often overlook the complexity of RNA across different fields, resulting in a fragmented understanding of RNA biology and its modeling challenges (**see detailed description in Appendix B**). To effectively evaluate RNA pLMs, further benchmarking frameworks need to integrate large-scale, versatile datasets that comprehensively represent the structural diversity and functional flexibility of RNA molecules.

## 3 Task Definition and Objectives

RNA molecules are often evolutionarily conserved in certain regions, reflecting their essential functions. However, RNA structure and behavior are highly contextual, displaying significant dynamism and diversity. Understanding the complex relationship between RNA sequences and their structures, interactions, and functions remains a major challenge. To address this, RNAscope introduces three hierarchically structured cohorts of sub-tasks for **evaluating RNA pLMs on their ability to capture and generalize RNA structural, interaction, and functional features beyond predominant evolutionary conservation across diverse biological contexts**. Detailed descriptions and dataset construction are provided in the Appendix C.

### 3.1 Structure-related Tasks

*Key Points:* RNA sequences determine structures. This panel outlines tasks for inferring RNA structure from one-dimensional (1D) sequence, including **secondary structure prediction** for base-pairing likelihood, **chemical reactivity prediction** for structural dynamics and nucleotide accessibility, and **contact map prediction** for spatial interactions essential to three-dimensional topology. Together, these tasks establish a sub-framework for understanding the structural diversity of RNA across families.

**(1) Secondary Structure Prediction (SSP)** is a binary classification task that determines the pairing status $y_i \in \{0, 1\}$ of each nucleotide $x_i$ within an RNA sequence, thus characterizing base-pairing conditions [28, 51].

*Data Split and Analysis*: Evaluation is conducted on three datasets: bpRNA, SetA, and SetB. bpRNA comprises bpRNA-1m (split into TR0, VL0, and TS0 for intra-family evaluation) and bpRNA-new (containing novel families for inter-family evaluation). SetA includes eight canonical RNA families, and SetB contains 22 structurally diverse families, each with independent train-test splits. To assess cross-family generalization, TestSetA and TestSetB are also used to evaluate models trained on the other set. All datasets are de-redundified based on sequence identity.

**(2) Chemical Reactivity Prediction (CRP)** is a regression task that predicts the chemical reactivity $y_i$ of each nucleotide $x_i$ in an RNA molecule within the range $[0, 1]$, primarily influenced by its secondary structure, where unpaired regions typically are more chemically reactive. Reactivity measurements reflect RNA's secondary structure and dynamic conformational changes [69, 71], thereby representing molecular conformations.

*Data split and analysis*: The dataset is derived from publicly available data from the 'Stanford Ribonanza RNA Folding competition' on Kaggle [19]. To assess the model's generalizability,

Table 1: Summary of 15 benchmark tasks across RNA structure, interaction and function in RNAscope. See Appendix C and D for detailed dataset and metric information.

| Task Type | Task Name | Dataset (Source) | Train | Validation | Test | Metric |
|---|---|---|---|---|---|---|
| **Structure (3.1)** | Second Structure Prediction | bpRNA [55] | 10,814 | 1,300 | 6,706 | Precision, Recall, Binary F1 |
| | | SetA [55] | 2,850 | 316 | 1,022 | Precision, Recall, Binary F1 |
| | | SetB [55] | 985 | 109 | 1,022 | Precision, Recall, Binary F1 |
| | Chemical Reactivity Prediction | 2A3-MaP [19] | 144,918 | 16,103 | 6,787 | MAE |
| | Contact Map Prediction | RNA3DB [61] | 9,101 | 1,011 | 1,375 | Short@L/5, Long@L/5 |
| **Interaction (3.2)** | Binary Binding Prediction | 22 RBPs [76] | 331,591 | 110,537 | 110,557 | F1 |
| | Systematic Binding Ranking | DAse [78] | 65,149 | 10,858 | 32,576 | Macro F1 |
| | | ISLETS [78] | 188,159 | 31,360 | 94,080 | Macro F1 |
| | | TARDBP [78] | 132,779 | 22,130 | 39,390 | Macro F1 |
| | Binding Affinity Prediction | GFP [63] | 189 | 48 | 181 | Spearman $p$ |
| | | NELF [63] | 168 | 43 | 2,442 | Spearman $p$ |
| **Function (3.3)** | Splicing Site Prediction | Donor [38] | 169,798 | 42,448 | 316,282 | Accuracy |
| | | Acceptor [38] | 164,946 | 41,236 | 315,268 | Accuracy |
| | Splicing Event Prediction | DeepASmRNA [7] | 52,008 | 17,313 | 32,052 | Macro F1 |
| | Polyadenylation Signal Prediction | DeeReCT-PolyA [74] | 22,536 | 7,514 | 78,291 | Accuracy |
| | Coding Potential Prediction | CPPred [65] | 51,770 | 5,753 | 122,710 | Accuracy |
| | mRNA Subcellular Localization | Allocator [36] | 13,838 | 1,730 | 1,730 | Macro F1 |
| | Ribosome Loading Prediction | HepG2 [8] | 759,594 | 10,000 | 20,000 | $R^2$ |
| | ncRNA Category Classification | Ncypred [37] | 28,626 | 3,175 | 13,646 | Accuracy |
| | microRNA Subcellular Localization | RNALocate [2] | 373 | 47 | 118 | subACC, Hloss |
| | gRNA Efficiency Prediction | gRNA Data[34] | 6,134 | 1,480 | 13631 | Spearman $p$ |

training samples include sequences of lengths {170, 177}, while the test sets TestS and TestL consist of sequences with lengths {115, 155} and {206}, respectively.

**(3) Contact Map Prediction (CMP)** assigns a label $y_{ij} \in \{0, 1\}$ to each nucleotide pair $(x_i, x_j)$, indicating whether they are within 8Å in three-dimensional space. The inherent flexibility of RNA leads to complex three-dimensional structures, which present significant challenges for accurate contact map prediction.

*Data split and analysis*: The dataset originates from RNA3DB [61], which comprises almost all 3D RNA structures obtained from the Protein Data Bank (PDB). It employs strict partitioning to eliminate sequence and structural redundancy between training and test sets, ensuring robust evaluation of the model's ability.

## 3.2 Interaction-related Tasks

*Key Points:* Both naturally and artificially evolved RNA sequences adopt defined structural shapes, enabling high-specificity interactions with a wide array of target molecules. This panel outlines tasks for inferring RNA interaction with different targets from 1D sequence, including **binary binding prediction** for RNA-proteins interactions within cells; **systematic binding ranking** for categorizing RNA interaction with targets of varying molecular sizes; and **binding affinity prediction** for RNA-protein interaction with varying strengths in vitro. Together, these tasks establish a sub-framework for understanding the interaction diversity of RNA across targets, encompassing both persistent and transient interaction.

**(1) Binary Binding Prediction** (**BBP**) is a binary classification task that predicts whether an RNA sequence interacts with the RNA-binding protein (RBP). Studies have shown that RNAs involved in these interactions often have conserved binding motifs in their sequences.

*Data split and analysis:* The original data is sourced from eCLIP experiments, comprising 22 datasets corresponding to 22 RBPs across K562 and HepG2 cell lines [40]. All sequences are unified to 100 nucleotides in length, with samples having over 80% sequence identity removed [76]. The positive-to-negative sample ratio is maintained at 1:2 for each RBP dataset.

**(2) Systematic Binding Ranking** (**SBR**) is a multi-label classification task that ranks the binding categories of RNA species targeting different molecular targets. This task aims to assess the ability of pLMs to understand the binding capacity of these species across multiple targets.

*Data split and analysis:* The data, derived from SELEX experiments [18] and curated by UltraGen[78], includes three datasets: DAse, TARDBP, and ISLETS. Each dataset targets different types of molecules, including small molecules, proteins, and (multi)cellular systems.

**(3) Binding Affinity Prediction** (**BAP**) is a regression task that quantifies the binding affinity of RNA sequences. It aims to assess how nucleotide mutations influence binding affinity and map the RNA-protein interaction landscape.

Table 2: Overview of the RNA and DNA pLMs evaluated in RNAscope. The representative versions of the models from multiple checkpoints were compared (see Appendix E for details).

| Model Type | Model Name | Pre-trained Data | Params. | Tokenizer | Embedding Size | Architecture |
|---|---|---|---|---|---|---|
| RNA pLMs | RNABERT [1] | Human ncRNA | 0.48M | Single Base | 120 | BERT |
| | RNA-FM [10] | Multispecies ncRNA | 100M | Single Base | 640 | BERT |
| | 3UTRBERT [76] | Human 3'UTR | 86M | k-mer[†] | 768 | BERT |
| | SpliceBERT [11] | Multispecies pre-mRNA | 20M | Single Base | 512 | BERT |
| | UTR-LM [14] | Multispecies 5'UTR | 1.21M | Single Base | 128 | BERT |
| | RiNALMo [46] | Multispecies ncRNA | 650M | Single Base | 1280 | BERT |
| DNA pLMs | DNABERT2 [80] | Multispecies genomes | 117M | BPE | 768 | BERT |
| | HyenaDNA [43] | Human genome | 1.6M | Single Base | 256 | Hyena |
| | Nucleotide Transformer (NT)[16] | Diverse human genomes | 480M | k-mer[‡] | 1280 | BERT |

Note: † Overlapping k-mer (stride 1, allowing overlap), ‡ Nonoverlapping k-mer (stride k, no overlap).

*Data split and analysis:* The data, from HiTS-RAP experiments, comprise two datasets: GFP and NELF [63]. These experiments measure the binding affinity of mutagenized aptamers of GFPapt [57] and NELEapt [45]. Following a setup similar to Flip [17], wild-type and single mutants are used for training, while double mutations form the test set, enabling a rigorous evaluation of the model's ability to understand the binding mechanism.

## 3.3 Function-related Tasks

*Key Points:* Functional RNA demands both conserved structure and specific environment. This panel outlines tasks for inferring RNA functions with different spatiotemporal conditions from 1D sequence across pre-mRNA, mRNA, and ncRNA, including **coding and non-coding classification** for general sequence motif conservation; **subcellular localization** for sequence-derived spatial conservation; **splicing and polyadenylation identification** for nuclear RNA maturation; ribosome loading estimation for cytosolic mRNA translation; **genome editing efficiency** for cellular on-target environment of guide RNA (gRNA). Together, these tasks establish a sub-framework for understanding the evolutionary function diversity of RNA across conservation levels and contexts.

### 3.3.1 Pre-mRNA-related Function Tasks

**(1) Splicing Site Prediction (SPS)** comprises two binary classification tasks: distinguishing whether a sequence corresponds to a splice donor or acceptor site. SPS is central to pre-mRNA processing, removing non-coding regions and joining coding regions by recognizing donor and acceptor sites.

*Data split and analysis*: The datasets for predicting splice donor and acceptor sites are curated from the genome sequences of human, Ara (arabidopsis), and rice, with redundant sequences removed [38]. Training is performed on the human dataset, while evaluation is conducted on the human, Ara and rice to assess cross-species generalization. All test sets are ensured to have less than 80% sequence identity with the training set.

**(2) Splicing Event Prediction (SPE)** is a multi-label classification task that maps RNA sequence $x$ to the splicing event label $y \in \{ES, AA, AD, IR\}$ [1]. SPE further reveals the diversity of splicing mechanisms.

*Data split and analysis*: The datasets are sourced from the RNA sequences of huamn, Ara, and rice species [7], with models trained on the human dataset and evaluated on all three species to assess cross-species performance. The sequence identity between the test and training sets is kept below 80%.

**(3) Polyadenylation Signal Prediction (PAS)** is a binary classification task that predicts the presence of the polyadenylation signal (PAS), a hexamal motif upstream of the RNA 3'-end cleavage site, critical for mRNA maturation.

*Data split and analysis*: The datasets from [74] include 18,786 true PAS sequences from 12 human individuals, balanced with pseudo-PAS sequences. Models are trained on human data and evaluated on poly(A) datasets from C57BL/6J (`mouse_bl`) and SPRET/EiJ (`mouse_sp`) mice, with test sets maintaining <80% sequence identity with the training set.

### 3.3.2 mRNA-related Function Tasks

**(1) Coding Potential Prediction (CPP)** is a binary classification task that distinguishes coding RNA from non-coding ones. This task aims to enchance transcript coding potential for mRNA drug design.

---

[1]ES: Exon Skipping, AA: Alternative Acceptor site, AD: Alternative Donor site, IR: Intron Retention.

*Data split and analysis*: The dataset, sourced from [65], consists of coding and non-coding sequences from human, mouse, zebrafish, fruit fly, and yeast. Training was conducted on the human dataset, with validation performed across all five species.

**(2) mRNA Subcellular Localization** (**mSL**) is a multi-label classification task that maps an mRNA sequence $x$ to a set of localization labels $y$ [2]. This task predicts the spatial distribution patterns of mRNAs, which interprets the contextual environment for protein synthesis and cellular processes.

*Data split and analysis*: The benchmark dataset from [36], contains 17,298 unique human mRNA sequences. Sequences were divided into training, validation, and testing sets in an 8:1:1 ratio, mirroring the original multi-label distribution.

**(3) Ribosome Loading Prediction** (**RLP**) is a regression task that predicts the Mean Ribosome Load (MRL) for the given 5' untranslated region (5'UTR), aiming to engineer optimization of translation efficiency by estimating ribosome occupancy.

*Data split and analysis*: The dataset, from MRL measurements in HepG2 cells [8], ranks 5'UTRs by sequencing read counts, assigning the top 20,000 to testing, 10,000 to validation, and the remainder to training.

### 3.3.3 ncRNA-related Function Tasks

**(1) ncRNA Category Classification** (**NCC**) is a multi-label classification task that assigns ncRNA sequences to 13 categories. This classification aims to feature conserved sequence patterns from different types of ncRNA, aiding in the understanding of RNA involved regulatory roles beyond protein production.

*Data split and analysis*: The dataset, sourced from [37], includes 31,000 non-coding RNA sequences obtained from the Rfam database [33]. A 20% non-redundant subset is used to form the testing set.

**(2) microRNA Subcellular Localization** (**miSL**) is a multi-label classification task that predicts the subcellular localization of microRNAs [3]. Accurate localization prediction is crucial for understanding microRNA function and informing drug design.

*Data split and analysis*: The dataset, sourced from RNALocate v2.0 [15] and constructed by [2], consists of 538 unique microRNAs, with redundancy removed using an 80% identity threshold.

**(3) gRNA Efficiency Prediction** (**gEP**) is a regression task that predicts the cleavage efficiency of guide RNAs (gRNAs) in the CRISPR-Cas9 system, aiming to engineer the optimization of gene editing by enhancing on-target efficiency.

*Data split and analysis*: The dataset, from [34], includes five public gRNA efficiency datasets for training. Testing is conducted on six additional datasets covering various cell lines and organisms (human, mouse, zebrafish) for unbiased evaluation.

## 4 Experiments

**Pre-trained Language Models (pLMs).** We evaluated a diverse set of RNA pLMs, including RNABERT, RNA-FM, 3UTRBERT, SpliceBERT, UTR-LM, and RiNALMo. Among them, RNABERT, RNA-FM, and RiNALMo were pre-trained on various non-coding RNAs (ncRNAs), while 3UTRBERT and UTR-LM focused on the 3' and 5' untranslated regions (UTRs) of mRNA, respectively. SpliceBERT was pre-trained on pre-mRNA sequences. To assess cross-modality transfer, we also included three state-of-the-art DNA pLMs—DNABERT2, HyenaDNA, and the Nucleotide Transformer—all trained on genomic DNA. This comparison provides insight into how pre-training domains influence performance on RNA-specific tasks. Additional model details are provided in Appendix E.

**Downstream Modules.** Three lightweight supervised modules were evaluated on frozen language model (LM) embeddings—a simple MLP, a shallow CNN, and a 24-layer ResNet—to consistently assess the capabilities and performance of various LMs across tasks. For each task, we frozen the

---

[2]Localization labels include nucleus, cytoplasm, exosome, ribosome, membrane, and mitochondria.

[3]The four subcellular localizations: nucleus, exosome, cytoplasm, and microvesicle.

backbones of pre-trained LMs and fine-tune only the downstream modules, enabling a comparable evaluation of embeddings across diverse architectures and parameter scales.

**Supervised Baselines.** A 24-layer ResNet was employed as a supervised baseline for each task. Inputs were encoded using two distinct methods: (1) one-hot encoding, which represents sequences as sparse binary vectors indicating the presence of specific tokens, and (2) 512-dimensional dense embeddings, which mapped sequences to a continuous feature space derived from a learned representation. These approaches, referred to as *One-hot* and *Dense*, respectively, enabled a comparative analysis of how input encoding schemes influence model performance.

**Traning Setups.** Models were optimized using AdamW (learning rate set 1e-4, weight decay 0.01). Training was employed an early stopping strategy, and was terminated if validation performance shows no improvement over 10 consecutive evaluations. For sequence-level tasks, average pooling was applied to aggregate sequence representations. To ensure compatibility and fair comparison across tasks, we set the maximum input length to 1024 nucleotides. For robust and reliable evaluation, each model was independently trained three times using different random seeds for each specific task.

# 5 Results and Discussion

## 5.1 RNA Sequence to Structure

*Key notes:* **RNA pLMs exhibit limited capacity to capture RNA's structural complexity**, as shown in the benchmark results for structure tasks in Table 16. While effective at modeling conserved intra-family base-pairing, further advancements are needed to enhance inter-family generalization and to capture the structural dynamics and spatial folding essential for three-dimensional topology.

Table 3: Benchmark results for RNA structure tasks. Reported as *mean (std)* of three runs with different seeds. The best and second-best models for each test set are highlighted in two shades of green. Classifier modules—simple MLP, shallow CNN, and ResNet—are used with frozen pLM weights. Results reflecting the best-performing classifier module for each dataset are additionally underlined.

| Model | Second Structure | | | | | | Chemical Reactivity | | Contact Map | |
|---|---|---|---|---|---|---|---|---|---|---|
| | $\text{bpRNA}_{\text{Intra-family}}$ F1↑ | $\text{bpRNA}_{\text{Inter-family}}$ F1↑ | $\text{SetA}_{\text{Intra-family}}$ F1↑ | $\text{SetA}_{\text{Inter-family}}$ F1↑ | $\text{SetB}_{\text{Intra-family}}$ F1↑ | $\text{SetB}_{\text{Inter-family}}$ F1↑ | TestS MAE↓ | TestL MAE↓ | Short@L/5↑ | Long@L/5↑ |
| One-hot | $0.548_{(0.003)}$ | $0.482_{(0.006)}$ | $0.689_{(0.007)}$ | $0.343_{(0.014)}$ | $0.369_{(0.008)}$ | $0.510_{(0.021)}$ | $0.179_{(0.001)}$ | $0.167_{(0.002)}$ | $0.158_{(0.006)}$ | $0.168_{(0.007)}$ |
| Dense | $0.552_{(0.002)}$ | $0.501_{(0.007)}$ | $0.695_{(0.005)}$ | $0.348_{(0.010)}$ | $0.384_{(0.006)}$ | $0.526_{(0.027)}$ | $0.176_{(0.001)}$ | $0.174_{(0.002)}$ | $0.190_{(0.015)}$ | $0.170_{(0.006)}$ |
| **(+MLP)** | | | | | | | | | | |
| RNABERT_mlp | $0.551_{(0.004)}$ | $0.541_{(0.003)}$ | $0.700_{(0.001)}$ | $0.607_{(0.000)}$ | $0.518_{(0.002)}$ | $0.524_{(0.004)}$ | $0.255_{(0.000)}$ | $0.266_{(0.001)}$ | $0.013_{(0.001)}$ | $0.021_{(0.003)}$ |
| RNA-FM_mlp | $0.766_{(0.004)}$ | $0.608_{(0.002)}$ | $0.837_{(0.001)}$ | $0.685_{(0.001)}$ | $0.869_{(0.000)}$ | $0.605_{(0.006)}$ | $0.214_{(0.001)}$ | $0.187_{(0.003)}$ | $0.137_{(0.003)}$ | $0.129_{(0.002)}$ |
| 3UTRBERT_mlp | $0.647_{(0.012)}$ | $0.597_{(0.003)}$ | $0.746_{(0.004)}$ | $0.598_{(0.004)}$ | $0.673_{(0.005)}$ | $0.512_{(0.009)}$ | $0.202_{(0.001)}$ | $0.195_{(0.002)}$ | $0.103_{(0.002)}$ | $0.086_{(0.003)}$ |
| SpliceBERT_mlp | $0.651_{(0.024)}$ | $0.586_{(0.011)}$ | $0.760_{(0.001)}$ | $0.587_{(0.010)}$ | $0.717_{(0.005)}$ | $0.548_{(0.022)}$ | $0.207_{(0.001)}$ | $0.207_{(0.001)}$ | $0.111_{(0.002)}$ | $0.121_{(0.002)}$ |
| UTR-LM_mlp | $0.621_{(0.007)}$ | $0.585_{(0.005)}$ | $0.723_{(0.005)}$ | $0.597_{(0.005)}$ | $0.653_{(0.016)}$ | $0.555_{(0.017)}$ | $0.201_{(0.001)}$ | $0.196_{(0.003)}$ | $0.111_{(0.002)}$ | $0.097_{(0.002)}$ |
| RiNALMo_mlp | $0.797_{(0.004)}$ | $0.625_{(0.006)}$ | $0.881_{(0.002)}$ | $0.713_{(0.003)}$ | $0.894_{(0.003)}$ | $0.726_{(0.006)}$ | $0.183_{(0.001)}$ | $0.195_{(0.002)}$ | $0.173_{(0.001)}$ | $0.164_{(0.002)}$ |
| **(+CNN)** | | | | | | | | | | |
| RNABERT_cnn | $0.598_{(0.003)}$ | $0.571_{(0.005)}$ | $0.696_{(0.003)}$ | $0.599_{(0.010)}$ | $0.575_{(0.008)}$ | $0.561_{(0.015)}$ | $0.228_{(0.000)}$ | $0.241_{(0.002)}$ | $0.117_{(0.005)}$ | $0.102_{(0.002)}$ |
| RNA-FM_cnn | $0.786_{(0.001)}$ | $0.590_{(0.005)}$ | $0.842_{(0.001)}$ | $0.685_{(0.005)}$ | $0.874_{(0.001)}$ | $0.569_{(0.007)}$ | $0.197_{(0.003)}$ | $0.176_{(0.001)}$ | $0.141_{(0.005)}$ | $0.150_{(0.004)}$ |
| 3UTRBERT_cnn | $0.672_{(0.005)}$ | $0.588_{(0.016)}$ | $0.762_{(0.002)}$ | $0.592_{(0.004)}$ | $0.704_{(0.001)}$ | $0.486_{(0.005)}$ | $0.192_{(0.000)}$ | $0.182_{(0.002)}$ | $0.138_{(0.004)}$ | $0.109_{(0.005)}$ |
| SpliceBERT_cnn | $0.691_{(0.003)}$ | $0.565_{(0.004)}$ | $0.774_{(0.000)}$ | $0.579_{(0.006)}$ | $0.737_{(0.008)}$ | $0.513_{(0.011)}$ | $0.196_{(0.001)}$ | $0.179_{(0.002)}$ | $0.150_{(0.007)}$ | $0.140_{(0.004)}$ |
| UTR-LM_cnn | $0.668_{(0.003)}$ | $0.595_{(0.007)}$ | $0.743_{(0.003)}$ | $0.587_{(0.003)}$ | $0.688_{(0.006)}$ | $0.486_{(0.003)}$ | $0.191_{(0.001)}$ | $0.176_{(0.007)}$ | $0.152_{(0.005)}$ | $0.132_{(0.003)}$ |
| RiNALMo_cnn | $0.803_{(0.000)}$ | $0.605_{(0.006)}$ | $0.881_{(0.002)}$ | $0.706_{(0.007)}$ | $0.903_{(0.002)}$ | $0.708_{(0.002)}$ | $0.173_{(0.001)}$ | $0.179_{(0.002)}$ | $0.182_{(0.002)}$ | $0.146_{(0.003)}$ |
| **(+ResNet)** | | | | | | | | | | |
| RNABERT_resnet | $0.685_{(0.003)}$ | $0.576_{(0.015)}$ | $0.743_{(0.006)}$ | $0.579_{(0.009)}$ | $0.670_{(0.004)}$ | $0.478_{(0.013)}$ | $0.181_{(0.002)}$ | $0.175_{(0.003)}$ | $0.084_{(0.003)}$ | $0.083_{(0.003)}$ |
| RNA-FM_resnet | $0.775_{(0.001)}$ | $0.607_{(0.001)}$ | $0.820_{(0.001)}$ | $0.663_{(0.008)}$ | $0.860_{(0.006)}$ | $0.591_{(0.023)}$ | $0.196_{(0.004)}$ | $0.166_{(0.004)}$ | $0.145_{(0.008)}$ | $0.150_{(0.007)}$ |
| 3UTRBERT_resnet | $0.685_{(0.007)}$ | $0.576_{(0.015)}$ | $0.746_{(0.032)}$ | $0.573_{(0.008)}$ | $0.695_{(0.009)}$ | $0.498_{(0.032)}$ | $0.187_{(0.002)}$ | $0.172_{(0.004)}$ | $0.137_{(0.016)}$ | $0.122_{(0.002)}$ |
| SpliceBERT_resnet | $0.695_{(0.006)}$ | $0.574_{(0.005)}$ | $0.772_{(0.004)}$ | $0.579_{(0.014)}$ | $0.712_{(0.003)}$ | $0.561_{(0.003)}$ | $0.193_{(0.004)}$ | $0.177_{(0.003)}$ | $0.155_{(0.002)}$ | $0.161_{(0.009)}$ |
| UTR-LM_resnet | $0.697_{(0.003)}$ | $0.568_{(0.014)}$ | $0.756_{(0.008)}$ | $0.569_{(0.013)}$ | $0.716_{(0.012)}$ | $0.472_{(0.042)}$ | $0.180_{(0.001)}$ | $0.171_{(0.005)}$ | $0.075_{(0.006)}$ | $0.101_{(0.002)}$ |
| RiNALMo_resnet | $0.805_{(0.003)}$ | $0.613_{(0.014)}$ | $0.882_{(0.002)}$ | $0.707_{(0.007)}$ | $0.897_{(0.001)}$ | $0.672_{(0.015)}$ | $0.163_{(0.000)}$ | $0.163_{(0.004)}$ | $0.133_{(0.002)}$ | $0.139_{(0.005)}$ |

As shown in Table 16, RNA pLMs outperform baseline models on intra-family secondary structure prediction, with performance further enhanced by deeper classifiers such as CNN and ResNet. RNA-FM and RiNALMo were pre-trained on large-scale ncRNA datasets (e.g., RNAcentral includes both endogenous and synthetic RNAs), achieving the highest average F1 scores in three benchmark datasets—0.834 and 0.861, respectively—surpassing the Dense baseline (0.552) by over 50%.

However, these improvements fail to generalize to inter-family settings, with F1 scores dropping by over 20% compared to intra-family results. Training curves (Fig.F.1) reveal that improvements in validation performance are often accompanied by declines in inter-family accuracy, suggesting that models may rely on family-specific features rather than broadly conserved base-pairing patterns. In more complex tasks, such as chemical reactivity, RNA pLMs exhibit reduced advantages; in contact map prediction, they are consistently outperformed by simpler Dense baselines, highlighting their limitations in modeling non-canonical interactions and long-range structural dependencies.

 **5.2 RNA Sequence to Interaction**

*Key notes:* **RNA pLMs show higher performance in predicting *in vivo* RNA-target interaction than in *in vitro*.** This disparity indicates that recognition of *in vivo* evolutionary patterns related to interaction contexts is more conserved. In contrast, predicting *in vitro* interactions in the absence of such evolutionary constraints, including binding strengths and affinity scores across diverse molecular targets, remains a challenge. These findings imply that current models may underestimate the full complexity of RNA interactome that arised from different evolutionary phases. Detailed benchmark comparison for interaction tasks are presented in Table 4.

In predicting endogenous RNA–RBP binary-binding tasks, RNA pLMs demonstrate a higher accuracy compared to baseline models. Using only a simple MLP probe, SpliceBERT achieves an average F1 score of 0.736 across 22 RBP datasets, with further gains observed when employing deeper classifier modules. This improvement may imply the conserved nature of RBP-binding sites, which are often defined by short, contiguous 5-mer motifs [20, 67]—patterns well captured by nucleotide-level tokenization. In contrast, subword tokenization strategies in DNA pLMs, such as BPE or non-overlapping k-mers (e.g., in DNABERT2 and NT), tend to fragment such motifs, resulting in lower performance than even simple baselines. Additionally, RNA-protein binding sites are widely distributed across the transcriptome and are frequently enriched in intronic regions. SpliceBERT's pretraining on pre-mRNA, which incorporates an extensive intronic content [68, 67], may contribute to its ability to capture these contextual binding patterns.

In contrast, RNA pLMs yield limited gains over baseline models on *in vitro* tasks, where the evolutionary constraints are greatly reduced and sequence conservation is minimal. In the systematic binding task derived from SELEX datasets, RNA pLMs underperform on protein (*TARDBP*) and (multi)cellular (*ISLETS*) targets, and yield only modest gains on small-molecule (*DAse*) binding. These trends consistently align with prior findings [78]: small-molecule binders exhibit distinct sequence features that separate them from non-binders, whereas for protein (*TARDBP*) and cellular (*ISLETS*) targets, distinguishing binders from non-binders is substantially more challenging due to the lack of distinctive sequence features.

Similarly, in the binding affinity regression task, most RNA and DNA pLMs underperforme than the Dense baseline on the *NELF* dataset. On the *GFP* dataset, probing with a simple MLP yields low correlation coefficients ($< 0.14$). Although model performance improves with deeper classifier heads, noticeable standard deviations across runs underscore the model sensitivity to experimental conditions, suggesting limited robustness in capturing fine-grained affinity variation under weak supervision.

Table 4: Benchmark results on interaction tasks. Results are *mean (std)* from three runs with different seeds, with the best and second-best models highlighted. Binary binding classification task comprises 22 RBP datasets, with average performance reported. Further detailed in Tables 17 and 18 in the Appendix.

| Model | Binary Binding 22 RBP's Average F1↑ | Systematic Binding DAse F1↑ | Systematic Binding TARDBP F1↑ | Systematic Binding ISLETS F1↑ | Binding Affinity GFP Spear $p$↑ | Binding Affinity NELF Spear $p$↑ |
|---|---|---|---|---|---|---|
| One-hot | 0.703 | 0.614(0.005) | 0.461(0.010) | 0.415(0.003) | 0.215(0.008) | 0.668(0.130) |
| Dense | 0.704 | 0.622(0.001) | 0.472(0.003) | 0.412(0.002) | 0.138(0.041) | 0.388(0.059) |
| **(+MLP)** | | | | | | |
| DNABERT2_mlp | 0.683 | 0.629(0.003) | 0.425(0.005) | 0.403(0.002) | -0.124(0.044) | 0.148(0.010) |
| HyenaDNA_mlp | 0.640 | 0.522(0.004) | 0.376(0.003) | 0.317(0.015) | 0.092(0.063) | 0.140(0.005) |
| NT_mlp | 0.666 | 0.599(0.010) | 0.391(0.001) | 0.360(0.035) | 0.139(0.050) | 0.314(0.004) |
| RNABERT_mlp | 0.539 | 0.528(0.006) | 0.307(0.006) | 0.309(0.007) | -0.049(0.036) | 0.129(0.005) |
| RNA-FM_mlp | 0.698 | 0.605(0.006) | 0.396(0.006) | 0.369(0.012) | 0.073(0.006) | 0.331(0.009) |
| 3UTRBERT_mlp | 0.716 | 0.611(0.007) | 0.401(0.005) | 0.397(0.005) | -0.043(0.005) | 0.317(0.046) |
| SpliceBERT_mlp | 0.736 | 0.620(0.012) | 0.396(0.007) | 0.367(0.012) | 0.079(0.010) | 0.298(0.003) |
| UTR-LM_mlp | 0.682 | 0.536(0.015) | 0.366(0.010) | 0.299(0.031) | 0.028(0.087) | 0.134(0.072) |
| RiNALMo_mlp | 0.726 | 0.571(0.002) | 0.373(0.004) | 0.332(0.004) | 0.076(0.070) | 0.157(0.018) |
| **(+CNN)** | | | | | | |
| DNABERT2_cnn | 0.685 | 0.640(0.002) | 0.456(0.001) | 0.414(0.002) | -0.030(0.002) | 0.241(0.021) |
| HyenaDNA_cnn | 0.697 | 0.597(0.006) | 0.438(0.004) | 0.390(0.007) | 0.139(0.011) | 0.181(0.009) |
| NT_cnn | 0.674 | 0.622(0.012) | 0.448(0.006) | 0.396(0.002) | 0.138(0.017) | 0.259(0.007) |
| RNABERT_cnn | 0.642 | 0.538(0.002) | 0.414(0.008) | 0.341(0.007) | -0.016(0.003) | 0.135(0.003) |
| RNA-FM_cnn | 0.724 | 0.640(0.016) | 0.460(0.013) | 0.410(0.004) | 0.224(0.073) | 0.384(0.002) |
| 3UTRBERT_cnn | 0.728 | 0.642(0.002) | 0.462(0.007) | 0.403(0.003) | 0.207(0.184) | 0.194(0.008) |
| SpliceBERT_cnn | 0.748 | 0.639(0.006) | 0.466(0.003) | 0.414(0.003) | 0.107(0.011) | 0.340(0.007) |
| UTR-LM_cnn | 0.728 | 0.571(0.019) | 0.432(0.004) | 0.372(0.012) | 0.037(0.041) | 0.244(0.069) |
| RiNALMo_cnn | 0.749 | 0.595(0.008) | 0.441(0.003) | 0.370(0.009) | 0.067(0.061) | 0.161(0.011) |
| **(+ResNet)** | | | | | | |
| DNABERT2_resnet | 0.669 | 0.617(0.009) | 0.438(0.002) | 0.401(0.006) | 0.267(0.014) | 0.580(0.029) |
| HyenaDNA_resnet | 0.727 | 0.623(0.005) | 0.471(0.003) | 0.411(0.005) | 0.376(0.032) | 0.625(0.143) |
| NT_resnet | 0.681 | 0.638(0.013) | 0.454(0.004) | 0.413(0.003) | 0.334(0.009) | 0.626(0.015) |
| RNABERT_resnet | 0.675 | 0.599(0.003) | 0.461(0.003) | 0.404(0.002) | 0.029(0.165) | 0.677(0.052) |
| RNA-FM_resnet | 0.719 | 0.634(0.004) | 0.460(0.008) | 0.409(0.001) | 0.407(0.046) | 0.646(0.059) |
| 3UTRBERT_resnet | 0.728 | 0.624(0.009) | 0.468(0.008) | 0.414(0.002) | 0.262(0.165) | 0.498(0.213) |
| SpliceBERT_resnet | 0.750 | 0.617(0.013) | 0.456(0.002) | 0.406(0.003) | 0.162(0.071) | 0.510(0.040) |
| UTR-LM_resnet | 0.723 | 0.597(0.005) | 0.446(0.004) | 0.372(0.011) | 0.233(0.098) | 0.671(0.050) |
| RiNALMo_resnet | 0.750 | 0.564(0.031) | 0.448(0.004) | 0.381(0.005) | 0.109(0.093) | 0.256(0.096) |

**5.3 RNA Sequence to Function**

*Key notes:* **RNA pLMs present remarkable capabilities for characterizing RNA biological processes than engineering.** Consistent with their leading performance on the interaction tasks, pLMs demonstrated greater improvement in cellular processes than engineering design that beyond the evolutionary frame when compared to the baseline. As expected, DNA-based pLMs did not outperform RNA-based pLMs on RNA-centric tasks. For model optimization with various downstream modules, shallow CNNs leveraging full-sequence embeddings outperform MLPs that rely on pooled embeddings from pLMs. Additionally, the performance of pLMs closely align with their pre-training

source, showcasing their strong capability in task-specific studies. Table 5 further presents benchmark results for function tasks across pre-mRNA, mRNA and ncRNA.

Table 5: Benchmark results for function tasks across pre-mRNA, mRNA, and ncRNA. Results are presented as *mean (std)* from three runs with different seeds, highlighting the best and second-best models. For additional results, please refer to Table 19 in Appendix. The gRNA Efficiency Prediction comprises six test sets, with averages presented here and details Tabel 20 in Appendix.

| Model | pre-mRNA Tasks | | | | | | | | mRNA Tasks | | | | ncRNA Tasks | | |
|---|---|---|---|---|---|---|---|---|---|---|---|---|---|---|---|
| | Splicing Site | | | | Splicing Event | | PAS | | Coding Potential | | mRNA SL | Ribosome Loading | ncRNA Category | miRNA SL | gRNA Efficiency |
| | Donor_human ACC↑ | Donor_ara ACC↑ | Acceptor_human ACC↑ | Acceptor_ara ACC↑ | human F1↑ | ara F1↑ | human ACC↑ | mouse_bl ACC↑ | human ACC↑ | mouse ACC↑ | F1↑ | R²↑ | ACC↑ | subACC↑ | Average Spearman ρ↑ |
| One-hot | $0.903_{(0.001)}$ | $0.893_{(0.012)}$ | $0.900_{(0.001)}$ | $0.746_{(0.013)}$ | $0.585_{(0.004)}$ | $0.310_{(0.027)}$ | $0.752_{(0.003)}$ | $0.633_{(0.010)}$ | $0.945_{(0.001)}$ | $0.924_{(0.002)}$ | $0.583_{(0.010)}$ | $0.648_{(0.012)}$ | $0.947_{(0.005)}$ | $0.299_{(0.017)}$ | 0.252 |
| Dense | $0.895_{(0.002)}$ | $0.762_{(0.019)}$ | $0.895_{(0.001)}$ | $0.738_{(0.010)}$ | $0.589_{(0.016)}$ | $0.330_{(0.033)}$ | $0.744_{(0.002)}$ | $0.629_{(0.012)}$ | $0.946_{(0.001)}$ | $0.922_{(0.002)}$ | $0.588_{(0.017)}$ | $0.641_{(0.027)}$ | $0.950_{(0.003)}$ | $0.322_{(0.002)}$ | 0.254 |
| **(+MLP)** | | | | | | | | | | | | | | | |
| DNABERT2_mlp | $0.808_{(0.001)}$ | $0.662_{(0.013)}$ | $0.799_{(0.001)}$ | $0.682_{(0.022)}$ | $0.582_{(0.006)}$ | $0.288_{(0.014)}$ | $0.744_{(0.002)}$ | $0.617_{(0.002)}$ | $0.942_{(0.001)}$ | $0.940_{(0.001)}$ | $0.516_{(0.014)}$ | $0.284_{(0.019)}$ | $0.839_{(0.005)}$ | $0.325_{(0.022)}$ | 0.130 |
| HyenaDNA_mlp | $0.780_{(0.001)}$ | $0.665_{(0.004)}$ | $0.797_{(0.002)}$ | $0.692_{(0.007)}$ | $0.533_{(0.003)}$ | $0.291_{(0.016)}$ | $0.738_{(0.003)}$ | $0.641_{(0.007)}$ | $0.896_{(0.004)}$ | $0.872_{(0.011)}$ | $0.494_{(0.010)}$ | $0.154_{(0.019)}$ | $0.676_{(0.017)}$ | $0.322_{(0.000)}$ | 0.178 |
| NT_mlp | $0.776_{(0.001)}$ | $0.667_{(0.007)}$ | $0.775_{(0.004)}$ | $0.655_{(0.001)}$ | $0.535_{(0.003)}$ | $0.296_{(0.020)}$ | $0.738_{(0.001)}$ | $0.628_{(0.006)}$ | $0.895_{(0.003)}$ | $0.877_{(0.003)}$ | $0.476_{(0.003)}$ | $0.113_{(0.008)}$ | $0.727_{(0.009)}$ | $0.322_{(0.000)}$ | 0.171 |
| RNABERT_mlp | $0.675_{(0.003)}$ | $0.600_{(0.003)}$ | $0.664_{(0.001)}$ | $0.612_{(0.015)}$ | $0.410_{(0.011)}$ | $0.215_{(0.014)}$ | $0.702_{(0.002)}$ | $0.624_{(0.002)}$ | $0.726_{(0.003)}$ | $0.744_{(0.001)}$ | $0.425_{(0.000)}$ | $-0.036_{(0.010)}$ | $0.523_{(0.007)}$ | $0.322_{(0.000)}$ | 0.129 |
| RNA-FM_mlp | $0.803_{(0.000)}$ | $0.694_{(0.001)}$ | $0.807_{(0.003)}$ | $0.708_{(0.013)}$ | $0.592_{(0.004)}$ | $0.344_{(0.063)}$ | $0.758_{(0.001)}$ | $0.643_{(0.000)}$ | $0.941_{(0.000)}$ | $0.942_{(0.002)}$ | $0.520_{(0.001)}$ | $0.169_{(0.009)}$ | $0.965_{(0.001)}$ | $0.333_{(0.000)}$ | 0.200 |
| 3UTRBERT_mlp | $0.814_{(0.006)}$ | $0.707_{(0.002)}$ | $0.814_{(0.000)}$ | $0.730_{(0.002)}$ | $0.599_{(0.003)}$ | $0.348_{(0.011)}$ | $0.765_{(0.002)}$ | $0.654_{(0.000)}$ | $0.905_{(0.002)}$ | $0.918_{(0.003)}$ | $0.516_{(0.010)}$ | $0.299_{(0.021)}$ | $0.829_{(0.003)}$ | $0.302_{(0.004)}$ | 0.157 |
| SpliceBERT_mlp | $0.886_{(0.000)}$ | $0.802_{(0.011)}$ | $0.887_{(0.001)}$ | $0.843_{(0.004)}$ | $0.646_{(0.009)}$ | $0.367_{(0.006)}$ | $0.763_{(0.003)}$ | $0.606_{(0.008)}$ | $0.962_{(0.000)}$ | $0.964_{(0.001)}$ | $0.560_{(0.004)}$ | $0.158_{(0.005)}$ | $0.849_{(0.008)}$ | $0.325_{(0.004)}$ | 0.134 |
| UTR-LM_mlp | $0.768_{(0.000)}$ | $0.632_{(0.007)}$ | $0.766_{(0.001)}$ | $0.625_{(0.009)}$ | $0.510_{(0.011)}$ | $0.263_{(0.030)}$ | $0.738_{(0.001)}$ | $0.642_{(0.003)}$ | $0.899_{(0.001)}$ | $0.891_{(0.002)}$ | $0.476_{(0.005)}$ | $-0.000_{(0.019)}$ | $0.576_{(0.009)}$ | $0.322_{(0.000)}$ | 0.167 |
| RiNALMo_mlp | $0.868_{(0.002)}$ | $0.780_{(0.001)}$ | $0.858_{(0.001)}$ | $0.779_{(0.002)}$ | $0.640_{(0.002)}$ | $0.353_{(0.006)}$ | $0.640_{(0.002)}$ | $0.635_{(0.000)}$ | $0.969_{(0.001)}$ | $0.976_{(0.000)}$ | $0.576_{(0.014)}$ | $0.230_{(0.003)}$ | $0.957_{(0.015)}$ | $0.328_{(0.005)}$ | 0.184 |
| **(+CNN)** | | | | | | | | | | | | | | | |
| DNABERT2_cnn | $0.861_{(0.004)}$ | $0.726_{(0.009)}$ | $0.851_{(0.001)}$ | $0.725_{(0.017)}$ | $0.619_{(0.014)}$ | $0.306_{(0.025)}$ | $0.753_{(0.007)}$ | $0.610_{(0.009)}$ | $0.955_{(0.001)}$ | $0.939_{(0.001)}$ | $0.530_{(0.008)}$ | $0.335_{(0.014)}$ | $0.900_{(0.007)}$ | $0.356_{(0.012)}$ | 0.137 |
| HyenaDNA_cnn | $0.954_{(0.001)}$ | $0.893_{(0.004)}$ | $0.946_{(0.001)}$ | $0.854_{(0.009)}$ | $0.663_{(0.010)}$ | $0.417_{(0.020)}$ | $0.760_{(0.010)}$ | $0.678_{(0.015)}$ | $0.927_{(0.003)}$ | $0.905_{(0.004)}$ | $0.492_{(0.005)}$ | $0.453_{(0.008)}$ | $0.804_{(0.018)}$ | $0.322_{(0.00)}$ | 0.228 |
| NT_cnn | $0.925_{(0.005)}$ | $0.837_{(0.009)}$ | $0.902_{(0.001)}$ | $0.796_{(0.009)}$ | $0.570_{(0.011)}$ | $0.346_{(0.011)}$ | $0.741_{(0.002)}$ | $0.632_{(0.007)}$ | $0.911_{(0.002)}$ | $0.880_{(0.001)}$ | $0.529_{(0.015)}$ | $0.285_{(0.009)}$ | $0.898_{(0.002)}$ | $0.322_{(0.000)}$ | 0.249 |
| RNABERT_cnn | $0.931_{(0.001)}$ | $0.844_{(0.006)}$ | $0.904_{(0.003)}$ | $0.799_{(0.009)}$ | $0.511_{(0.011)}$ | $0.295_{(0.012)}$ | $0.727_{(0.002)}$ | $0.640_{(0.003)}$ | $0.784_{(0.005)}$ | $0.780_{(0.003)}$ | $0.425_{(0.000)}$ | $0.407_{(0.022)}$ | $0.715_{(0.009)}$ | $0.322_{(0.000)}$ | 0.195 |
| RNA-FM_cnn | $0.958_{(0.001)}$ | $0.891_{(0.002)}$ | $0.940_{(0.001)}$ | $0.845_{(0.002)}$ | $0.717_{(0.014)}$ | $0.448_{(0.017)}$ | $0.773_{(0.001)}$ | $0.672_{(0.005)}$ | $0.952_{(0.000)}$ | $0.948_{(0.003)}$ | $0.544_{(0.000)}$ | $0.488_{(0.008)}$ | $0.967_{(0.003)}$ | $0.342_{(0.020)}$ | 0.256 |
| 3UTRBERT_cnn | $0.957_{(0.001)}$ | $0.889_{(0.002)}$ | $0.947_{(0.002)}$ | $0.841_{(0.002)}$ | $0.713_{(0.004)}$ | $0.440_{(0.017)}$ | $0.779_{(0.003)}$ | $0.639_{(0.011)}$ | $0.925_{(0.001)}$ | $0.921_{(0.003)}$ | $0.569_{(0.009)}$ | $0.342_{(0.015)}$ | $0.918_{(0.003)}$ | $0.311_{(0.011)}$ | 0.175 |
| SpliceBERT_cnn | $0.957_{(0.000)}$ | $0.906_{(0.007)}$ | $0.941_{(0.001)}$ | $0.890_{(0.009)}$ | $0.760_{(0.004)}$ | $0.495_{(0.014)}$ | $0.778_{(0.005)}$ | $0.630_{(0.014)}$ | $0.967_{(0.001)}$ | $0.964_{(0.001)}$ | $0.567_{(0.021)}$ | $0.461_{(0.022)}$ | $0.915_{(0.006)}$ | $0.322_{(0.000)}$ | 0.187 |
| UTR-LM_cnn | $0.925_{(0.001)}$ | $0.797_{(0.002)}$ | $0.921_{(0.001)}$ | $0.801_{(0.003)}$ | $0.657_{(0.010)}$ | $0.414_{(0.013)}$ | $0.766_{(0.006)}$ | $0.659_{(0.012)}$ | $0.953_{(0.002)}$ | $0.946_{(0.001)}$ | $0.509_{(0.016)}$ | $0.401_{(0.019)}$ | $0.909_{(0.004)}$ | $0.322_{(0.000)}$ | 0.209 |
| RiNALMo_cnn | $0.903_{(0.002)}$ | $0.855_{(0.002)}$ | $0.895_{(0.002)}$ | $0.859_{(0.001)}$ | $0.691_{(0.002)}$ | $0.383_{(0.005)}$ | $0.770_{(0.004)}$ | $0.634_{(0.008)}$ | $0.928_{(0.003)}$ | $0.913_{(0.000)}$ | $0.594_{(0.010)}$ | $0.501_{(0.004)}$ | $0.974_{(0.002)}$ | $0.325_{(0.010)}$ | 0.253 |
| **(+ResNet)** | | | | | | | | | | | | | | | |
| DNABERT2_resnet | $0.937_{(0.000)}$ | $0.866_{(0.008)}$ | $0.927_{(0.003)}$ | $0.853_{(0.004)}$ | $0.697_{(0.003)}$ | $0.366_{(0.011)}$ | $0.740_{(0.007)}$ | $0.616_{(0.007)}$ | $0.941_{(0.003)}$ | $0.922_{(0.005)}$ | $0.597_{(0.010)}$ | $0.427_{(0.010)}$ | $0.883_{(0.011)}$ | $0.299_{(0.062)}$ | 0.103 |
| HyenaDNA_resnet | $0.964_{(0.000)}$ | $0.903_{(0.009)}$ | $0.954_{(0.003)}$ | $0.848_{(0.013)}$ | $0.621_{(0.002)}$ | $0.389_{(0.041)}$ | $0.781_{(0.003)}$ | $0.993_{(0.007)}$ | $0.935_{(0.000)}$ | $0.908_{(0.003)}$ | $0.593_{(0.012)}$ | $0.637_{(0.002)}$ | $0.941_{(0.008)}$ | $0.302_{(0.017)}$ | 0.216 |
| NT_resnet | $0.954_{(0.001)}$ | $0.893_{(0.012)}$ | $0.937_{(0.002)}$ | $0.859_{(0.002)}$ | $0.797_{(0.005)}$ | $0.526_{(0.012)}$ | $0.741_{(0.015)}$ | $0.648_{(0.000)}$ | $0.919_{(0.001)}$ | $0.888_{(0.002)}$ | $0.598_{(0.007)}$ | $0.419_{(0.015)}$ | $0.905_{(0.010)}$ | $0.291_{(0.028)}$ | 0.189 |
| RNABERT_resnet | $0.965_{(0.001)}$ | $0.908_{(0.005)}$ | $0.952_{(0.001)}$ | $0.856_{(0.011)}$ | $0.725_{(0.001)}$ | $0.493_{(0.026)}$ | $0.738_{(0.005)}$ | $0.636_{(0.007)}$ | $0.854_{(0.003)}$ | $0.832_{(0.005)}$ | $0.523_{(0.018)}$ | $0.630_{(0.011)}$ | $0.947_{(0.004)}$ | $0.311_{(0.016)}$ | 0.164 |
| RNA-FM_resnet | $0.967_{(0.000)}$ | $0.913_{(0.004)}$ | $0.955_{(0.001)}$ | $0.859_{(0.003)}$ | $0.768_{(0.005)}$ | $0.511_{(0.026)}$ | $0.776_{(0.002)}$ | $0.671_{(0.013)}$ | $0.935_{(0.001)}$ | $0.948_{(0.000)}$ | $0.591_{(0.000)}$ | $0.644_{(0.023)}$ | $0.975_{(0.002)}$ | $0.362_{(0.035)}$ | 0.225 |
| 3UTRBERT_resnet | $0.967_{(0.000)}$ | $0.912_{(0.000)}$ | $0.954_{(0.000)}$ | $0.869_{(0.004)}$ | $0.727_{(0.006)}$ | $0.476_{(0.013)}$ | $0.775_{(0.003)}$ | $0.656_{(0.016)}$ | $0.916_{(0.002)}$ | $0.916_{(0.001)}$ | $0.587_{(0.004)}$ | $0.608_{(0.001)}$ | $0.941_{(0.002)}$ | $0.285_{(0.026)}$ | 0.166 |
| SpliceBERT_resnet | $0.965_{(0.001)}$ | $0.933_{(0.002)}$ | $0.950_{(0.001)}$ | $0.898_{(0.002)}$ | $0.792_{(0.004)}$ | $0.514_{(0.011)}$ | $0.775_{(0.002)}$ | $0.628_{(0.011)}$ | $0.925_{(0.002)}$ | $0.946_{(0.001)}$ | $0.607_{(0.023)}$ | $0.635_{(0.006)}$ | $0.945_{(0.006)}$ | $0.308_{(0.034)}$ | 0.184 |
| UTR-LM_resnet | $0.945_{(0.001)}$ | $0.861_{(0.006)}$ | $0.941_{(0.001)}$ | $0.826_{(0.007)}$ | $0.678_{(0.003)}$ | $0.436_{(0.004)}$ | $0.766_{(0.004)}$ | $0.654_{(0.011)}$ | $0.954_{(0.001)}$ | $0.948_{(0.000)}$ | $0.608_{(0.004)}$ | $0.647_{(0.011)}$ | $0.958_{(0.006)}$ | $0.331_{(0.009)}$ | 0.246 |
| RiNALMo_resnet | $0.931_{(0.002)}$ | $0.871_{(0.004)}$ | $0.928_{(0.002)}$ | $0.864_{(0.007)}$ | $0.711_{(0.003)}$ | $0.453_{(0.015)}$ | $0.769_{(0.002)}$ | $0.661_{(0.010)}$ | $0.972_{(0.001)}$ | $0.977_{(0.001)}$ | $0.604_{(0.004)}$ | $0.643_{(0.014)}$ | $0.969_{(0.002)}$ | $0.302_{(0.005)}$ | 0.237 |

In function prediction tasks, RNA pLMs exhibit strong alignment between pre-training domain and downstream performance. Under probing with a simple MLP, SpliceBERT achieves the highest accuracy across all pre-mRNA splicing benchmarks, while 3UTRBERT excels in polyadenylation site (PAS) prediction involving 3' UTRs. RNA-FM and RiNALMo, pretrained on a broad corpus of non-coding RNAs, achieve top performance in ncRNA classification and microRNA subcellular localization (see Fig. 6).

In contrast, RNA pLMs underperform relative to the baseline on ribosome loading prediction, a regression task designed to engineer translation efficiency through modeling 5' UTR-driven ribosome occupancy. Similarly, in the gRNA efficiency prediction task, another synthetic engineering scenario, pLMs perform comparably to the baseline. These results suggest limited generalizability to design-driven tasks beyond the evolutionary distributions represented in the training corpus.

## 5.4 Limitation and Future Directions

Evaluating models' ability to capture both conserved and diverse RNA features is essential. In RNA biology, increasing complexity—from structure to interactions to function—introduces stronger evolutionary constraints, shaping sequence conservation. RNA structure (e.g. synthetic RNA) alone is subject to minimal constraint, whereas RNA interaction requires the recognition of biophysically compatible features, such as electrostatic forces and suitable geometric shapes. Further cellular RNA function imposes even higher constraints, demanding specific target interactions while avoiding nonspecific binding with others. Accordingly, RNA pLMs perform well in identifying predominant RNA patterns using binary labels but show reduced effectiveness in handling more complex, multi-class characteristics. For instance, while they can recognize on/off binding signals between RNA and RBP, their performance declines in tasks involving varying levels of binding strength and specificity. This limitation may reflect the inherent imbalance in pre-training source, where certain RNA patterns are pervasive and cryptic features are likely underrepresented, ultimately hindering pLM generalization across diverse RNA subtasks.

To address this complexity, expanding RNAscope to incorporate more diverse and underrepresented RNA patterns from both *in vitro* and *in vivo* source will help to evaluate the generalizability of RNA pLMs. However, it is important to note that RNA pLM may still limit to fully reflect RNA behavior in spatiotemporal cellular contexts, such as transcriptome-wide saturation in guide RNA off-target. From model perspectives, further efforts involving more advanced approaches, such as deeper architectures, structure-aware embeddings, and multi-task learning, may help to tackle these challenges, enabling RNA pLMs to characterize RNA features beyond conserved sequence patterns and thereby support RNA engineering and therapeutics.

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

# Contents

# A  Glossary

- **Pre-mRNA (Precursor mRNA)**: Pre-mRNA is the precursor RNA transcript synthesized from genomic DNA, containing both exons (coding regions) and introns (non-coding regions). It undergoes various maturation processes, including splicing, 5' capping, and 3' polyadenylation, to be processed into mature mRNA.

- **Splicing**: The process of removing introns and joining exons to form a continuous coding sequence within pre-mRNA, facilitating the efficient translation of genetic information into protein.

- **Alternative Splicing**: A process that allows a single pre-mRNA to be spliced in various ways, producing different mRNA isoforms and increasing protein diversity.

- **3' Polyadenylation**: The addition of a poly(A) tail to the 3' end of pre-mRNA, enhancing its stability, facilitating nuclear export, and promoting translation initiation.

- **mRNA (Messenger RNA)**: Mature mRNA is the processed transcript that carries genetic information from DNA in the nucleus to ribosomes for protein synthesis. It consists of three parts: the 5' untranslated region (UTR), coding sequence (CDS), and 3' UTR.

- **5' UTR (5' Untranslated Region)**: The 5' UTR, located upstream of the coding sequence in mRNA, regulates translation initiation through interactions with translation initiation factors and ribosomal machinery, without encoding protein.

- **3' UTR (3' Untranslated Region)**: The 3' UTR is the noncoding region downstream of the coding sequence in mRNA. It regulates mRNA stability, translation, and localization through interactions with RNA-binding proteins and microRNAs.

- **ncRNA (Non-Coding RNA)**: Non-coding RNAs do not translate into proteins but are vital for regulating gene expression, RNA processing, and cellular functions.

- **miRNA (MicroRNA)**: Small, essential non-coding RNAs that regulate gene expression post-transcriptionally by promoting the degradation or inhibiting the translation of target mRNAs.

- **gRNA (Guide RNA)**: Engineered non-coding RNAs that directs the Cas9 protein to specific genomic region for gene editing, as in the CRISPR-Cas9 system.

# B  Towards Robust Evaluation of RNA Language Models

RNA pLMs have shown strong performance on established benchmarks, many of which focus on tasks dominated by evolutionarily conserved sequence features. While such conservation provides helpful inductive biases, it also narrows the scope of evaluation: models may overfit to recurring patterns without learning representations that generalize to biologically diverse or synthetic contexts. In practice, many challenges in RNA biology occur in settings with weak even absent conservation signals—such as transient or low-affinity RNA–protein interactions, cross-species annotation, or the design of functional de novo sequences. Consequently, it remains unclear whether RNA pLMs capture generalizable principles or primarily reflect dataset-specific regularities.

To address this limitation, RNAscope presents a hierarchical evaluation framework that spans a continuum of evolutionary constraints (Fig. 2). It incorporates: (i) dataset splits promoting diversity across sequence families, species, and experimental contexts; (ii) task types beyond binary classification, including regression, ranking, and multi-label prediction; and (iii) functionally diverse tasks, ranging from

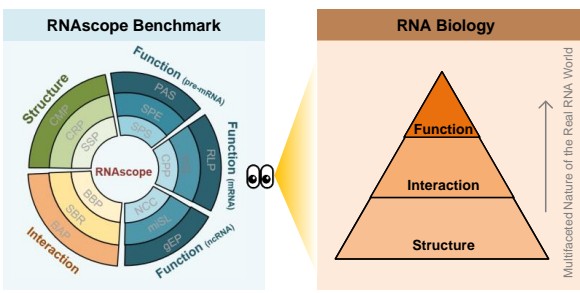

Figure 2: Assessment of RNA pLMs performance in structure, interaction, and function, mirroring the complexity of RNA biology.

conserved biological processes (e.g., canonical splicing) to design-driven challenges with limited

evolutionary precedent (e.g., synthetic gRNA optimization). This design enables a more principled analysis of model generalization across varying biological contexts.

## B.1 Challenges of Existing RNA Benchmarking Frameworks

A representative prior benchmark is BEACON [50], which evaluates RNA pLMs across 13 tasks spanning structure, function, and engineering. However, like many existing evaluation frameworks, **BEACON falls short in capturing the large complexity of RNA biology, leading to limited and potentially biased assessments that constrain generalization and real-world applicability.** In particular, it overlooks critical challenges such as structural redundancy, functional heterogeneity, and limited model robustness on synthetic or under-constrained sequence distributions. In particular, **RNA interaction tasks, crucial for linking structure to regulatory function, are absent from BEACON.** This gap restricts assessment of a model's capacity to capture context-sensitive binding behavior. Specifically, critical evaluation aspects—particularly within structure, function, and engineering tasks—remain underexplored, including:

### (1) Structural Tasks: Lack of Cross-Family and Cross-Structure Validation

RNA structure prediction remains challenging due to the limited availability of experimentally resolved structures and the low diversity across RNA families [56]. Public datasets are often highly imbalanced—for example, over 95% of sequences in some benchmarks originate from rRNA and tRNA [23], which exhibit highly conserved structures even at sequence identities below 30%. Despite this, many models are trained and evaluated on overlapping structural families, leading to inflated performance estimates [60, 31, 48]. BEACON adopts sequence-similarity-based splits (e.g., 80% identity), but fails to account for structural similarity, thereby limiting assessment of generalization to novel RNA folds.

### (2) Functional Tasks: Inadequate Cross-Type and Cross-Species Assessment

RNA molecules span diverse types—pre-mRNA, mRNA, and ncRNA—each governed by distinct regulatory mechanisms and functional contexts. Effectively modeling these differences requires task definitions and evaluation criteria tailored to the unique biology of each RNA type. However, current benchmarks often aggregate these tasks under shared metrics and architectures without distinction, potentially masking type-specific modeling challenges.

Moreover, functional benchmarks are predominantly derived from higher eukaryotes (e.g., human and mouse), with minimal support for cross-species evaluation. This limits assessment of a model's ability to generalize regulatory patterns conserved across evolutionary lineages. While BEACON includes tasks covering multiple RNA types (e.g., splicing, APA, ncRNA classification), it does not differentiate them in task formulation or analysis, reducing the granularity of functional assessment.

### (3) Engineering Tasks: Dataset Bias and Experimental Context Dependency

In design-oriented tasks—such as predicting CRISPR on-target and off-target efficiency—models often exhibit strong performance under in-dataset cross-validation, yet fail to generalize to independent datasets derived from distinct biological systems or experimental conditions [34]. Such performance drops highlight the tendency of current models to overfit to dataset-specific features, rather than learning robust, transferable representations. BEACON does not incorporate cross-system or cross-condition evaluation protocols, thereby limiting its ability to assess model reliability in synthetic or translational settings that deviate from the training distribution.

## B.2 The RNAscope Framework

Table 6: Comparison of RNAscope with BEACON benchmark.

| Benchmark | Tasks | Task Variants | Test Sets | Total Samples | Downstream Modules | Task Relationship[*] |
|---|---|---|---|---|---|---|
| RNAscope | 15 | 42 | 63 | 3,693,205 | MLP, CNN, ResNet | Complexity levels |
| BEACON | 13 | 13 | 13 | 967,166 | ResNet | Unintended correlations |

**Note**: *Task Relationship indicates whether the relationships between tasks are deliberately designed, including the hierarchical setup of sub-tasks and the criteria for splitting test sets.

In light of these limitations, we propose **RNAscope**, an expanded benchmark designed to provide hierarchical, evolution-aware evaluations across structure, interaction, and function. As summarized in Table 7, RNAscope integrates dataset diversity, task variety, and context specificity to offer a more rigorous and interpretable assessment of RNA pLMs. Compared to prior work (see Table 6), RNAscope explicitly addresses the key challenges outlined above, enabling more realistic and generalizable model evaluation.

Table 7: Task features in RNAscope across structure, interaction, and function.

| Task Name | Description | Task Variants | Test Pre-task Variant | Total Test Sets | Test Split |
|---|---|---|---|---|---|
| **Structure Tasks** | | | | | |
| Secondary Structure Prediction | *Foundational complexity*; *Assessing base pairing interactions*. | 3 | 2 | 6 | Cross-family evaluation with two test sets: Intra-family Test for within-family prediction and Inter-family Test for cross-family prediction. |
| Chemical Reactivity Prediction | *Moderate complexity*; *Assessing base pairing and dynamic conformational changes*. | 1 | 2 | 2 | Cross-length evaluation with two test sets: TestS for short sequences, and TestL for long sequences. |
| Contact Map Prediction | *High complexity*; *Assessing 3D structural interactions*. | 1 | 1 | 1 | Cross-structure evaluation with redundancy removal based on structural and sequence similarity. |
| **Interaction Tasks** | | | | | |
| Binary Binding Prediction | *Foundational complexity*; *In vivo data*; *Assessing binary classification for in vivo RNA binding to RBPs*. | 22 | 1 | 22 | 22 independent training and test sets for 22 distinct targets. |
| Systematic Binding Ranking | *Moderate complexity*; *In vitro data*; *Assessing multi-level classification for in vitro RNA binding across three target types*. | 3 | 1 | 3 | Three independent training and test sets for three different target types. |
| Binding Affinity Prediction | *High complexity*; *In vitro data*; *Assessing prediction of RNA binding affinities for two protein targets*. | 2 | 1 | 2 | Two independent training and test sets for two different protein targets. |
| **Function Tasks** | | | | | |
| Splicing Site Prediction | *Pre-mRNA task*; *Biological function task*; *Highly conserved biological process*. | 2 | 3 | 6 | Cross-species validation including human, Arabidopsis (Ara), and rice. |
| Splicing Event Prediction | *Pre-mRNA task*; *Biological function task*; *Moderately conserved biological process*. | 1 | 3 | 3 | Cross-species validation with three test sets: human, Arabidopsis (Ara), and rice. |
| Polyadenylation Signal Prediction | *Pre-mRNA task*; *Biological function task*; *Moderately conserved biological process*. | 1 | 3 | 3 | Cross-species validation including human, and C57BL/6J and SPRET/EiJ mouse strains. |
| Coding Potential Prediction | *mRNA task*; *Biological function task*; *Highly conserved biological process*. | 1 | 5 | 5 | Cross-species validation including five species. |
| Subcellular Localization Prediction | *mRNA task*; *Biological function task*; *Lowly conserved biological process*. | 1 | 1 | 1 | Multi-label classification task. |
| Ribosome Loading Prediction | *mRNA task*; *Functional engineering task designed to surpass natural sequence properties*. | 1 | 1 | 1 | Task partitioned based on predefined criteria. |
| ncRNA Category Classification | *ncRNA task*; *Biological function task*; *Highly conserved biological process*. | 1 | 1 | 1 | Task partitioned for classification. |
| microRNA Subcellular Localization | *ncRNA task*; *Biological function task*; *Lowly conserved biological process*. | 1 | 1 | 1 | Task partitioned for microRNA classification. |
| gRNA Efficiency Prediction | *ncRNA task*; *Functional engineering task designed to surpass natural sequence properties*. | 1 | 6 | 6 | Testing conducted on six datasets covering multiple cell lines and organisms (human, mouse, zebrafish). |

# C  Detailed Dataset Construction and Analysis

## C.1  Structure-related Tasks

• **Secondary structure prediction** includes three benchmarks: bpRNA, SetA, and SetB. The bpRNA dataset comprises two subsets that enable both intra- and inter-family evaluation. bpRNA-1m, derived from Rfam 12.2 [32], is used for within-family testing, with TR0, VL0, and TS0 serving as the training, validation, and intra-family test sets, respectively. bpRNA-new [55] includes families newly added in Rfam 14.2, which are absent from the original training distribution, and is used to evaluate generalization to novel RNA families. Redundant sequences were filtered using an 80% identity threshold via CD-HIT [24], and sequences longer than 500 nucleotides were excluded to ensure consistency. SetA and SetB, introduced by Rivas et al. [52], offer additional benchmarks with varying structural similarity. SetA includes TrainSetA and TestSetA, curated from literature and covering eight common RNA families (e.g., tRNA, SRP RNA, tmRNA), used primarily for intra-family evaluation. SetB includes TrainSetB and TestSetB, constructed from 22 structurally distinct RNA families in Rfam 10.0 [26] with 3D annotations. Sequences in SetB share less than 70% identity with those in SetA to ensure structural independence. All pseudoknotted structures were removed from all datasets. TestSetA and TestSetB serve as intra- and inter-family test sets, respectively.

• **Chemical reactivity prediction dataset** is sourced from the publicly available 'Stanford Ribonanza RNA Folding competition' on Kaggle [61], which includes experimental reactivity measurements for each position in RNA molecules. These measurements are highly sensitive to the in vitro structures (or multiple structures) formed by the RNA. Accurate prediction of chemical reactivity necessitates an implicit comprehension of RNA structure. The dataset, extracted from `train_data_QUICK_START.csv`[4] and based on 2A3 chemical modifier measurements, is partitioned into training and testing sets according to specific length distributions: $\{115, 155, 170, 177, 206\}$. The shorter test set (TestS) includes lengths $\{115, 155\}$, while the longer test set (TestL) contains sequences of length $\{206\}$. TestS contains 4,403 samples, TestL contains 2,384 samples, totaling 6,787 samples.

• **Contact map prediction dataset** is based on RNA3DB [61], a structured RNA dataset sourced from the Protein Data Bank (PDB), designed for training and benchmarking deep learning models. RNA3DB encompasses all PDB RNA 3D structures, clustering RNA 3D chains into distinct, non-redundant groups in terms of both sequence and structure. This organization facilitates a reliable approach for partitioning training, validation, and test sets. RNA3DB is periodically updated by its maintainers, and the version utilized in this study is `2024-12-04-full-release`[5]. There is only one test set, containing 1,375 samples.

## C.2  Interaction-related Tasks

• **Binary binding prediction dataset** originates from the eCLIP dataset [40] curated by [76], which includes 22 distinct datasets from 22 RNA-binding proteins (RBPs) across two cell lines, K562 and HepG2. eCLIP (enhanced Crosslinking and Immunoprecipitation) [68] is a high-resolution technique used to map RNA-protein interactions *in vivo*, enabling the precise identification of RBP binding sites across the transcriptome. All sequencing data is standardized to 100 nucleotide lengths, and sequences with more than 80% identity are removed. The dataset maintains a positive-to-negative ratio of 1:2. The number of training, validation, and test samples for the 22 datasets is detailed in Table 8.

• **Systematic binding ranking dataset**, curated by [78], is based on raw data derived from SELEX (Systematic Evolution of Ligands by Exponential Enrichment) experiments, which are designed to identify high-affinity RNA molecules from a large pool of *in vitro* candidates targeting specific molecules. Specifically, the dataset consists of three distinct subsets: DAse, TARDBP, and ISLETS. These subsets correspond to different targets, including small molecules (Maleimide involved in Diels-Alderase), proteins (TAR DNA-binding protein 43), and multi-cellular entities (human islets), respectively. For each original dataset, RNA species are ranked according to their enrichment levels in the final SELEX round and classified into categories based on their ranking range. Subsequently,

---

[4]https://www.kaggle.com/competitions/stanford-ribonanza-rna-folding/data
[5]https://github.com/marcellszi/rna3db/releases/tag/2024-12-04-full-release

each RNA category within the benchmark dataset was partitioned into training, validation, and test sets at a ratio of 6:1:3.

• **Binding affinity prediction dataset**, sourced from [63], utilizes the high-throughput sequencing–RNA affinity profiling (HiTS-RAP) method to measure the binding affinities of *in vitro* mutagenized libraries of GFP-binding and NELF-E-binding aptamers to their respective targets. The dataset includes two subsets: GFP and NELF, which measure the binding affinities of wild-type, single-mutant, and double-mutant variants of GFPapt and NELEapt. To assess the model's applicability to real-world scenarios, we adopt a '1-vs-rest' splitting strategy, following the approach of FLIP [17]. Specifically, the wild-type and single-mutant variants are assigned to the training set, while the remaining mutant variants are used for testing.

Table 8: Dataset splits for the binary binding Prediction task of 22 RNA-binding proteins (RBPs) across HepG2 and K562 cell lines, detailing the number of samples in training, validation, and test sets.

| Dataset name | Train | Validation | Test |
|---|---|---|---|
| AKAP1_HepG2 | 9,635 | 3,212 | 3,213 |
| BCLAF1_HepG2 | 40,890 | 13,630 | 13,631 |
| DDX24_K562 | 10,632 | 3,544 | 3,546 |
| FAM120A_K562 | 7,572 | 2,524 | 2,525 |
| G3BP1_HepG2 | 9,515 | 3,172 | 3,173 |
| GRWD1_HepG2 | 29,326 | 9,776 | 9,776 |
| IGF2BP1_K562 | 8,543 | 2,848 | 2,848 |
| LARP4_HepG2 | 8,386 | 2,796 | 2,796 |
| LIN28B_K562 | 8,743 | 2,916 | 2,914 |
| PABPC4_K562 | 9,293 | 3,098 | 3,098 |
| PPIG_HepG2 | 24,836 | 8,279 | 8,280 |
| PUM2_K562 | 8,472 | 2,824 | 2,825 |
| RBM15_K562 | 12,468 | 4,156 | 4,156 |
| RPS3_HepG2 | 8,807 | 2,935 | 2,937 |
| SND1_HepG2 | 11,069 | 3,690 | 3,691 |
| UCHL5_K562 | 23,411 | 7,804 | 7,805 |
| YBX3_K562 | 26,165 | 8,721 | 8,723 |
| ZNF622_K562 | 19,195 | 6,399 | 6,400 |
| DDX3X_HepG2 | 10,859 | 3,620 | 3,621 |
| DDX3X_K562 | 7,378 | 2,460 | 2,461 |
| UPF1_HepG2 | 15,567 | 5,189 | 5,191 |
| UPF1_K562 | 20,829 | 6,943 | 6,945 |
| **Total** | **331,591** | **110,537** | **110,555** |

## C.3    Function-related Tasks

### C.3.1    Pre-mRNA-related Function Tasks

• **Splicing site prediction dataset**, sourced from [38], includes donor and acceptor splice site data for human (H. sapiens), Arabidopsis (A. thaliana), and rice (O. sativa japonica). Splice sites are conserved regions in pre-mRNA that mark the boundaries between exons and introns. Donor sites (5' sites) are located at exon-intron junctions and are often characterized by the dinucleotide *GU*, while acceptor sites (3' sites) are found at intron-exon junctions, typically marked by *AG* [5]. Positive samples are generated by extracting 200-nucleotide flanking sequences from both sides of splice sites, forming 402-nucleotide input sequences. Negative samples consist of non-splice sites following the *GU-AG* rule, with redundancy removed to balance the number of positive and negative samples. Models are trained on the human dataset and validated across all three species for cross-species generalization, as detailed in Table 9. Test sets are filtered using CD-HIT [24] to ensure less than 80% sequence identity with the training data.

• **Splicing event prediction dataset** is provided by [7], which includes transcript data from human, Arabidopsis, and rice, with alternative splicing events identified and classified into five distinct types

for the transcripts of each species using SUPPA [66]. Alternative splicing is vital for biological processes, yet genome-wide splicing patterns in many non-model organisms remain largely unexplored. Here, training is performed on human data and evaluation is conducted across all three species, as detailed in Table 10. To prevent redundancy, test sets are filtered via CD-HIT [24] to maintain sequence identity below 80% relative to the training set.

• **Polyadenylation signal prediction dataset** focuses on identifying true polyadenylation signals (PAS) in the 10–30 nt upstream region, characterized by a 6-nt PAS motif (e.g., AAUAAA) [47]. The dataset includes the Omni human poly(A) dataset with 18,786 true PAS sequences and an equal number of pseudo-PAS sequences from human Chromosome 21 [41]. It also contains poly(A) datasets from C57BL/6J (BL) and SPRET/EiJ (SP) mouse strains, with 46,224 and 40,230 sequences, respectively, balanced for true and pseudo-PAS [74]. Training is performed on the human dataset, with evaluation across all three species, as detailed in Table 11. CD-HIT [24] is applied to filter the test sets, ensuring no more than 80% sequence identity with sequences in the training data.

Table 9: Dataset splits for the splicing site prediction task, including two sub-tasks: donor splice site prediction and acceptor splice site prediction. The table details the number of samples in training, validation, and test sets. For each sub-task, models are trained on the human dataset and evaluated on test sets from all three species: human, arabidopsis, and rice.

| Sub-Task | Split | Species | Number | Total |
|---|---|---|---|---|
| Donor | Train | Human | 169,798 | 169,798 |
| | Validation | Human | 42,448 | 42,448 |
| | Test | Human
Arabidopsis
Rice | 68,180
119,491
128,611 | 316,282 |
| Acceptor | Train | Human | 164,946 | 164,946 |
| | Validation | Human | 41,236 | 41,236 |
| | Test | Human
Arabidopsis
Rice | 66,114
119,780
129,374 | 315,268 |

Table 10: Dataset splits for the splicing event prediction task, detailing the number of samples in training, validation, and test sets. Models are trained on the human dataset and evaluated on test sets from three species: human, arabidopsis, and rice.

| Split | Number | Species | Total |
|---|---|---|---|
| Train | 52,008 | Human | 52,008 |
| Validation | 17,313 | Human | 17,313 |
| Test | 14,383
11,237
6,432 | Human
Arabidopsis
Rice | 32,051 |

Table 11: Dataset splits for the polyadenylation signal prediction task, detailing the number of samples in training, validation, and test sets. Models are trained on the human dataset and evaluated on test sets from three species: human, mouse_bl, and mouse_sp, where mouse represents the species, and bl and sp represent different strains.

| Split | Number | Species | Total |
|---|---|---|---|
| Train | 22,536 | Human | 22,536 |
| Validation | 7,514 | Human | 7,514 |
| Test | 7,363 | Human | 78,291 |
| | 37,868 | mouse_bl | |
| | 33,060 | mouse_sp | |

## C.3.2 mRNA-related Function Tasks

• **Coding potential prediction** involves binary classification to differentiate between coding RNAs and non-coding RNAs (ncRNAs). Rapid and accurate prediction of coding potential is essential for understanding transcript functionality. The dataset, derived from [65], includes both coding and non-coding RNAs from human, mouse, zebrafish, fruit fly, and *Saccharomyces cerevisiae* (s.cerevisiae). To minimize redundancy, a sequence identity cutoff of $\geq 80\%$ is applied between the training and testing sets. Models are trained on the human dataset and evaluated on test sets from all five species, as detailed in Table 12.

• **mRNA subcellular localization** aims to predict the spatial distribution of mRNA. The asymmetric distribution of mRNA across different subcellular compartments tightly regulates protein synthesis within human cells. Accurate identification of mRNA subcellular localization is crucial for deepening our understanding of gene regulatory networks. The benchmark dataset, sourced from [36], includes data from the RNALocate and DM3Loc databases, comprising a total of 17,298 unique mRNA sequences. Sequence redundancy is removed by applying an 80% sequence similarity threshold. The dataset is randomly split into training, validation, and test sets in an 8:1:1 ratio.

• **Ribosome loading prediction** is a functional engineering task aimed at optimizing and designing 5' UTRs to enhance mRNA translation efficiency through accurate ribosome loading prediction. The dataset, derived from MRL measurements in HepG2 cells [8], ranks 5' UTRs by sequencing read counts, with the top 20,000 sequences assigned to the test set, 10,000 to the validation set, and the remaining sequences to the training set.

Table 12: Dataset splits for the coding potential prediction task, detailing the number of samples in training, validation, and test sets. Models are trained on the human dataset and evaluated on test sets from all five species including human, mouse, zebrafish, fruit fly, and s. cerevisiae.

| Split | Number | Species | Total |
|---|---|---|---|
| Train | 51,770 | Human | 51,770 |
| Validation | 5,753 | Human | 5,753 |
| Test | 16,798 | Human | 122,710 |
| | 51,032 | Mouse | |
| | 26,256 | Zebrafish | |
| | 21,498 | Fruit_fly | |
| | 7,126 | S. cerevisiae | |

## C.3.3 ncRNA-related Function Tasks

• **ncRNA Category Classification** aims to classify short non-coding RNAs (sncRNAs) based on conserved sequence patterns across different ncRNA types. The dataset, sourced from [37],

comprises 31,000 sncRNA sequences from various organisms with 13 ncRNA categories [6]. A 20% non-redundant subset is used to form the testing set.

• **MicroRNA subcellular localization** aims to predict the subcellular localization of microRNAs, a class of non-coding RNAs (ncRNAs) with critical roles in gene regulation. Identifying the subcellular localization of microRNAs is of significant importance for drug design. The dataset, derived from [2], consists of 538 unique microRNAs, with redundancy removed using an 80% identity threshold. The dataset is split into training and testing sets at an 8:2 ratio.

• **gRNA efficiency prediction** is a functional engineering task focused on optimizing gene editing by accurately predicting the on-target efficiency of guide RNAs (gRNAs). The dataset, derived from [34], includes five public gRNA efficiency datasets for training, with testing conducted on six additional datasets from various cell lines and organisms (human, mouse, zebrafish). Independent datasets from diverse biological systems and experimental conditions validate the model's robustness, mitigating overfitting to specific contexts, as detailed in Table 13.

Table 13: Dataset splits for the gRNA Efficiency Prediction task, listing each dataset by split, name, and number of gRNAs.

| Split | Dataset Name | Number |
|---|---|---|
| **Train** | Chari_293t | 984 |
| | Doench_hg | 1,862 |
| | Doench_mel4 | 781 |
| | Moreno-Mateos | 835 |
| | Wang-Xu_hl60 | 1,672 |
| **Validation** | Chari_293t | 250 |
| | Doench_hg | 471 |
| | Doench_mel4 | 170 |
| | Moreno-Mateos | 185 |
| | Wang-Xu_hl60 | 404 |
| **Test** | mESC(Koike-Yusa) | 1,064 |
| | HEL(Labuhn) | 424 |
| | A375(Shalem) | 1,278 |
| | HEK293T(Xi Xiang) | 10,592 |
| | Zebrafish(Gagnon) | 111 |
| | Zebrafish(Shkumatava) | 162 |

# D  Metrics

• **Precision, Recall, and F1-score**: These metrics are used to evaluate classification performance. For binary classification tasks, we adopt the standard **binary F1-score**, which focuses on the positive class (label = 1). Precision measures the proportion of predicted positives that are correct, Recall measures the proportion of actual positives that are recovered, and the F1-score is the harmonic mean of Precision and Recall.

$$\text{Precision} = \frac{TP}{TP + FP}, \quad \text{Recall} = \frac{TP}{TP + FN}, \quad \text{F1-score} = 2 \cdot \frac{\text{Precision} \cdot \text{Recall}}{\text{Precision} + \text{Recall}}. \quad (1)$$

where $TP$, $FP$, and $FN$ denote the number of true positives, false positives, and false negatives, respectively.

For multi-class settings, we report the **macro F1-score**, computed as the arithmetic mean of per-class F1-scores. Unlike micro-averaging, macro F1 gives equal weight to each class, making it more sensitive to performance on underrepresented classes.

---

[6]These categories includes 5.8S rRNA, 5S rRNA, CD-box, HACA-box, Intron-gpI, Intron-gpII, Leader, miRNA, Riboswitch, Ribozyme, tRNA, Y RNA (vertebrates), sbRNA and CeY RNA (nematodes), sbRNA (insects), Y RNA like (bacterial).

- **Accuracy**: Accuracy is defined as the proportion of correctly classified instances (both positive and negative) to the total number of instances.

For a classification task with $N$ instances, let $TP$ be the number of true positives, $TN$ be the number of true negatives, $FP$ be the number of false positives, and $FN$ be the number of false negatives. The Accuracy is given by:

$$\text{ACC} = \frac{TP + TN}{TP + TN + FP + FN}.\tag{2}$$

- **MAE**: In the chemical reactivity regression task, the Mean Absolute Error (MAE) is used to evaluate the performance of models. MAE is defined as:

$$\text{MAE} = \frac{1}{N} \sum_{i=1}^{N} |y_i - \hat{y}_i|,\tag{3}$$

where

- $N$: the total number of samples,
- $y_i$: the true chemical reactivity value of the $i$-th sample,
- $\hat{y}_i$: the predicted chemical reactivity value of the $i$-th sample.

Before calculating the MAE, the actual values $y_i$ are clipped to be within the range [0, 1], as follows:

$$y_i = \max\left(\min\left(y_{\text{RAW}_i}, 1.0\right), 0.0\right),\tag{4}$$

where $y_{\text{RAW}_i}$ are the raw data values. The clipping ensures that the predicted values stay within the valid range of [0, 1] for the scoring process.

- **Spearman** $p$: Spearman's rank correlation coefficient, denoted by $\rho$, measures the monotonic relationship between two variables. Given two sets of observations

$$X = \{x_1, x_2, \ldots, x_n\} \quad \text{and} \quad Y = \{y_1, y_2, \ldots, y_n\},$$

let $R(x_i)$ and $R(y_i)$ be the ranks of $x_i$ and $y_i$ respectively. The Spearman rank correlation coefficient is then defined as

$$\rho = 1 - \frac{6\sum_{i=1}^{n} d_i^2}{n(n^2 - 1)},\tag{5}$$

where

$$d_i = R(x_i) - R(y_i)$$

is the difference between the ranks of the $i$-th pair of observations, and n is the number of observations.

- **R²**: The coefficient of determination, denoted as $R^2$, is defined as the proportion of the variance in the dependent variable that is predictable from the independent variable(s).

For a regression task with $N$ data points, let $y_i$ be the actual value, $\hat{y}_i$ the predicted value, and $\bar{y}$ the mean of the actual values. The formula for $R^2$ is:

$$R^2 = 1 - \frac{\sum_{i=1}^{N} (y_i - \hat{y}_i)^2}{\sum_{i=1}^{N} (y_i - \bar{y})^2},\tag{6}$$

where:

- $N$ is the number of observations,
- $y_i$ is the actual value of the $i$-th observation,
- $\hat{y}_i$ is the predicted value of the $i$-th observation,

842     • $\bar{y}$ is the mean of the actual values.

843 • **Short@L/5, Long@L/5**: For an RNA sequence of length $L$, a mean precision of long-range
844 contacts is used at cutoff of $L = 5$ for benchmarking various predictors, where $i$ and $j$ denote the
845 sequence positions of any two nucleotides in the sequence. The definition of long-range contacts is
846 $|i - j| \geq 24$. This means that long-range contacts are defined as those between nucleotides that are
847 at least 24 positions apart. The definition of short-range contacts is $|i - j| \in [5, 24)$.

848 The definition of Precision is given by:

$$\text{Precision} = \frac{TP}{TP + FP},$$

849 • **Hloss**: Hamming loss (Hloss) reflects the overall accuracy of a model by measuring the number of
850 incorrect labels in classification tasks. A lower value indicates better performance.

851 $Hloss(h)$ is defined as:

$$Hloss(h) = \frac{1}{p} \sum_{i=1}^{p} \frac{1}{q} |h(x_i) \Delta Y_i|$$

852 Where:

853     • $p$ is the total number of data points.

854     • $q$ is the total number of possible labels.

855     • $h(x_i)$ is the predicted label set and $x_i$ is the RNA sequence.

856     • $\Delta Y_i$ is the difference between the predicted and actual microRNA subcellular localization
857     label for $x_i$.

858 • **subACC**: Subset Accuracy (Subset Acc) reflects a model's ability to make precise predictions, as it
859 measures the proportion of samples where all predicted labels exactly match the true labels. A higher
860 value indicates better performance.

861 The $SubsetAcc(h)$ is defined as:

$$SubsetAcc(h) = \frac{1}{p} \sum_{i=1}^{M} \left[ h(x^{(i)}) = y^{(i)} \right]$$

862 Where:

863     • $p$: The total number of data points in the dataset.

864     • $M$: The number of data points that are being evaluated in this sum.

865     • $x^{(i)}$: The feature vector (input data) for the $i$-th instance.

866     • $y^{(i)}$: The true label for the $i$-th instance.

867     • $h(x^{(i)})$: The predicted label for the $i$-th instance.

868     • $\left[ h(x^{(i)}) = y^{(i)} \right]$: An indicator function that equals 1 if the predicted label matches the true
869     label, and 0 otherwise.

870 # E   RNA pLMs and DNA pLMs

871 This study aims to encompass a diverse range of publicly available RNA and DNA pLMs. Here, we
872 provide a detailed description of the RNA and DNA pre-trained models selected for this work. For
873 models that offer multiple pre-trained checkpoints, we focus on a few representative ones, primarily
874 based on the results from the original studies, to optimize computational resource usage.

## E.1 RNA pLMs and checkpoint selection

• **RNABERT**: RNABERT [1] adopts the BERT architecture with six layers. The pretraining objective incorporates both the masked language modelling (MLM) and structural alignment learning, which is a structural alignment loss to capture conserved secondary structure information. The MLM pretraining data consists of 76,237 human-derived small ncRNA sequences, ranging in length from 20 to 440 bases, sourced from RNAcentral [53].

• **RNA-FM**: RNA-FM [10] is built upon the BERT architecture with 12 layers. It employs a single-base tokenization scheme, utilizing unique tokens such as `<cls>` at the start and `<eos>` at the end. Pretraining was conducted using the MLM objective on a dataset of 23 million samples from RNAcentral.

• **3UTRBERT**: 3UTRBERT [76] is built on the BERT architecture with 12 layers. Instead of treating individual bases as separate tokens, it groups consecutive bases into k-mers, with `[CLS]` at the start and `[SEP]` at the end. The model was pre-trained using the MLM objective on 20,362 3'UTR sequences from human transcripts. Multiple checkpoints were pre-trained with k-mer sizes ranging from 3 to 6, with the original study reporting the 3-mer checkpoint achieved the best performance. Hence, we adopt the 3-mer checkpoint for benchmarking in this work.

• **SpliceBERT**: SpliceBERT [11] is based on the BERT architecture, consisting of six transformer layers. It uses a single-base tokenization approach, with `[CLS]` at the start and `[SEP]` at the end. Pretraining was performed using the MLM objective, with 2 million pre-mRNA sequences from 72 vertebrate species.

• **UTR-LM**: UTR-LM [14] is a BERT-based model with six transformer layers and single-base tokenization. It is pretrained on a hybrid corpus of endogenous and randomly synthesized 5' UTR sequences drawn from multiple species, using the MLM objective. To enhance semantic representations, UTR-LM incorporates auxiliary supervision derived from sequence-computable signals, including predicted secondary structures and minimum free energy (MFE).

• **RiNALMo**: RiNALMo [46] adopts a BERT architecture with 33 transformer layers and single-base tokenization. It is pretrained using the MLM objective on 36 million non-coding RNA sequences aggregated from multiple databases, totaling 650 million parameters. RiNALMo emphasizes its generalization capability, particularly in overcoming the limitations of prior deep learning methods that fail to transfer to unseen RNA families in secondary structure prediction tasks.

## E.2 DNA pLMs and checkpoint selection

• **DNABERT2**: DNABERT2 [80] adapts the Transformer Encoder architecture with relative positional encoding via Attention with Linear Biases (ALiBi), consisting of 12 Transformer layers. It employs the Byte Pair Encoding (BPE) tokenizer, enabling efficient tokenization of genomic sequences that often span thousands of bases. It was pre-trained using the MLM objective on a multi-species genome dataset, which includes genomes from 135 species and spans 32.49 billion nucleotide bases.

• **HyenaDNA**: HyenaDNA [43] architecture is a simple stack of Hyena operators. It uses a single-base tokenization approach and is pre-trained on the human reference genome using next-token prediction.

• **Nucleotide Transformer (NT)**: NT [16] uses an encoder-only Transformer architecture with learnable absolute positional encodings. It employs a 6-mer tokenization approach and is pre-trained with the MLM objective. It has four variants: `NT-500M-human`, `NT-500M-1000g`, `NT-2500M-1000g`, and `NT-2500M-multi`, with 500M and 2500M indicating model sizes, and `human`, `1000g`, and `multi` referring to different pretraining datasets (human reference genome, 3202 high-coverage human genomes from the 1000 Genome project [6], and multispecies genomes). We chosen the `NT-500M-1000g` checkpoint for its balanced parameter size and strong performance.

# F Comprehensive Results and Evaluation of RNA Structure, Interaction, and Function Prediction

## F.1 Structure-related Tasks

To assess generalization during training, Fig. F.1 shows the F1 score trajectories of RiNALMo—the top-performing RNA pLM for secondary structure prediction—on three independent datasets: bpRNA, SetA, and SetB. In each case, frozen embeddings from RiNALMo are combined with MLP, shallow CNN, or ResNet classifiers. Overall, deeper classifiers such as ResNet yield higher F1 scores on both validation and intra-family test sets, with performance steadily improving and eventually plateauing. In contrast, inter-family test performance often degrades over time, particularly with ResNet, underscoring the challenge of learning structural representations that generalize across RNA families.

For secondary structure prediction, we further include two widely used thermodynamic models, RNAfold [28] and RNAstructure [51], which estimate the minimum free energy structure by summing contributions from nearest-neighbor loops. Their performance is evaluated across four independent RNA secondary structure test sets (see Table 14). Thermodynamics-based models demonstrate consistently strong performance across RNA secondary structure prediction tasks. On the bpRNA inter-family test set (bpRNA-new), RNAfold achieves an F1 score of 0.746, substantially outperforming the best RNA pLM (RiNALMo_MLP, 0.625). This performance gap is further evident in the SetA/SetB benchmarks. When trained within SetA, RNA pLMs reach up to 0.882 on TestSetA (intra-family). However, when trained on SetB and tested on TestSetA (inter-family), performance drops to 0.726—falling below both RNAfold (0.785) and RNAstructure (0.735). These results highlight the strong cross-family generalization of thermodynamic models and reveal that current pLMs struggle to transfer learned structural features beyond training families.

Table 14: Evaluation of thermodynamics-based secondary structure modles (RNAfold and RNAstructure) on four independent RNA test sets (TS0, bpRNA-new, TestSetA, TestSetB), using Precision, Recall, and F1 as metrics.

| Model | TS0 | | | bpRNA-new | | |
|---|---|---|---|---|---|---|
| | Precision | Recall | F1 | Precision | Recall | F1 |
| RNAfold | 0.619 | 0.825 | 0.707 | 0.644 | 0.886 | 0.746 |
| RNAstructure | 0.572 | 0.782 | 0.661 | 0.573 | 0.817 | 0.673 |

| Model | TestSetA | | | TestSetB | | |
|---|---|---|---|---|---|---|
| | Precision | Recall | F1 | Precision | Recall | F1 |
| RNAfold | 0.735 | 0.843 | 0.785 | 0.627 | 0.824 | 0.712 |
| RNAstructure | 0.683 | 0.795 | 0.735 | 0.541 | 0.745 | 0.627 |

Table 15: Performance of RNA pLMs and baselines on secondary structure prediction, including Precision and Recall for both intra-family and inter-family tests. Results are *mean (std)* from three runs with different seeds, with the best and second-best models highlighted.

| Model | Second Structure | | | | | | | | | | | |
|---|---|---|---|---|---|---|---|---|---|---|---|---|
| | bpRNA$_{Intra-family}$ | | bpRNA$_{Inter-family}$ | | SetA$_{Intra-family}$ | | SetA$_{Inter-family}$ | | SetB$_{Intra-family}$ | | SetB$_{Inter-family}$ | |
| | Precision ↑ | Recall ↑ | Precision ↑ | Recall ↑ | Precision ↑ | Recall ↑ | Precision ↑ | Recall ↑ | Precision ↑ | Recall ↑ | Precision ↑ | Recall ↑ |
| One-hot | 0.465$_{(0.008)}$ | 0.668$_{(0.021)}$ | 0.399$_{(0.002)}$ | 0.608$_{(0.022)}$ | 0.630$_{(0.023)}$ | 0.763$_{(0.050)}$ | 0.278$_{(0.010)}$ | 0.455$_{(0.073)}$ | 0.438$_{(0.022)}$ | 0.320$_{(0.021)}$ | 0.605$_{(0.008)}$ | 0.442$_{(0.035)}$ |
| Dense | 0.433$_{(0.015)}$ | 0.763$_{(0.039)}$ | 0.387$_{(0.006)}$ | 0.712$_{(0.047)}$ | 0.628$_{(0.013)}$ | 0.777$_{(0.029)}$ | 0.277$_{(0.006)}$ | 0.472$_{(0.047)}$ | 0.435$_{(0.027)}$ | 0.346$_{(0.022)}$ | 0.600$_{(0.007)}$ | 0.469$_{(0.044)}$ |
| **(+MLP)** | | | | | | | | | | | | |
| RNABERT_mlp | 0.551$_{(0.003)}$ | 0.551$_{(0.007)}$ | 0.517$_{(0.000)}$ | 0.567$_{(0.007)}$ | 0.614$_{(0.000)}$ | 0.814$_{(0.002)}$ | 0.488$_{(0.000)}$ | 0.803$_{(0.000)}$ | 0.525$_{(0.001)}$ | 0.512$_{(0.004)}$ | 0.649$_{(0.006)}$ | 0.440$_{(0.005)}$ |
| RNA-FM_mlp | 0.747$_{(0.001)}$ | 0.788$_{(0.007)}$ | 0.563$_{(0.003)}$ | 0.661$_{(0.008)}$ | 0.810$_{(0.002)}$ | 0.867$_{(0.004)}$ | 0.657$_{(0.009)}$ | 0.716$_{(0.009)}$ | 0.867$_{(0.002)}$ | 0.871$_{(0.002)}$ | 0.699$_{(0.005)}$ | 0.533$_{(0.011)}$ |
| 3UTRBERT_mlp | 0.600$_{(0.008)}$ | 0.702$_{(0.022)}$ | 0.529$_{(0.006)}$ | 0.686$_{(0.017)}$ | 0.683$_{(0.009)}$ | 0.822$_{(0.006)}$ | 0.500$_{(0.002)}$ | 0.743$_{(0.016)}$ | 0.670$_{(0.005)}$ | 0.676$_{(0.014)}$ | 0.616$_{(0.003)}$ | 0.483$_{(0.014)}$ |
| SpliceBERT_mlp | 0.613$_{(0.018)}$ | 0.696$_{(0.040)}$ | 0.533$_{(0.007)}$ | 0.652$_{(0.034)}$ | 0.708$_{(0.019)}$ | 0.822$_{(0.023)}$ | 0.496$_{(0.005)}$ | 0.718$_{(0.039)}$ | 0.723$_{(0.010)}$ | 0.712$_{(0.018)}$ | 0.615$_{(0.003)}$ | 0.495$_{(0.036)}$ |
| UTR-LM_mlp | 0.599$_{(0.014)}$ | 0.646$_{(0.015)}$ | 0.543$_{(0.004)}$ | 0.634$_{(0.018)}$ | 0.671$_{(0.007)}$ | 0.785$_{(0.010)}$ | 0.505$_{(0.002)}$ | 0.729$_{(0.014)}$ | 0.639$_{(0.014)}$ | 0.626$_{(0.007)}$ | 0.499$_{(0.030)}$ | |
| RiNALMo_mlp | 0.781$_{(0.006)}$ | 0.814$_{(0.009)}$ | 0.575$_{(0.007)}$ | 0.683$_{(0.016)}$ | 0.871$_{(0.004)}$ | 0.892$_{(0.007)}$ | 0.675$_{(0.008)}$ | 0.756$_{(0.010)}$ | 0.887$_{(0.011)}$ | 0.901$_{(0.007)}$ | 0.789$_{(0.004)}$ | 0.673$_{(0.013)}$ |
| **(+CNN)** | | | | | | | | | | | | |
| RNABERT_cnn | 0.587$_{(0.004)}$ | 0.610$_{(0.011)}$ | 0.542$_{(0.002)}$ | 0.603$_{(0.013)}$ | 0.652$_{(0.010)}$ | 0.746$_{(0.007)}$ | 0.507$_{(0.004)}$ | 0.730$_{(0.031)}$ | 0.548$_{(0.005)}$ | 0.606$_{(0.023)}$ | 0.588$_{(0.002)}$ | 0.537$_{(0.028)}$ |
| RNA-FM_cnn | 0.763$_{(0.009)}$ | 0.812$_{(0.007)}$ | 0.541$_{(0.007)}$ | 0.650$_{(0.015)}$ | 0.813$_{(0.006)}$ | 0.874$_{(0.009)}$ | 0.649$_{(0.020)}$ | 0.727$_{(0.016)}$ | 0.879$_{(0.015)}$ | 0.870$_{(0.014)}$ | 0.682$_{(0.012)}$ | 0.489$_{(0.017)}$ |
| 3UTRBERT_cnn | 0.612$_{(0.015)}$ | 0.748$_{(0.019)}$ | 0.509$_{(0.011)}$ | 0.697$_{(0.037)}$ | 0.700$_{(0.001)}$ | 0.837$_{(0.006)}$ | 0.496$_{(0.005)}$ | 0.733$_{(0.013)}$ | 0.704$_{(0.002)}$ | 0.704$_{(0.003)}$ | 0.616$_{(0.003)}$ | 0.401$_{(0.008)}$ |
| SpliceBERT_cnn | 0.634$_{(0.003)}$ | 0.760$_{(0.011)}$ | 0.496$_{(0.005)}$ | 0.656$_{(0.013)}$ | 0.727$_{(0.007)}$ | 0.829$_{(0.016)}$ | 0.492$_{(0.003)}$ | 0.703$_{(0.013)}$ | 0.730$_{(0.004)}$ | 0.744$_{(0.020)}$ | 0.614$_{(0.004)}$ | 0.441$_{(0.015)}$ |
| UTR-LM_cnn | 0.618$_{(0.006)}$ | 0.728$_{(0.012)}$ | 0.524$_{(0.007)}$ | 0.688$_{(0.019)}$ | 0.689$_{(0.005)}$ | 0.807$_{(0.012)}$ | 0.498$_{(0.002)}$ | 0.716$_{(0.012)}$ | 0.709$_{(0.008)}$ | 0.668$_{(0.005)}$ | 0.620$_{(0.003)}$ | 0.400$_{(0.003)}$ |
| RiNALMo_cnn | 0.785$_{(0.006)}$ | 0.823$_{(0.007)}$ | 0.572$_{(0.014)}$ | 0.643$_{(0.025)}$ | 0.874$_{(0.002)}$ | 0.888$_{(0.007)}$ | 0.678$_{(0.003)}$ | 0.738$_{(0.020)}$ | 0.908$_{(0.005)}$ | 0.897$_{(0.002)}$ | 0.779$_{(0.010)}$ | 0.650$_{(0.018)}$ |
| **(+ResNet)** | | | | | | | | | | | | |
| RNABERT_resnet | 0.593$_{(0.026)}$ | 0.818$_{(0.058)}$ | 0.479$_{(0.011)}$ | 0.730$_{(0.077)}$ | 0.690$_{(0.014)}$ | 0.805$_{(0.032)}$ | 0.484$_{(0.004)}$ | 0.723$_{(0.039)}$ | 0.705$_{(0.005)}$ | 0.639$_{(0.009)}$ | 0.610$_{(0.007)}$ | 0.393$_{(0.021)}$ |
| RNA-FM_resnet | 0.734$_{(0.008)}$ | 0.820$_{(0.012)}$ | 0.540$_{(0.009)}$ | 0.693$_{(0.015)}$ | 0.783$_{(0.006)}$ | 0.861$_{(0.006)}$ | 0.585$_{(0.015)}$ | 0.766$_{(0.005)}$ | 0.851$_{(0.017)}$ | 0.870$_{(0.004)}$ | 0.665$_{(0.015)}$ | 0.533$_{(0.045)}$ |
| 3UTRBERT_resnet | 0.597$_{(0.010)}$ | 0.804$_{(0.036)}$ | 0.485$_{(0.004)}$ | 0.713$_{(0.051)}$ | 0.699$_{(0.009)}$ | 0.801$_{(0.018)}$ | 0.489$_{(0.006)}$ | 0.692$_{(0.029)}$ | 0.706$_{(0.022)}$ | 0.686$_{(0.034)}$ | 0.609$_{(0.004)}$ | 0.422$_{(0.049)}$ |
| SpliceBERT_resnet | 0.627$_{(0.012)}$ | 0.781$_{(0.033)}$ | 0.491$_{(0.008)}$ | 0.692$_{(0.041)}$ | 0.718$_{(0.020)}$ | 0.835$_{(0.035)}$ | 0.486$_{(0.006)}$ | 0.719$_{(0.056)}$ | 0.719$_{(0.017)}$ | 0.706$_{(0.015)}$ | 0.599$_{(0.013)}$ | 0.528$_{(0.006)}$ |
| UTR-LM_resnet | 0.625$_{(0.012)}$ | 0.788$_{(0.028)}$ | 0.491$_{(0.006)}$ | 0.676$_{(0.042)}$ | 0.720$_{(0.012)}$ | 0.797$_{(0.033)}$ | 0.491$_{(0.005)}$ | 0.678$_{(0.047)}$ | 0.738$_{(0.028)}$ | 0.698$_{(0.043)}$ | 0.613$_{(0.008)}$ | 0.387$_{(0.058)}$ |
| RiNALMo_resnet | 0.788$_{(0.014)}$ | 0.823$_{(0.010)}$ | 0.573$_{(0.007)}$ | 0.660$_{(0.036)}$ | 0.861$_{(0.005)}$ | 0.903$_{(0.009)}$ | 0.664$_{(0.022)}$ | 0.757$_{(0.044)}$ | 0.902$_{(0.015)}$ | 0.891$_{(0.016)}$ | 0.773$_{(0.022)}$ | 0.596$_{(0.035)}$ |

Table 16: Results of DNA pLMs on structure tasks. Since structure prediction is a nucleotide-level classification task, and DNABERT2 uses a BPE tokenizer while NT employs nonoverlapping k-mers, which are not well-suited for these tasks, only results from HyenaDNA are reported.

| Model | Second Structure | | | | | | | | | | | |
| --- | --- | --- | --- | --- | --- | --- | --- | --- | --- | --- | --- | --- |
| | $bpRNA_{Intra-family}$ | | | $bpRNA_{Inter-family}$ | | | $SetA_{Intra-family}$ | | | $SetA_{Inter-family}$ | | |
| | Precision ↑ | Recall ↑ | F1 ↑ | Precision ↑ | Recall ↑ | F1 ↑ | Precision ↑ | Recall ↑ | F1 ↑ | Precision ↑ | Recall ↑ | F1 ↑ |
| (+MLP) HyenaDNA_mlp | $0.501_{(0.004)}$ | $0.449_{(0.042)}$ | $0.473_{(0.024)}$ | $0.432_{(0.002)}$ | $0.356_{(0.033)}$ | $0.389_{(0.020)}$ | $0.554_{(0.001)}$ | $0.946_{(0.016)}$ | $0.698_{(0.004)}$ | $0.438_{(0.001)}$ | $0.946_{(0.009)}$ | $0.599_{(0.002)}$ |
| (+CNN) HyenaDNA_cnn | $0.515_{(0.002)}$ | $0.469_{(0.015)}$ | $0.491_{(0.008)}$ | $0.434_{(0.001)}$ | $0.383_{(0.027)}$ | $0.406_{(0.015)}$ | $0.555_{(0.001)}$ | $0.967_{(0.005)}$ | $0.706_{(0.002)}$ | $0.439_{(0.000)}$ | $0.961_{(0.008)}$ | $0.603_{(0.002)}$ |
| (+ResNet) HyenaDNA_resnet | $0.492_{(0.016)}$ | $0.764_{(0.091)}$ | $0.596_{(0.019)}$ | $0.422_{(0.015)}$ | $0.305_{(0.130)}$ | $0.342_{(0.087)}$ | $0.563_{(0.003)}$ | $0.934_{(0.010)}$ | $0.702_{(0.001)}$ | $0.443_{(0.003)}$ | $0.919_{(0.027)}$ | $0.598_{(0.003)}$ |

| Model | Second Structure | | | | | | Chemical Reactivity | | Contact Map | |
| --- | --- | --- | --- | --- | --- | --- | --- | --- | --- | --- |
| | $SetB_{Intra-family}$ | | | $SetB_{Inter-family}$ | | | TestS MAE ↓ | TestL MAE ↓ | Short@L/5 ↑ | Long@L/5 ↑ |
| | Precision ↑ | Recall ↑ | F1 ↑ | Precision ↑ | Recall ↑ | F1 ↑ | | | | |
| (+MLP) HyenaDNA_mlp | $0.472_{(0.017)}$ | $0.185_{(0.078)}$ | $0.258_{(0.086)}$ | $0.552_{(0.005)}$ | $0.247_{(0.053)}$ | $0.339_{(0.051)}$ | $0.738_{(0.065)}$ | $0.297_{(0.008)}$ | $0.008_{(0.000)}$ | $0.005_{(0.001)}$ |
| (+CNN) HyenaDNA_cnn | $0.472_{(0.004)}$ | $0.240_{(0.034)}$ | $0.317_{(0.030)}$ | $0.579_{(0.004)}$ | $0.266_{(0.013)}$ | $0.359_{(0.012)}$ | $0.344_{(0.010)}$ | $0.283_{(0.005)}$ | $0.011_{(0.002)}$ | $0.008_{(0.001)}$ |
| (+ResNet) HyenaDNA_resnet | $0.473_{(0.004)}$ | $0.239_{(0.002)}$ | $0.315_{(0.006)}$ | $0.564_{(0.003)}$ | $0.257_{(0.010)}$ | $0.349_{(0.005)}$ | $0.275_{(0.003)}$ | $0.297_{(0.010)}$ | $0.010_{(0.003)}$ | $0.007_{(0.001)}$ |

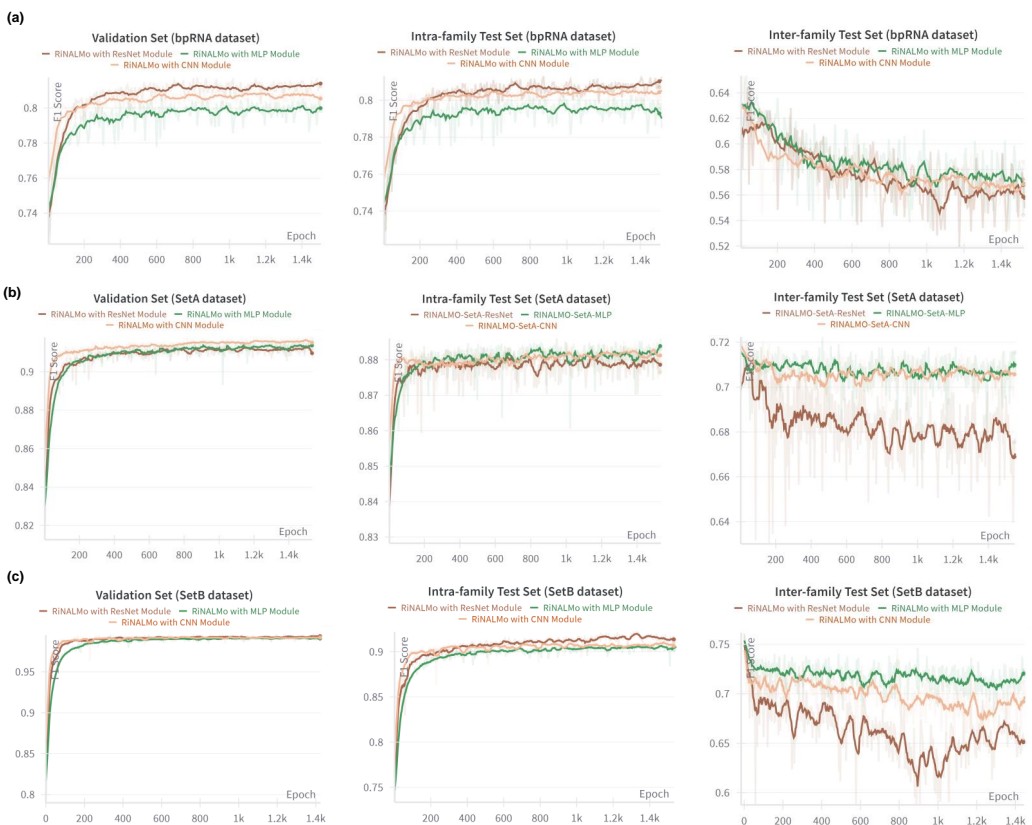

Figure 3: F1 performance of RiNALMo on secondary structure prediction across three datasets. Panels (a), (b), and (c) correspond to bpRNA, SetA, and SetB, respectively. Each panel reports the F1 score on the validation set (left), intra-family test set (middle), and inter-family test set (right) throughout training. Curves represent models using MLP, CNN, and ResNet classifiers applied to frozen pLM embeddings. While validation and intra-family performance improves steadily, inter-family generalization remains limited or declines, highlighting challenges in cross-family structure prediction.

## F.2 Interaction-related Tasks

We evaluate *in vivo* RNA–protein interactions across 22 RNA-binding proteins (RBPs). Detailed results are provided in Table 17 and Table 18, with a summary of overall performance comparison shown in Figure 4.

Overall, RNA pLMs demonstrate stronger performance than DNA pLMs across most datasets. Incorporating increasingly expressive downstream modules—from simple MLPs to shallow CNNs and ResNets—consistently improves performance. Notably, while a few RBP benchmarks show only marginal improvements with deeper downstream modules, most benefit from architectural complexity. This suggests that modules with greater modeling capacity are better able to capture subtle or diffuse binding signals, particularly on RBP datasets where binding motifs or features are poorly characterized.

Among the evaluated models, SpliceBERT consistently achieves robust performance across the 22 RBP datasets, even when utilizing a simple MLP head. This robustness may stem from its pre-training on intronic regions of pre-mRNA, which are common targets for RNA-binding proteins. Such pre-training likely enables SpliceBERT to capture conserved and functionally relevant binding patterns, enhancing its generalizability across diverse RBP datasets.

Table 17: Detailed results for binary binding prediction across 22 datasets, each corresponding to one of 22 RNA-binding proteins (RBPs) in K562 and HepG2 cell lines. The reported values represent *mean (std)* obtained from three independent runs with distinct random seeds. The top-performing and second-best models for each dataset are highlighted in two shades of green, denoted as best and second-best, respectively (Part 1).

| Model | AKAP1_HepG2 F1↑ | BCLAF1_HepG2 F1↑ | DDX24_K562 F1↑ | DDX3X_HepG2 F1↑ | DDX3X_K562 F1↑ | FAM120A_K562 F1↑ | G3BP1_HepG2 F1↑ | GRWD1_HepG2 F1↑ | IGF2BP1_K562 F1↑ | LARP4_HepG2 F1↑ | LIN28B_K562 F1↑ |
|---|---|---|---|---|---|---|---|---|---|---|---|
| One-hot | 0.753(0.007) | 0.733(0.009) | 0.638(0.005) | 0.850(0.001) | 0.808(0.002) | 0.731(0.006) | 0.608(0.016) | 0.654(0.007) | 0.698(0.001) | 0.650(0.003) | 0.603(0.006) |
| Dense | 0.757(0.002) | 0.743(0.008) | 0.651(0.009) | 0.847(0.001) | 0.808(0.002) | 0.736(0.004) | 0.623(0.006) | 0.657(0.002) | 0.695(0.004) | 0.656(0.008) | 0.593(0.008) |
| **(+MLP)** | | | | | | | | | | | |
| DNABERT2_mlp | 0.745(0.005) | 0.697(0.001) | 0.599(0.010) | 0.846(0.001) | 0.807(0.004) | 0.717(0.014) | 0.584(0.007) | 0.621(0.008) | 0.691(0.002) | 0.645(0.008) | 0.584(0.007) |
| HyenaDNA_mlp | 0.722(0.007) | 0.674(0.005) | 0.536(0.026) | 0.846(0.002) | 0.793(0.007) | 0.678(0.018) | 0.527(0.020) | 0.532(0.013) | 0.695(0.002) | 0.580(0.037) | 0.412(0.036) |
| NT_mlp | 0.744(0.002) | 0.676(0.006) | 0.572(0.008) | 0.840(0.006) | 0.796(0.001) | 0.700(0.003) | 0.586(0.001) | 0.593(0.001) | 0.680(0.006) | 0.641(0.009) | 0.467(0.053) |
| RNABERT_mlp | 0.548(0.009) | 0.617(0.007) | 0.456(0.012) | 0.798(0.003) | 0.689(0.007) | 0.501(0.021) | 0.436(0.004) | 0.457(0.012) | 0.615(0.007) | 0.408(0.000) | 0.406(0.000) |
| RNA-FM_mlp | 0.751(0.000) | 0.715(0.001) | 0.628(0.002) | 0.852(0.001) | 0.811(0.004) | 0.751(0.003) | 0.638(0.001) | 0.638(0.008) | 0.734(0.006) | 0.507(0.009) | 0.602(0.015) |
| 3UTRBERT_mlp | 0.788(0.002) | 0.720(0.002) | 0.641(0.003) | 0.854(0.001) | 0.826(0.002) | 0.755(0.001) | 0.661(0.003) | 0.650(0.002) | 0.722(0.002) | 0.609(0.001) | 0.609(0.002) |
| SpliceBERT_mlp | 0.789(0.002) | 0.742(0.002) | 0.697(0.003) | 0.863(0.001) | 0.838(0.002) | 0.747(0.004) | 0.689(0.001) | 0.692(0.004) | 0.743(0.001) | 0.709(0.001) | 0.649(0.001) |
| UTR-LM_mlp | 0.760(0.003) | 0.696(0.007) | 0.595(0.009) | 0.850(0.002) | 0.813(0.002) | 0.735(0.003) | 0.590(0.009) | 0.596(0.030) | 0.701(0.003) | 0.662(0.002) | 0.555(0.055) |
| RiNALMo_mlp | 0.797(0.003) | 0.740(0.008) | 0.664(0.008) | 0.861(0.001) | 0.824(0.004) | 0.768(0.002) | 0.663(0.007) | 0.665(0.004) | 0.742(0.001) | 0.690(0.003) | 0.633(0.004) |
| **(+CNN)** | | | | | | | | | | | |
| DNABERT2_cnn | 0.739(0.005) | 0.697(0.001) | 0.599(0.010) | 0.845(0.004) | 0.799(0.003) | 0.724(0.002) | 0.602(0.006) | 0.614(0.008) | 0.694(0.002) | 0.638(0.010) | 0.589(0.004) |
| HyenaDNA_cnn | 0.760(0.004) | 0.733(0.007) | 0.614(0.023) | 0.851(0.006) | 0.813(0.000) | 0.755(0.007) | 0.613(0.002) | 0.626(0.011) | 0.738(0.004) | 0.643(0.006) | 0.406(0.000) |
| NT_cnn | 0.742(0.008) | 0.689(0.002) | 0.598(0.005) | 0.842(0.004) | 0.796(0.003) | 0.712(0.005) | 0.593(0.007) | 0.608(0.004) | 0.685(0.005) | 0.637(0.003) | 0.453(0.067) |
| RNABERT_cnn | 0.704(0.002) | 0.670(0.002) | 0.527(0.002) | 0.835(0.002) | 0.792(0.002) | 0.685(0.001) | 0.515(0.004) | 0.584(0.006) | 0.679(0.003) | 0.598(0.006) | 0.507(0.009) |
| RNA-FM_cnn | 0.788(0.002) | 0.740(0.002) | 0.658(0.001) | 0.859(0.003) | 0.812(0.001) | 0.761(0.007) | 0.654(0.003) | 0.662(0.001) | 0.743(0.002) | 0.662(0.001) | 0.619(0.003) |
| 3UTRBERT_cnn | 0.789(0.002) | 0.755(0.002) | 0.666(0.003) | 0.853(0.006) | 0.826(0.002) | 0.763(0.002) | 0.663(0.010) | 0.675(0.006) | 0.727(0.002) | 0.701(0.000) | 0.625(0.008) |
| SpliceBERT_cnn | 0.792(0.002) | 0.770(0.000) | 0.712(0.004) | 0.860(0.000) | 0.838(0.003) | 0.759(0.004) | 0.694(0.004) | 0.710(0.003) | 0.746(0.001) | 0.716(0.001) | 0.642(0.010) |
| UTR-LM_cnn | 0.791(0.004) | 0.751(0.001) | 0.666(0.012) | 0.853(0.002) | 0.815(0.003) | 0.759(0.003) | 0.657(0.003) | 0.680(0.008) | 0.740(0.007) | 0.681(0.006) | 0.631(0.000) |
| RiNALMo_cnn | 0.800(0.004) | 0.776(0.002) | 0.720(0.005) | 0.865(0.002) | 0.820(0.006) | 0.778(0.003) | 0.681(0.007) | 0.703(0.002) | 0.751(0.004) | 0.702(0.002) | 0.643(0.006) |
| **(+ResNet)** | | | | | | | | | | | |
| DNABERT2_resnet | 0.726(0.008) | 0.697(0.007) | 0.577(0.001) | 0.843(0.001) | 0.795(0.001) | 0.701(0.021) | 0.572(0.005) | 0.599(0.006) | 0.678(0.008) | 0.639(0.019) | 0.561(0.008) |
| HyenaDNA_resnet | 0.771(0.003) | 0.753(0.001) | 0.683(0.004) | 0.853(0.001) | 0.810(0.001) | 0.717(0.007) | 0.653(0.005) | 0.679(0.005) | 0.732(0.004) | 0.658(0.004) | 0.607(0.009) |
| NT_resnet | 0.738(0.006) | 0.708(0.003) | 0.601(0.014) | 0.841(0.003) | 0.790(0.006) | 0.701(0.007) | 0.601(0.008) | 0.623(0.003) | 0.688(0.015) | 0.624(0.013) | 0.561(0.003) |
| RNABERT_resnet | 0.732(0.010) | 0.721(0.002) | 0.571(0.016) | 0.838(0.008) | 0.789(0.008) | 0.709(0.010) | 0.559(0.008) | 0.628(0.004) | 0.689(0.003) | 0.624(0.015) | 0.528(0.005) |
| RNA-FM_resnet | 0.779(0.006) | 0.746(0.001) | 0.644(0.005) | 0.854(0.004) | 0.811(0.004) | 0.752(0.009) | 0.639(0.005) | 0.665(0.008) | 0.732(0.005) | 0.656(0.002) | 0.609(0.012) |
| 3UTRBERT_resnet | 0.792(0.005) | 0.751(0.001) | 0.660(0.004) | 0.853(0.000) | 0.824(0.002) | 0.748(0.009) | 0.656(0.015) | 0.676(0.005) | 0.721(0.001) | 0.683(0.007) | 0.618(0.004) |
| SpliceBERT_resnet | 0.785(0.002) | 0.776(0.001) | 0.730(0.004) | 0.863(0.001) | 0.840(0.004) | 0.756(0.004) | 0.691(0.013) | 0.720(0.001) | 0.743(0.005) | 0.710(0.005) | 0.646(0.005) |
| UTR-LM_resnet | 0.785(0.008) | 0.745(0.009) | 0.659(0.014) | 0.850(0.003) | 0.821(0.007) | 0.752(0.006) | 0.649(0.004) | 0.673(0.013) | 0.730(0.002) | 0.663(0.011) | 0.620(0.002) |
| RiNALMo_resnet | 0.798(0.004) | 0.781(0.001) | 0.721(0.010) | 0.863(0.006) | 0.823(0.004) | 0.778(0.006) | 0.681(0.002) | 0.711(0.005) | 0.754(0.001) | 0.699(0.007) | 0.641(0.001) |

Table 18: Detailed results for binary binding prediction across 22 datasets, each corresponding to one of 22 RNA-binding proteins (RBPs) in K562 and HepG2 cell lines. The reported values represent *mean (std)* obtained from three independent runs with distinct random seeds. The top-performing and second-best models for each dataset are highlighted in two shades of green, denoted as best and second-best, respectively (Part 2).

| Model | PABPC4_K562 F1↑ | PPIG_HepG2 F1↑ | PUM2_K562 F1↑ | RBM15_K562 F1↑ | RPS3_HepG2 F1↑ | SND1_HepG2 F1↑ | UPF1_HepG2 F1↑ | UPF1_K562 F1↑ | UCHL5_K562 F1↑ | YBX3_K562 F1↑ | ZNF622_K562 F1↑ |
|---|---|---|---|---|---|---|---|---|---|---|---|
| One-hot | 0.682(0.004) | 0.694(0.012) | 0.899(0.002) | 0.646(0.004) | 0.675(0.006) | 0.670(0.005) | 0.765(0.004) | 0.717(0.003) | 0.685(0.003) | 0.634(0.002) | 0.679(0.005) |
| Dense | 0.679(0.003) | 0.698(0.005) | 0.901(0.003) | 0.651(0.006) | 0.678(0.006) | 0.657(0.007) | 0.768(0.004) | 0.721(0.006) | 0.679(0.012) | 0.626(0.005) | 0.673(0.001) |
| **(+MLP)** | | | | | | | | | | | |
| DNABERT2_mlp | 0.655(0.003) | 0.669(0.005) | 0.820(0.003) | 0.658(0.006) | 0.681(0.006) | 0.629(0.007) | 0.755(0.004) | 0.711(0.002) | 0.668(0.005) | 0.591(0.002) | 0.657(0.006) |
| HyenaDNA_mlp | 0.579(0.019) | 0.659(0.002) | 0.794(0.005) | 0.631(0.008) | 0.665(0.003) | 0.609(0.014) | 0.732(0.003) | 0.684(0.013) | 0.608(0.004) | 0.509(0.023) | 0.607(0.007) |
| NT_mlp | 0.647(0.003) | 0.649(0.011) | 0.814(0.002) | 0.638(0.004) | 0.663(0.003) | 0.599(0.019) | 0.743(0.001) | 0.688(0.002) | 0.642(0.002) | 0.635(0.001) | 0.635(0.001) |
| RNABERT_mlp | 0.407(0.000) | 0.567(0.007) | 0.726(0.001) | 0.550(0.011) | 0.605(0.008) | 0.522(0.004) | 0.601(0.014) | 0.480(0.017) | 0.538(0.008) | 0.415(0.000) | 0.519(0.004) |
| RNA-FM_mlp | 0.696(0.001) | 0.684(0.003) | 0.819(0.004) | 0.674(0.002) | 0.698(0.002) | 0.669(0.002) | 0.779(0.005) | 0.740(0.003) | 0.679(0.002) | 0.590(0.018) | 0.672(0.005) |
| 3UTRBERT_mlp | 0.695(0.005) | 0.707(0.002) | 0.853(0.003) | 0.681(0.003) | 0.685(0.001) | 0.662(0.003) | 0.783(0.002) | 0.746(0.003) | 0.702(0.006) | 0.606(0.004) | 0.690(0.003) |
| SpliceBERT_mlp | 0.724(0.003) | 0.746(0.006) | 0.842(0.004) | 0.679(0.001) | 0.708(0.005) | 0.687(0.003) | 0.787(0.003) | 0.744(0.005) | 0.742(0.002) | 0.652(0.002) | 0.727(0.001) |
| UTR-LM_mlp | 0.667(0.007) | 0.673(0.003) | 0.815(0.003) | 0.645(0.007) | 0.671(0.006) | 0.625(0.018) | 0.654(0.004) | 0.766(0.002) | 0.722(0.005) | 0.572(0.001) | 0.650(0.007) |
| RiNALMo_mlp | 0.709(0.004) | 0.715(0.003) | 0.843(0.001) | 0.689(0.002) | 0.704(0.012) | 0.668(0.007) | 0.715(0.002) | 0.790(0.002) | 0.755(0.005) | 0.621(0.002) | 0.714(0.007) |
| **(+CNN)** | | | | | | | | | | | |
| DNABERT2_cnn | 0.670(0.010) | 0.656(0.003) | 0.843(0.006) | 0.657(0.004) | 0.677(0.005) | 0.652(0.004) | 0.750(0.009) | 0.702(0.007) | 0.651(0.002) | 0.618(0.002) | 0.644(0.006) |
| HyenaDNA_cnn | 0.670(0.009) | 0.730(0.003) | 0.883(0.005) | 0.704(0.005) | 0.702(0.008) | 0.669(0.010) | 0.767(0.002) | 0.716(0.004) | 0.657(0.007) | 0.605(0.018) | 0.684(0.007) |
| NT_cnn | 0.641(0.006) | 0.668(0.002) | 0.835(0.006) | 0.658(0.003) | 0.670(0.014) | 0.625(0.007) | 0.752(0.001) | 0.695(0.007) | 0.647(0.002) | 0.646(0.002) | 0.634(0.003) |
| RNABERT_cnn | 0.631(0.010) | 0.607(0.014) | 0.815(0.002) | 0.615(0.004) | 0.642(0.007) | 0.567(0.005) | 0.724(0.002) | 0.684(0.002) | 0.606(0.011) | 0.535(0.008) | 0.613(0.003) |
| RNA-FM_cnn | 0.702(0.002) | 0.722(0.003) | 0.875(0.004) | 0.707(0.004) | 0.709(0.002) | 0.689(0.006) | 0.788(0.002) | 0.755(0.002) | 0.693(0.005) | 0.642(0.001) | 0.697(0.002) |
| 3UTRBERT_cnn | 0.708(0.011) | 0.734(0.000) | 0.890(0.004) | 0.688(0.010) | 0.698(0.007) | 0.678(0.006) | 0.791(0.001) | 0.752(0.001) | 0.718(0.004) | 0.639(0.008) | 0.711(0.004) |
| SpliceBERT_cnn | 0.729(0.000) | 0.769(0.004) | 0.872(0.002) | 0.708(0.002) | 0.708(0.002) | 0.717(0.003) | 0.789(0.002) | 0.759(0.003) | 0.748(0.001) | 0.676(0.002) | 0.743(0.006) |
| UTR-LM_cnn | 0.690(0.037) | 0.744(0.004) | 0.887(0.005) | 0.704(0.011) | 0.698(0.006) | 0.671(0.004) | 0.709(0.004) | 0.785(0.004) | 0.749(0.002) | 0.645(0.006) | 0.716(0.004) |
| RiNALMo_cnn | 0.720(0.005) | 0.764(0.001) | 0.895(0.005) | 0.729(0.003) | 0.722(0.005) | 0.701(0.002) | 0.740(0.006) | 0.793(0.001) | 0.769(0.002) | 0.666(0.003) | 0.748(0.004) |
| **(+ResNet)** | | | | | | | | | | | |
| DNABERT2_resnet | 0.628(0.004) | 0.659(0.005) | 0.823(0.006) | 0.653(0.004) | 0.664(0.009) | 0.684(0.007) | 0.749(0.004) | 0.684(0.007) | 0.626(0.009) | 0.594(0.005) | 0.634(0.012) |
| HyenaDNA_resnet | 0.686(0.003) | 0.739(0.005) | 0.896(0.004) | 0.720(0.004) | 0.699(0.004) | 0.684(0.007) | 0.778(0.003) | 0.737(0.003) | 0.704(0.001) | 0.639(0.002) | 0.710(0.008) |
| NT_resnet | 0.644(0.002) | 0.672(0.009) | 0.829(0.004) | 0.673(0.012) | 0.678(0.006) | 0.644(0.003) | 0.742(0.003) | 0.700(0.015) | 0.655(0.006) | 0.634(0.002) | 0.634(0.002) |
| RNABERT_resnet | 0.647(0.021) | 0.684(0.003) | 0.878(0.005) | 0.624(0.005) | 0.657(0.006) | 0.607(0.006) | 0.755(0.004) | 0.708(0.002) | 0.658(0.008) | 0.606(0.003) | 0.643(0.010) |
| RNA-FM_resnet | 0.686(0.002) | 0.720(0.005) | 0.886(0.006) | 0.706(0.006) | 0.704(0.012) | 0.680(0.003) | 0.780(0.004) | 0.744(0.002) | 0.716(0.003) | 0.631(0.001) | 0.686(0.002) |
| 3UTRBERT_resnet | 0.694(0.007) | 0.732(0.003) | 0.893(0.004) | 0.693(0.008) | 0.685(0.010) | 0.673(0.014) | 0.791(0.001) | 0.743(0.005) | 0.716(0.003) | 0.637(0.004) | 0.704(0.008) |
| SpliceBERT_resnet | 0.716(0.005) | 0.789(0.005) | 0.895(0.002) | 0.713(0.002) | 0.710(0.006) | 0.705(0.001) | 0.788(0.002) | 0.750(0.004) | | 0.675(0.007) | 0.749(0.006) |
| UTR-LM_resnet | 0.689(0.010) | 0.736(0.008) | 0.892(0.006) | 0.693(0.008) | 0.690(0.009) | 0.692(0.007) | 0.699(0.004) | 0.781(0.001) | 0.739(0.004) | 0.639(0.009) | 0.700(0.003) |
| RiNALMo_resnet | 0.712(0.005) | 0.768(0.006) | 0.893(0.004) | 0.720(0.006) | 0.716(0.006) | 0.707(0.002) | 0.746(0.005) | 0.799(0.004) | 0.766(0.004) | 0.667(0.003) | 0.749(0.009) |

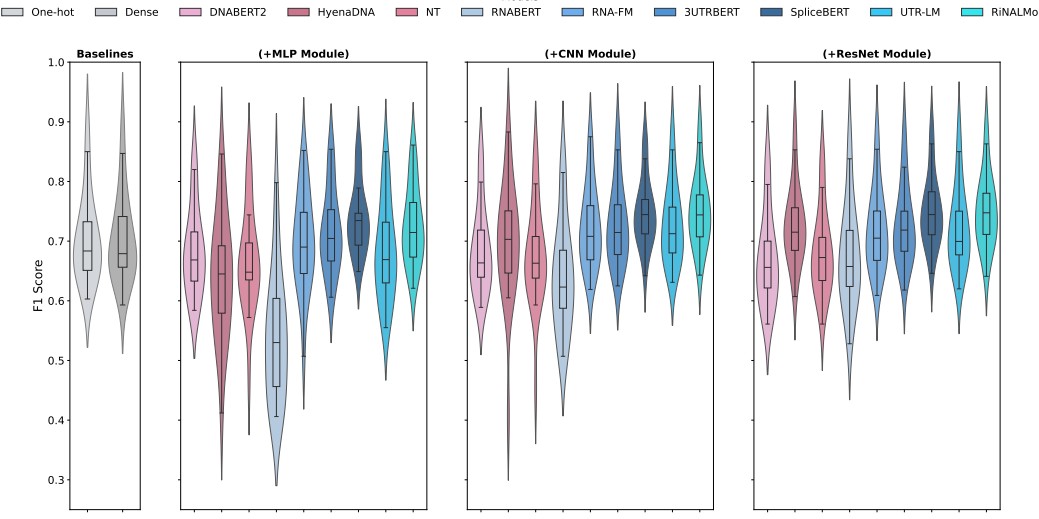

Figure 4: Performance on 22 RBPs datasets for the *in vivo* binary binding prediction task. Baselines (One-hot and Dense) are colored gray, while DNA- and RNA-based pLMs are in pink and blue, respectively.

## F.3 Function-related Tasks

Table 19: Additional benchmark results on pre-mRNA, mRNA and ncRNA tasks.

| Model | pre-mRNA Tasks | | | | mRNA Tasks | | | ncRNA Tasks |
|---|---|---|---|---|---|---|---|---|
| | Splicing Site | | Splicing Event | PAS | Coding Potential Prediction | | | MicroRNA Subcellular |
| | Donor_rice ACC↑ | Acceptor_rice ACC↑ | rice F1↑ | mouse_sp ACC↑ | zebrafish ACC↑ | fruit_fly ACC↑ | s.cerevisiae ACC↑ | HLoss↓ |
| One-hot | $0.754_{(0.018)}$ | $0.735_{(0.017)}$ | $0.350_{(0.032)}$ | $0.638_{(0.012)}$ | $0.900_{(0.005)}$ | $0.924_{(0.009)}$ | $0.721_{(0.017)}$ | $0.308_{(0.001)}$ |
| Dense | $0.738_{(0.006)}$ | $0.720_{(0.015)}$ | $0.380_{(0.025)}$ | $0.633_{(0.014)}$ | $0.900_{(0.007)}$ | $0.872_{(0.074)}$ | $0.713_{(0.012)}$ | $0.307_{(0.000)}$ |
| **(+MLP)** | | | | | | | | |
| DNABERT2_mlp | $0.624_{(0.008)}$ | $0.671_{(0.027)}$ | $0.364_{(0.012)}$ | $0.628_{(0.002)}$ | $0.895_{(0.004)}$ | $0.873_{(0.007)}$ | $0.774_{(0.007)}$ | $0.295_{(0.028)}$ |
| HyenaDNA_mlp | $0.631_{(0.004)}$ | $0.652_{(0.010)}$ | $0.378_{(0.014)}$ | $0.647_{(0.006)}$ | $0.886_{(0.005)}$ | $0.897_{(0.003)}$ | $0.606_{(0.005)}$ | $0.307_{(0.000)}$ |
| NT_mlp | $0.597_{(0.006)}$ | $0.607_{(0.014)}$ | $0.367_{(0.011)}$ | $0.637_{(0.005)}$ | $0.874_{(0.004)}$ | $0.912_{(0.001)}$ | $0.694_{(0.006)}$ | $0.307_{(0.000)}$ |
| RNABERT_mlp | $0.587_{(0.003)}$ | $0.573_{(0.010)}$ | $0.301_{(0.008)}$ | $0.633_{(0.002)}$ | $0.640_{(0.009)}$ | $0.482_{(0.014)}$ | $0.165_{(0.011)}$ | $0.307_{(0.000)}$ |
| RNA-FM_mlp | $0.669_{(0.009)}$ | $0.680_{(0.007)}$ | $0.402_{(0.007)}$ | $0.655_{(0.008)}$ | $0.908_{(0.007)}$ | $0.912_{(0.018)}$ | $0.779_{(0.018)}$ | $0.295_{(0.004)}$ |
| 3UTRBERT_mlp | $0.669_{(0.005)}$ | $0.725_{(0.002)}$ | $0.384_{(0.021)}$ | $0.664_{(0.009)}$ | $0.885_{(0.003)}$ | $0.744_{(0.003)}$ | $0.827_{(0.016)}$ | $0.308_{(0.003)}$ |
| SpliceBERT_mlp | $0.765_{(0.017)}$ | $0.833_{(0.005)}$ | $0.410_{(0.017)}$ | $0.618_{(0.008)}$ | $0.928_{(0.002)}$ | $0.916_{(0.012)}$ | $0.790_{(0.013)}$ | $0.300_{(0.007)}$ |
| UTR-LM_mlp | $0.603_{(0.006)}$ | $0.601_{(0.006)}$ | $0.308_{(0.028)}$ | $0.648_{(0.003)}$ | $0.867_{(0.008)}$ | $0.864_{(0.014)}$ | $0.672_{(0.018)}$ | $0.307_{(0.000)}$ |
| RiNALMo_mlp | $0.734_{(0.003)}$ | $0.776_{(0.002)}$ | $0.400_{(0.010)}$ | $0.649_{(0.019)}$ | $0.937_{(0.004)}$ | $0.948_{(0.012)}$ | $0.713_{(0.010)}$ | $0.278_{(0.006)}$ |
| **(+CNN)** | | | | | | | | |
| DNABERT2_cnn | $0.710_{(0.015)}$ | $0.748_{(0.003)}$ | $0.359_{(0.020)}$ | $0.618_{(0.009)}$ | $0.899_{(0.001)}$ | $0.885_{(0.007)}$ | $0.759_{(0.001)}$ | $0.264_{(0.003)}$ |
| HyenaDNA_cnn | $0.890_{(0.007)}$ | $0.855_{(0.005)}$ | $0.464_{(0.020)}$ | $0.682_{(0.014)}$ | $0.875_{(0.002)}$ | $0.911_{(0.002)}$ | $0.677_{(0.008)}$ | $0.307_{(0.000)}$ |
| NT_cnn | $0.819_{(0.014)}$ | $0.777_{(0.033)}$ | $0.384_{(0.006)}$ | $0.640_{(0.006)}$ | $0.892_{(0.002)}$ | $0.905_{(0.005)}$ | $0.719_{(0.024)}$ | $0.307_{(0.000)}$ |
| RNABERT_cnn | $0.821_{(0.011)}$ | $0.778_{(0.003)}$ | $0.345_{(0.021)}$ | $0.645_{(0.002)}$ | $0.752_{(0.003)}$ | $0.834_{(0.003)}$ | $0.538_{(0.014)}$ | $0.307_{(0.000)}$ |
| RNA-FM_cnn | $0.891_{(0.003)}$ | $0.848_{(0.004)}$ | $0.499_{(0.018)}$ | $0.681_{(0.013)}$ | $0.912_{(0.001)}$ | $0.888_{(0.023)}$ | $0.713_{(0.036)}$ | $0.268_{(0.007)}$ |
| 3UTRBERT_cnn | $0.883_{(0.010)}$ | $0.846_{(0.009)}$ | $0.469_{(0.011)}$ | $0.648_{(0.012)}$ | $0.901_{(0.003)}$ | $0.819_{(0.020)}$ | $0.807_{(0.009)}$ | $0.301_{(0.005)}$ |
| SpliceBERT_cnn | $0.894_{(0.018)}$ | $0.892_{(0.011)}$ | $0.535_{(0.014)}$ | $0.638_{(0.017)}$ | $0.924_{(0.004)}$ | $0.872_{(0.012)}$ | $0.719_{(0.041)}$ | $0.307_{(0.000)}$ |
| UTR-LM_cnn | $0.800_{(0.007)}$ | $0.798_{(0.008)}$ | $0.447_{(0.012)}$ | $0.663_{(0.013)}$ | $0.911_{(0.005)}$ | $0.869_{(0.008)}$ | $0.748_{(0.026)}$ | $0.307_{(0.000)}$ |
| RiNALMo_cnn | $0.831_{(0.003)}$ | $0.850_{(0.002)}$ | $0.450_{(0.007)}$ | $0.647_{(0.003)}$ | $0.906_{(0.001)}$ | $0.925_{(0.002)}$ | $0.669_{(0.039)}$ | $0.264_{(0.010)}$ |
| **(+ResNet)** | | | | | | | | |
| DNABERT2_resnet | $0.858_{(0.010)}$ | $0.854_{(0.006)}$ | $0.421_{(0.011)}$ | $0.625_{(0.010)}$ | $0.887_{(0.007)}$ | $0.895_{(0.008)}$ | $0.732_{(0.014)}$ | $0.294_{(0.031)}$ |
| HyenaDNA_resnet | $0.900_{(0.010)}$ | $0.853_{(0.014)}$ | $0.437_{(0.013)}$ | $0.697_{(0.007)}$ | $0.899_{(0.005)}$ | $0.914_{(0.004)}$ | $0.643_{(0.003)}$ | $0.285_{(0.017)}$ |
| NT_resnet | $0.896_{(0.009)}$ | $0.864_{(0.002)}$ | $0.566_{(0.012)}$ | $0.655_{(0.007)}$ | $0.884_{(0.003)}$ | $0.913_{(0.005)}$ | $0.637_{(0.007)}$ | $0.309_{(0.020)}$ |
| RNABERT_resnet | $0.901_{(0.006)}$ | $0.843_{(0.009)}$ | $0.538_{(0.026)}$ | $0.641_{(0.006)}$ | $0.790_{(0.010)}$ | $0.834_{(0.009)}$ | $0.637_{(0.038)}$ | $0.308_{(0.001)}$ |
| RNA-FM_resnet | $0.917_{(0.005)}$ | $0.869_{(0.004)}$ | $0.550_{(0.033)}$ | $0.679_{(0.013)}$ | $0.911_{(0.002)}$ | $0.871_{(0.037)}$ | $0.704_{(0.037)}$ | $0.263_{(0.011)}$ |
| 3UTRBERT_resnet | $0.919_{(0.003)}$ | $0.876_{(0.003)}$ | $0.514_{(0.011)}$ | $0.663_{(0.015)}$ | $0.889_{(0.006)}$ | $0.840_{(0.004)}$ | $0.817_{(0.014)}$ | $0.302_{(0.020)}$ |
| SpliceBERT_resnet | $0.940_{(0.003)}$ | $0.904_{(0.005)}$ | $0.562_{(0.012)}$ | $0.641_{(0.013)}$ | $0.931_{(0.007)}$ | $0.914_{(0.011)}$ | $0.737_{(0.028)}$ | $0.273_{(0.011)}$ |
| UTR-LM_resnet | $0.854_{(0.009)}$ | $0.829_{(0.004)}$ | $0.471_{(0.006)}$ | $0.657_{(0.011)}$ | $0.920_{(0.004)}$ | $0.911_{(0.019)}$ | $0.745_{(0.010)}$ | $0.294_{(0.005)}$ |
| RiNALMo_resnet | $0.854_{(0.003)}$ | $0.857_{(0.015)}$ | $0.502_{(0.011)}$ | $0.676_{(0.012)}$ | $0.932_{(0.000)}$ | $0.972_{(0.002)}$ | $0.843_{(0.000)}$ | $0.266_{(0.001)}$ |

Table 20: Benchmark results for gRNA efficiency prediction across six datasets from various cell lines and organisms, including human, mouse, and zebrafish. The pLMs did not demonstrate performance advantages over the baselines, including One-hot and Dense models.

| Model | gRNA | | | | | | |
|---|---|---|---|---|---|---|---|
| | mESC (Koike-Yusa) Spearman $p\uparrow$ | HEL (Labuhn) Spearman $p\uparrow$ | A375 (Shalem) Spearman $p\uparrow$ | HEK293T (Xi Xiang) Spearman $p\uparrow$ | Zebrafish (Gagnon) Spearman $p\uparrow$ | Zebrafish (Shkumatava) Spearman $p\uparrow$ | Average |
| One-hot | $0.262_{(0.028)}$ | $0.095_{(0.011)}$ | $0.202_{(0.016)}$ | $0.457_{(0.043)}$ | $0.307_{(0.033)}$ | $0.191_{(0.017)}$ | 0.252 |
| Dense | $0.294_{(0.026)}$ | $0.105_{(0.006)}$ | $0.207_{(0.020)}$ | $0.488_{(0.011)}$ | $0.264_{(0.029)}$ | $0.166_{(0.022)}$ | 0.254 |
| **(+MLP)** | | | | | | | |
| DNABERT2_mlp | $0.149_{(0.010)}$ | $0.019_{(0.014)}$ | $0.098_{(0.016)}$ | $0.207_{(0.016)}$ | $0.206_{(0.020)}$ | $0.101_{(0.024)}$ | 0.130 |
| HyenaDNA_mlp | $0.140_{(0.005)}$ | $0.080_{(0.008)}$ | $0.143_{(0.002)}$ | $0.324_{(0.016)}$ | $0.239_{(0.022)}$ | $0.143_{(0.008)}$ | 0.178 |
| NT_mlp | $0.146_{(0.006)}$ | $0.025_{(0.004)}$ | $0.118_{(0.005)}$ | $0.325_{(0.013)}$ | $0.210_{(0.011)}$ | $0.201_{(0.010)}$ | 0.171 |
| RNABERT_mlp | $0.115_{(0.005)}$ | $0.060_{(0.003)}$ | $0.032_{(0.002)}$ | $0.278_{(0.003)}$ | $0.218_{(0.007)}$ | $0.071_{(0.023)}$ | 0.129 |
| RNA-FM_mlp | $0.193_{(0.002)}$ | $-0.015_{(0.001)}$ | $0.146_{(0.003)}$ | $0.386_{(0.007)}$ | $0.323_{(0.012)}$ | $0.168_{(0.013)}$ | 0.200 |
| 3UTRBERT_mlp | $0.163_{(0.006)}$ | $0.048_{(0.001)}$ | $0.111_{(0.006)}$ | $0.279_{(0.038)}$ | $0.206_{(0.037)}$ | $0.134_{(0.008)}$ | 0.157 |
| SpliceBERT_mlp | $0.142_{(0.015)}$ | $0.017_{(0.012)}$ | $0.081_{(0.019)}$ | $0.233_{(0.069)}$ | $0.183_{(0.039)}$ | $0.151_{(0.035)}$ | 0.134 |
| UTR-LM_mlp | $0.156_{(0.011)}$ | $0.014_{(0.006)}$ | $0.140_{(0.003)}$ | $0.294_{(0.006)}$ | $0.267_{(0.020)}$ | $0.132_{(0.011)}$ | 0.167 |
| RiNALMo_mlp | $0.163_{(0.005)}$ | $0.022_{(0.026)}$ | $0.127_{(0.021)}$ | $0.324_{(0.029)}$ | $0.284_{(0.001)}$ | $0.183_{(0.030)}$ | 0.184 |
| **(+CNN)** | | | | | | | |
| DNABERT2_cnn | $0.145_{(0.006)}$ | $0.065_{(0.010)}$ | $0.084_{(0.010)}$ | $0.223_{(0.015)}$ | $0.212_{(0.009)}$ | $0.094_{(0.032)}$ | 0.137 |
| HyenaDNA_cnn | $0.218_{(0.007)}$ | $0.089_{(0.004)}$ | $0.174_{(0.008)}$ | $0.392_{(0.016)}$ | $0.300_{(0.007)}$ | $0.196_{(0.012)}$ | 0.228 |
| NT_cnn | $0.241_{(0.004)}$ | $0.067_{(0.012)}$ | $0.232_{(0.006)}$ | $0.485_{(0.007)}$ | $0.208_{(0.014)}$ | $0.264_{(0.001)}$ | 0.249 |
| RNABERT_cnn | $0.222_{(0.011)}$ | $0.141_{(0.009)}$ | $0.119_{(0.003)}$ | $0.395_{(0.012)}$ | $0.201_{(0.012)}$ | $0.091_{(0.007)}$ | 0.195 |
| RNA-FM_cnn | $0.253_{(0.008)}$ | $0.072_{(0.005)}$ | $0.208_{(0.014)}$ | $0.519_{(0.015)}$ | $0.293_{(0.024)}$ | $0.193_{(0.020)}$ | 0.256 |
| 3UTRBERT_cnn | $0.207_{(0.014)}$ | $0.033_{(0.003)}$ | $0.130_{(0.023)}$ | $0.316_{(0.035)}$ | $0.236_{(0.014)}$ | $0.128_{(0.012)}$ | 0.175 |
| SpliceBERT_cnn | $0.198_{(0.027)}$ | $0.041_{(0.016)}$ | $0.102_{(0.012)}$ | $0.343_{(0.021)}$ | $0.274_{(0.026)}$ | $0.166_{(0.034)}$ | 0.187 |
| UTR-LM_cnn | $0.197_{(0.004)}$ | $0.070_{(0.017)}$ | $0.155_{(0.002)}$ | $0.348_{(0.012)}$ | $0.296_{(0.036)}$ | $0.191_{(0.023)}$ | 0.209 |
| RiNALMo_cnn | $0.269_{(0.023)}$ | $0.073_{(0.031)}$ | $0.216_{(0.013)}$ | $0.480_{(0.023)}$ | $0.302_{(0.016)}$ | $0.180_{(0.010)}$ | 0.253 |
| **(+ResNet)** | | | | | | | |
| DNABERT2_resnet | $0.131_{(0.011)}$ | $0.043_{(0.016)}$ | $0.073_{(0.020)}$ | $0.180_{(0.007)}$ | $0.131_{(0.034)}$ | $0.060_{(0.053)}$ | 0.103 |
| HyenaDNA_resnet | $0.229_{(0.043)}$ | $0.111_{(0.018)}$ | $0.180_{(0.025)}$ | $0.423_{(0.076)}$ | $0.207_{(0.060)}$ | $0.146_{(0.046)}$ | 0.216 |
| NT_resnet | $0.179_{(0.022)}$ | $0.033_{(0.045)}$ | $0.162_{(0.018)}$ | $0.382_{(0.015)}$ | $0.173_{(0.089)}$ | $0.206_{(0.049)}$ | 0.189 |
| RNABERT_resnet | $0.188_{(0.024)}$ | $0.102_{(0.026)}$ | $0.124_{(0.019)}$ | $0.359_{(0.046)}$ | $0.172_{(0.015)}$ | $0.037_{(0.021)}$ | 0.164 |
| RNAFM_resnet | $0.229_{(0.015)}$ | $0.045_{(0.024)}$ | $0.175_{(0.008)}$ | $0.440_{(0.044)}$ | $0.273_{(0.040)}$ | $0.188_{(0.036)}$ | 0.225 |
| 3UTRBERT_resnet | $0.169_{(0.022)}$ | $0.039_{(0.007)}$ | $0.126_{(0.007)}$ | $0.319_{(0.014)}$ | $0.230_{(0.033)}$ | $0.110_{(0.026)}$ | 0.166 |
| SpliceBERT_resnet | $0.192_{(0.033)}$ | $0.047_{(0.037)}$ | $0.131_{(0.024)}$ | $0.349_{(0.061)}$ | $0.232_{(0.020)}$ | $0.153_{(0.036)}$ | 0.184 |
| UTR-LM_resnet | $0.240_{(0.011)}$ | $0.074_{(0.018)}$ | $0.191_{(0.010)}$ | $0.445_{(0.020)}$ | $0.345_{(0.045)}$ | $0.183_{(0.030)}$ | 0.246 |
| RiNALMo_resnet | $0.233_{(0.020)}$ | $0.049_{(0.049)}$ | $0.209_{(0.006)}$ | $0.468_{(0.011)}$ | $0.287_{(0.017)}$ | $0.177_{(0.030)}$ | 0.237 |

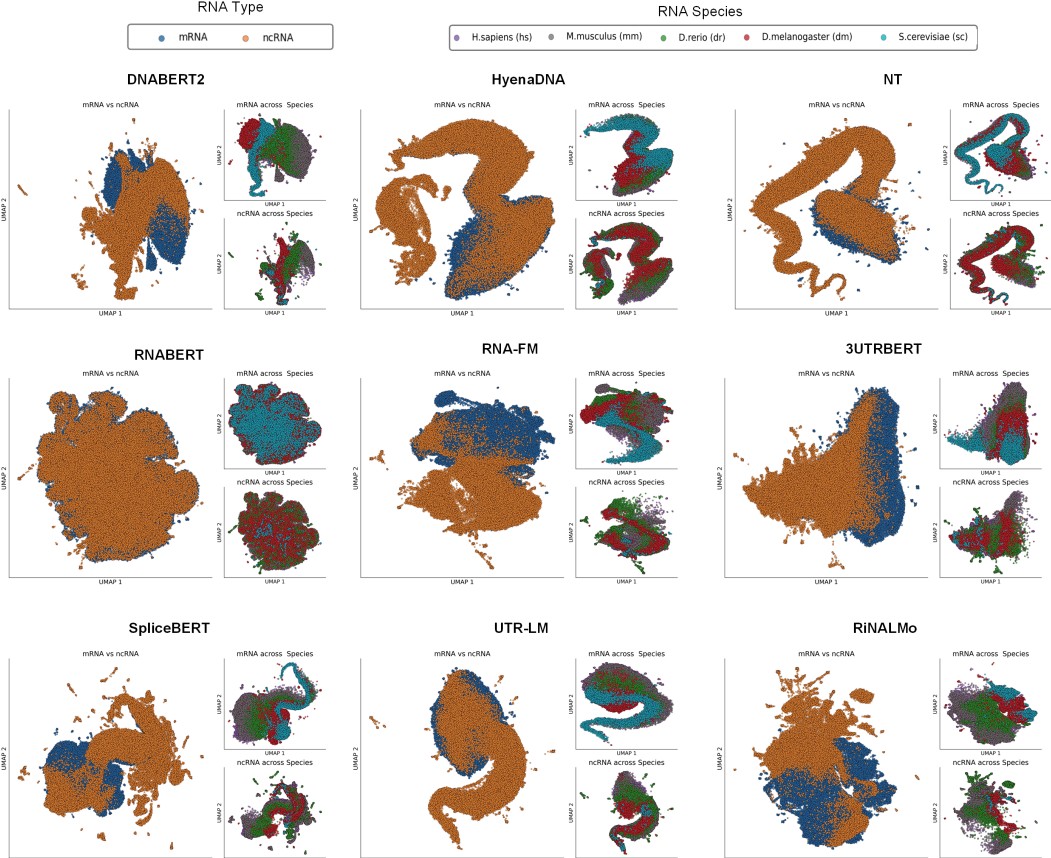

Figure 5: UMAP visualization of mRNA and ncRNA representations from different pLMs across five species: human, mouse, zebrafish, fruit fly, and yeast. The left panel shows mRNA (blue) and ncRNA (yellow) distributions. The right panels further separate mRNA (top) and ncRNA (bottom), with species distinguished by color. DNA pLMs, including DNABERT2, and RNA pLMs, including RNA-FM, 3UTRBERT, SpliceBERT and RiNALMo, demonstrate zero-shot capabilities in distinguishing between mRNA and ncRNA, as well as across different species. This highlights the distinct semantic representations of mRNA and ncRNA, which can be captured by pLMs. Furthermore, there are particularly notable semantic similarities in mRNA across species that are evolutionarily closely related, such as human and mouse.

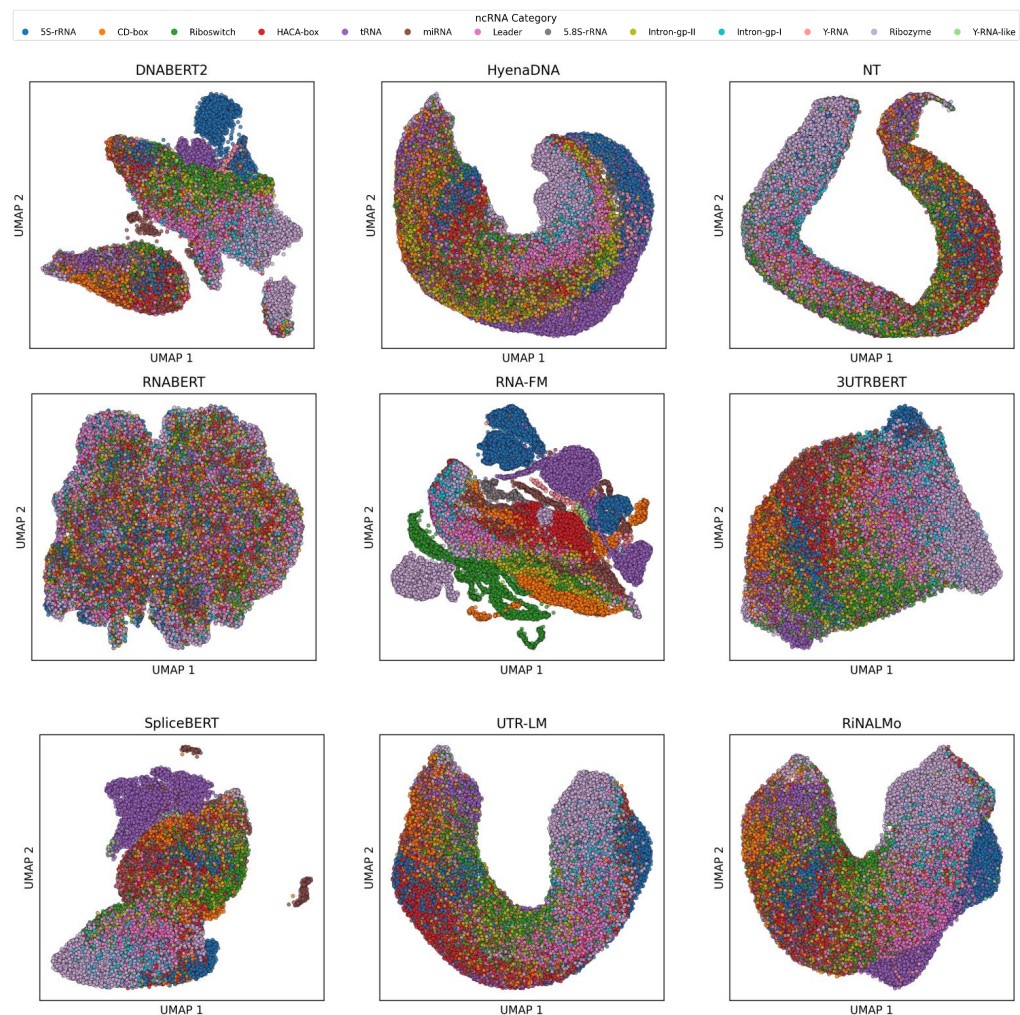

Figure 6: UMAP visualization of 13 distinct ncRNA types using representations from different pLMs. The RNA-FM model, pre-trained on 23 million ncRNAs, demonstrated superior zero-shot capabilities in distinguishing ncRNA types, highlighting its ability to capture the unique semantic characteristics of various ncRNA classes.

## G  Details of Compute Resources

All benchmark experiments were executed on eight 40GB A100 GPUs and two 80GB A800 GPUs. Most tasks were run on the A100 GPUs, while the Contact Map Prediction task, requiring larger memory capacity, was executed on the 80GB A800 GPUs.

## H  Open-Source Platform: Leaderboard, Datasets, and Code

RNAscope offers an open-source platform with public leaderboards (`https://rnascope-board.github.io/`) to promote transparency and standardize RNA model evaluations. It enables fair and reproducible comparisons of RNA pre-trained language models (pLMs) across various tasks. In addition to the leaderboard, RNAscope provides access to datasets[7] and code[8], enabling researchers to replicate, refine, and build on existing work. These resources encourage community participation, model testing, and benchmark development.

RNAscope aims to foster collaboration between computational and experimental researchers to discover novel RNA molecules beyond evolutionary patterns. By offering a broader perspective, RNAscope enhances the understanding of RNA models' strengths and limitations, thereby improving their practical application. As related technologies advance, additional datasets and tasks will be incorporated, including single-cell RNA data and expanded tasks such as predicting cell-type or tissue-specific patterns.

With datasets covering various RNA types and species, the platform enables evaluations across multiple biological contexts, improving model generalization. RNAscope ultimately aims to facilitate the development of more robust RNA models.

---

[7]https://kaggle.com/datasets/b0aeeed2f6b3dfd43c1ab33b58467a466308d056b6868d861ad00e1074ce384d
[8]https://anonymous.4open.science/r/RNAscope

