# OpenReview forum: "RNAscope: Benchmarking RNA Language Models for RNA Sequence Understanding"
_NeurIPS.cc/2025/Datasets_and_Benchmarks_Track — Submitted to NeurIPS 2025 Datasets and Benchmarks Track_

### Official Review · Reviewer_okrs · 2025-06-05

**Rating:** 4
**Confidence:** 3

**Summary:**

RNAscope is a comprehensive benchmarking framework for evaluating RNA pre-trained language models (pLMs) across structure prediction, interaction classification, and function characterization tasks. Covering 1,253 diverse experiments, it enables systematic and unbiased model comparison. Results highlight persistent challenges in generalizing across RNA families and contexts, positioning RNAscope as a robust tool to advance RNA modeling.

**Dataset Code Accessibility:**

Yes

**Dataset Code Comments:**

The dataset is available on https://www.kaggle.com/datasets/b0aeeed2f6b3dfd43c1ab33b58467a466308d056b6868d861ad00e1074ce384d.

**Ethical Considerations:**

No, there are no or only very minor ethics concerns

**Final Justification:**

Thank you for your detailed rebuttal. I have read it carefully and it has successfully resolved some of my initial questions about the work.

Here is a summary of my final assessment:

Resolved Issues: Your rebuttal clarified my confusion regarding the DNA models.

Remaining Issues / Uncertainty: My primary hesitation in giving a higher score stems from the core contributions related to the specific details of the RNA domain. As I indicated with my confidence score of 3/5, this is not my primary area of expertise. Therefore, while the methodology appears sound from a machine learning perspective, I cannot confidently vouch for the technical novelty or the significance of the contribution within the specialized field of RNA research. The other expert reviewers will be better positioned to assess this crucial aspect.

Weighting: I am weighing the general clarity and methodological soundness of the paper positively. However, my uncertainty about the core domain-specific contribution is a significant factor, preventing me from recommending a stronger acceptance at this stage.

In summary, the authors have done a good job clarifying parts of their work. My final recommendation is based on the aspects I can confidently evaluate, balanced by my stated limitations.

Final Rating: 4/6
Confidence: 3/5

**Limitations Weaknesses:**

1. Missing Important Model Evaluations: The benchmark does not include evaluations of three important models released before March 1st, 2025: EVO1 [1], EVO2 [2], and GENERATOR [3]. Including these models would make the comparison more complete.

[1] Nguyen E, Poli M, Durrant M G, et al. Sequence modeling and design from molecular to genome scale with Evo[J]. Science, 2024, 386(6723): eado9336.

[2] Brixi G, Durrant M G, Ku J, et al. Genome modeling and design across all domains of life with Evo 2[J]. BioRxiv, 2025: 2025.02. 18.638918.

[3] Wu W, Li Q, Li M, et al. GENERator: A Long-Context Generative Genomic Foundation Model[J]. arXiv preprint arXiv:2502.07272, 2025.

**Strengths Contributions:**

1. Good Visualization: The visual presentation is clear and attractive, especially Figure 1, which helps readers understand the framework effectively.
2. Large-Scale Experiments: The work conducts a large number of experiments, covering various subtasks and complexity levels.
3. Detailed Task Descriptions: The paper provides detailed descriptions of each task, which helps improve clarity and reproducibility.

---

> ### Author Rebuttal · Authors · 2025-07-30
>
> &nbsp;
>
> We genuinely appreciate the reviewer *okrs*'s encouraging feedback and positive comments. According to the recommendations, we have provided the following information:
>
> &nbsp;
>
> > **Q17. More benchmark models**.
>
> We thank the reviewer for their positive feedback and for highlighting the importance of including recently released DNA language models—Evo1, Evo2, and GENERator.
>
> For the Evo series models, we currently have tested the Evo2 base model (*evo2_1b_base*) due to resource constraints. Evo2 poor performance was observed across all tasks, which might due to the extreme smallness of its representation embedding activations,  with values on the order of $10^{-8}$, e.g.
>
> ```Python
> tensor([[[ 0.0000000257,  0.0000013113,  0.0000000594, ...,  0.0000000591,
>          -0.0000000168, -0.0000000997],
>          ...
>           [ 0.0000000210,  0.0000009537,  0.0000000433, ...,  0.0000000442,
>          -0.0000000120, -0.0000000712]]])
> ```
>
> This instability in activation values has been widely discussed in the Evo2 GitHub issues.  For this reason, we excluded Evo2 from our benchmark until the developers release official embedding‑scaling or normalization guidelines.
>
> Below are detailed results for the DNA model, GENERator (*GENERator-eukaryote-1.2b-base*) and the two additional RNA models, RNAErnie and AIDO.RNA (*AIDO.RNA-650M*).  We have also provided further analysis in **Appendix K** ("Comprehensive DNA vs. RNA Model Performance") in the manuscript, as follows:
>
> "*For DNA language models pretrained on whole‑genome sequences, their performance on RNA tasks fails to exceed that of many RNA models (e.g., RNA‑FM)—even the GENERator model with 1.2 B‑parameters. This gap might stem from DNA modeling's emphasis on long‑range dependencies, which drives the design of specialized architectures and tokenizers, whereas RNA tasks demand fine‑grained, single‑nucleotide understanding. Differences in pre‑training corpora may further exacerbate this discrepancy. Overall, these findings underscore the strengths of our RNAscope benchmark: by applying a unified evaluation pipeline across every model and task and covering the full spectrum of RNA structure, interaction, and function challenges, RNAscope provides the most comprehensive and granular comparison to date.*"
>
> &nbsp;
>
> **Table R2:** Results for structure‑related tasks. **Bold** marks the best score achieved by fine‑tuning only the specified module across all pre‑trained models. (Note: GENERator's discontinuous 6‑mer tokenizer is not suited for single‑nucleotide‑resolution structure tasks.)
>
> |                  | SSP (bpRNA-1m-intra) | SSP (bpRNA-1m-inter) | SSP (SetA-intra) | SSP (SetA-inter) | SSP (SetB-intra) | SSP (SetB-inter) | CRP (TestS) | CRP (TestL) | CMP       | CMP       |
> | ---------------- | ------------------- | ------------------- | --------------- | --------------- | --------------- | --------------- | ----------- | ---------- | --------- | --------- |
> |                  | F1                  | F1                  | F1              | F1              | F1              | F1              | MAE         | MAE        | short@L/5 | Long@L/5  |
> | (+MLP module)    |                     |                     |                 |                 |                 |                 |             |            |           |           |
> | RNAErnie         | 0.671               | 0.579               | 0.704           | 0.600           | 0.563           | 0.565           | 0.233       | 0.244      | 0.081     | 0.083     |
> | AIDO.RNA         | 0.746               | **0.641**           | 0.838           | 0.710           | 0.839           | 0.684           | 0.198       | 0.198      | 0.176     | 0.132     |
> | (+CNN module)    |                     |                     |                 |                 |                 |                 |             |            |           |           |
> | RNAErnie         | 0.684               | 0.568               | 0.738           | 0.592           | 0.680           | 0.538           | 0.202       | 0.191      | 0.171     | 0.146     |
> | AIDO.RNA         | 0.791               | **0.703**           | 0.879           | **0.774**       | **0.915**       | 0.707           | 0.182       | 0.177      | 0.182     | 0.134     |
> | (+ResNet module) |                     |                     |                 |                 |                 |                 |             |            |           |           |
> | RNAErnie         | 0.726               | 0.567               | 0.769           | 0.586           | 0.785           | 0.520           | 0.177       | 0.175      | **0.190** | **0.173** |
> | AIDO.RNA         | 0.786               | **0.692**           | 0.874           | **0.767**       | **0.907**       | **0.715**       | 0.177       | 0.164      | 0.123     | 0.121     |
>
> &nbsp;
>
> **Table R3:** Results for interaction‑related tasks.
>
> |                  | BBP (22 average) | SBR (DAse) | SBR (TARDBP) | SBR (TSLETS) | BAP (GFP) | BAP (NELF) |
> | ---------------- | ---------------- | --------- | ----------- | ----------- | -------- | --------- |
> |                  | F1               | F1        | F1          | F1          | Spear p  | Spear p   |
> | (+MLP module)    |                  |           |             |             |          |           |
> | GENERator        | 0.674            | 0.550     | 0.360       | 0.322       | 0.120    | 0.212     |
> | RNAErnie         | 0.671            | 0.557     | 0.367       | 0.353       | 0.101    | 0.125     |
> | AIDO.RNA         | 0.708            | 0.610     | 0.405       | 0.368       | 0.060    | 0.083     |
> | (+CNN module)    |                  |           |             |             |          |           |
> | GENERator        | 0.671            | 0.563     | 0.402       | 0.362       | 0.056    | 0.012     |
> | RNAErnie         | 0.716            | 0.596     | 0.434       | 0.373       | 0.210    | 0.184     |
> | AIDO.RNA         | 0.730            | 0.630     | 0.459       | 0.408       | 0.057    | 0.125     |
> | (+ResNet module) |                  |           |             |             |          |           |
> | GENERator        | 0.733            | 0.594     | 0.453       | 0.396       | 0.093    | 0.229     |
> | RNAErnie         | 0.734            | 0.613     | 0.440       | 0.398       | 0.118    | 0.217     |
> | AIDO.RNA         | 0.733            | 0.610     | 0.460       | 0.408       | 0.209    | 0.648     |
>
> &nbsp;
>
> **Table R4:** Results for function‑related tasks. **Bold** marks the best score achieved by fine‑tuning only the specified module across all pre‑trained models.
>
> |                  | SPS  (Donor_human) | SPS  (Acceptor_human) | SPE (human) | PAS (human) | CPP (human) | mSL       | RLP    | NCC       | miSL  | gEP       |
> | ---------------- | ------------------ | ------------------- | ---------- | ---------- | ---------- | --------- | ------ | --------- | ----- | --------- |
> |                  | ACC                | ACC                 | F1         | ACC        | ACC        | F1        | R²     | ACC       | F1    | Spear p   |
> | (+MLP module)    |                    |                     |            |            |            |           |        |           |       |           |
> | GENERator        | 0.827              | 0.826               | 0.544      | 0.731      | 0.962      | 0.553     | 0.135  | 0.736     | 0.322 | 0.168     |
> | RNAErnie         | 0.761              | 0.792               | 0.535      | 0.722      | 0.823      | 0.479     | -0.016 | 0.838     | 0.348 | 0.178     |
> | AIDO.RNA         | 0.798              | 0.794               | 0.583      | 0.759      | 0.955      | 0.534     | 0.218  | **0.986** | 0.398 | 0.186     |
> | (+CNN module)    |                    |                     |            |            |            |           |        |           |       |           |
> | GENERator        | 0.894              | 0.852               | 0.630      | 0.740      | **0.971**  | 0.526     | 0.137  | 0.874     | 0.348 | 0.239     |
> | RNAErnie         | 0.959              | 0.942               | 0.776      | 0.782      | 0.869      | 0.484     | 0.472  | 0.933     | 0.348 | 0.255     |
> | AIDO.RNA         | 0.891              | 0.867               | 0.698      | 0.777      | 0.960      | 0.526     | 0.494  | **0.989** | 0.398 | 0.235     |
> | (+ResNet module) |                    |                     |            |            |            |           |        |           |       |           |
> | GENERator        | 0.954              | 0.940               | 0.806      | 0.754      | **0.973**  | **0.613** | 0.509  | 0.780     | 0.348 | 0.234     |
> | RNAErnie         | 0.970              | 0.959               | 0.800      | 0.785      | 0.899      | 0.545     | 0.577  | 0.962     | 0.330 | **0.253** |
> | AIDO.RNA         | 0.933              | 0.932               | 0.686      | 0.780      | 0.966      | 0.615     | 0.653  | **0.986** | 0.390 | 0.250     |

---

> > ### Comment · Reviewer_okrs · 2025-08-01
> > **Response to rebuttal**
> >
> > Thank you for your response. I've reviewed your rebuttal, and it has clarified several points concerning DNA models.

---

> > > ### Author Response · Authors · 2025-08-01
> > > **Thanks for your rebuttal !**
> > >
> > > We appreciate your supportive feedback and recognizing our efforts to address the DNA models’ constraints on the RNA sequence tasks. We will meticulously integrate these valuable insights into our manuscript!

---

### Official Review · Reviewer_Mk3U · 2025-07-02

**Rating:** 4
**Confidence:** 4

**Summary:**

This manuscript presents RNAscope, yet another benchmark for RNA foundation models.

This manuscript collects 15 RNA benchmarks across 3 task domains, encompassing 1253 experiments.

**Dataset Code Accessibility:**

Yes

**Ethical Considerations:**

No, there are no or only very minor ethics concerns

**Final Justification:**

Taking into account the authors' efforts and the reviewers' comments, I am inclined to recommend a Borderline Accept.

**Limitations Weaknesses:**

### Wrong Dataset Setting

#### Secondary Structure

##### bpRNA$^1$

The bpRNA is an annotation tool capable of parsing RNA structures.

The bpRNA-1m$^1$ is a dataset of 1 million RNA sequences with their secondary structures annotated by bpRNA.
The TR0, VL0, and TS0 datasets are subsets of the bpRNA-1m dataset, which are used for training, validation, and testing, respectively$^2$.

The bpRNA-new$^3$ dataset is another dataset that uses the bpRNA tool to annotate RNA secondary structures.

Both the bpRNA-1m and bpRNA-new datasets uses sequences from the Rfam$^4$, but from different versions and with different family selections.

It is inappropriate to collectively refer these datasets into the bpRNA, as bpRNA is an annotation tool, rather than a dataset.

##### Rivas$^5$

The Rivas$^5$ dataset is a widely used benchmark for evaluating the cross-family capacity of RNA secondary structure prediction models.
However, the correct setting for this dataset is to use the TrainSet-A for training, TestSet-A for validation, and TestSet-B for testing$^3$.
This is because the TrainSet-A and TestSet-A share the same families, and the TrainSet-B and TestSet-B share the same families.
For evaluating the cross-family capacity, the TrainSet-B should not be used for training, as it contains sequences from families that are present in the TestSet-B.

#### Chemical Reactivity

The Ribonanza$^6$ dataset is a large-scale dataset for evaluating RNA chemical reactivity prediction models.

The Ribonanza dataset contains 806,578 sequences for training, 311,935 sequences for validation, and 1,031,888 sequences for testing (Supplementary Table S6).
Which significantly differs from the data count reported in the manuscript (144,918 for training, 16,103 for validation, and 6,787 for testing).

The authors are referred to the [Google Drive](https://drive.google.com/drive/folders/15dTkZoVHioacjgcGfwsYBF7xyt4qYOVc) for accessing the sequences and labels of the Ribonanza dataset.

### Questionable Task Selection

2/3 of the Functional tasks are messenger RNA (mRNA) tasks, which are not suitable for RNA foundation models.

RNA foundation models mostly focus on non-coding RNA sequences, such as miRNA, lncRNA, and circRNA.
This is because messenger RNA are directly translated into proteins, and thus it is more straightforward to use protein foundation models for these tasks.
Most RNA foundation models are pre-trained solely on non-coding RNA sequences, such as RNA-FM, RNAErnie, Ernie-RNA.
To the best of my knowledge, only the RiNALMo and the Uni-RNA models have messenger RNA (mRNA) sequences in their pre-training datasets.
However, neither of these models are evaluated in this benchmark.

### Limited Evaluation

The benchmark misses some important RNA foundation models, such as RiNALMo$^7$, RNAErnie$^8$, AIDO.RNA$^9$.

### Miscellaneous Comments

In Table 3, the Secondary Structure is incorrectly listed as Second Structure.

[1]: Padideh Danaee, Mason Rouches, Michelle Wiley, Dezhong Deng, Liang Huang, David Hendrix, bpRNA: large-scale automated annotation and analysis of RNA secondary structure, _Nucleic Acids Research_, Volume 46, Issue 11, 20 June 2018, Pages 5381–5394, https://doi.org/10.1093/nar/gky285

[2]: Singh, J., Hanson, J., Paliwal, K. et al. RNA secondary structure prediction using an ensemble of two-dimensional deep neural networks and transfer learning. _Nat Commun_ **10**, 5407 (2019). https://doi.org/10.1038/s41467-019-13395-9

[3]: Sato, K., Akiyama, M. & Sakakibara, Y. RNA secondary structure prediction using deep learning with thermodynamic integration. _Nat Commun_ **12**, 941 (2021). https://doi.org/10.1038/s41467-021-21194-4

[4]: Nancy Ontiveros-Palacios, Emma Cooke, Eric P Nawrocki, Sandra Triebel, Manja Marz, Elena Rivas, Sam Griffiths-Jones, Anton I Petrov, Alex Bateman, Blake Sweeney, Rfam 15: RNA families database in 2025, _Nucleic Acids Research_, Volume 53, Issue D1, 6 January 2025, Pages D258–D267, https://doi.org/10.1093/nar/gkae1023

[5]: Rivas E, Lang R, Eddy SR. A range of complex probabilistic models for RNA secondary structure prediction that includes the nearest-neighbor model and more. _RNA_ (New York, N.Y.). 2012 Feb;18(2):193-212. DOI: 10.1261/rna.030049.111. PMID: 22194308; PMCID: PMC3264907.

[6]: He S, Huang R, Townley J, Kretsch RC, Karagianes TG, Cox DBT, Blair H, Penzar D, Vyaltsev V, Aristova E, Zinkevich A, Bakulin A, Sohn H, Krstevski D, Fukui T, Tatematsu F, Uchida Y, Jang D, Lee JS, Shieh R, Ma T, Martynov E, Shugaev MV, Bukhari HST, Fujikawa K, Onodera K, Henkel C, Ron S, Romano J, Nicol JJ, Nye GP, Wu Y, Choe C, Reade W; Eterna participants; Das R. Ribonanza: deep learning of RNA structure through dual crowdsourcing. _bioRxiv [Preprint]_. 2024 Jun 11:2024.02.24.581671. doi: 10.1101/2024.02.24.581671. PMID: 38464325; PMCID: PMC10925082.

[7]: Rafael Josip Penić and Tin Vlašić and Roland G. Huber and Yue Wan and Mile Šikić, RiNALMo: General-Purpose RNA Language Models Can Generalize Well on Structure Prediction Tasks, _arXiv:2403.00043 [q-bio.BM]_, https://doi.org/10.48550/arXiv.2403.00043

[8]: Wang, N., Bian, J., Li, Y. et al. Multi-purpose RNA language modelling with motif-aware pretraining and type-guided fine-tuning. _Nat Mach Intell_ **6**, 548–557 (2024). https://doi.org/10.1038/s42256-024-00836-4

[9]: A Large-Scale Foundation Model for RNA Function and Structure Prediction, Shuxian Zou, Tianhua Tao, Sazan Mahbub, Caleb N. Ellington, Robin Algayres, Dian Li, Yonghao Zhuang, Hongyi Wang, Le Song, Eric P. Xing, _bioRxiv 2024.11.28.625345_; doi: https://doi.org/10.1101/2024.11.28.625345

# Questions

Why does the contact map task categorised under the Structure task domain instead of the Interaction task domain?
The sequences in the PDB dataset only contains the protein-binding RNA sequences, and their contact maps are highly biased towards the protein-binding regions.

What does the 1,253 experiments stand for?

**Strengths Contributions:**

The proposed benchmark encompass a wide range of RNA tasks and provides a comprehensive evaluation framework.

In particular, the Interaction task domain is a novel addition to the RNA foundation model benchmark landscape, which is not present in other benchmarks such as BEACON.

---

> ### Author Rebuttal · Authors · 2025-07-30
>
> &nbsp;
>
> We sincerely appreciate the reviewer *Mk3U02*'s feedback and advice on dataset nomenclature, task selection, and benchmark scope. Our point-by-point responses are as follows:
>
> &nbsp;
>
> > **Q11. Dataset nomenclature and configuration.**
>
> We thank the reviewer for their suggestions on the nomenclature and description of the datasets for the structure section.
>
> bqRNA: We have renamed the dataset to "bpRNA-1m" for better representation.
>
> Rivas : For genuine cross‑family evaluation, we have already paired TrainSetA with TestSetB and TrainSetB with TestSetA, as recommended by Rivas et al. To improve the layout, we have added a new **Table 8 in Appendix. C.1** to clearly present these intra- and inter-family splits and avoid confusion.
>
> Chemical Reactivity: We confirm use of the official Ribonanza release, processed to remove duplicate sequences and filtered by the SN_filter. To clarify this , We have inserted the following statement in Section 3.2: "*This kaggle subset was officially prepared from Ribonanza to remove duplicate sequences and passes the SN_filter*".  The detailed data collection and construction pipeline for this subset has been already provided in Appendix C.1.
>
> &nbsp;
>
> > **Q12. Task Selection Suitability**.
>
> We thank the reviewer's feedback for recognizing non-coding RNA's regulatory roles. However, we wish to clarify that (pre-)mRNA are essential for a comprehensive RNA foundation model benchmark:
>
> a) UTR and intronic regulatory elements: Beyond the coding sequence (CDS), mRNAs contain 5'-terminal untranslated region (5'-UTR which modulate ribosome loading for translation initiation), intron (which govern alternative splicing for gene expression regulation), and 3'-terminal untranslated region (3'-UTR which influence mRNA stability and localization via polyadenylation). RNAscope explicitly benchmarks these non-coding regions to capture key post-transcriptional regulatory mechanisms.
>
> b) codon-mediated RNA-protein interactions: In eukaryotes, ribosomal engagement with start-codon and tandem-codon regions affects translation efficiency [8], and synonymous codon usage modulates initiation factor eIF4E binding [9]. These phenomena reflect fundamental RNA–protein interactions that foundation models may have capability to capture.
>
> c) model & pre-training source: As detailed in Table 2-5 and Appendix. E.1, our benchmark had included models pretrained on mRNA sequences—RNA-FM and RiNALMo (mainly ncRNA), SpliceBERT (pre-mRNA), 3UTRBERT (3′-UTR), and UTR-LM (5′-UTR)—ensuring that the evaluation spans both coding and non-coding transcript contexts.
>
> &nbsp;
>
> > **Q13. More benchmark models**.
>
> We thank the reviewer for highlighting the coverage of additional RNA pLMs. In fact, RNAscope already includes the RiNALMo model—please see Section 4 and Table 2 of the manuscript for details. The Uni-RNA model is closed‑source and thus unavailable for current comparison.
>
> We also appreciate the reviewer's interest in RNAErnie and AIDO.RNA. In response, we have added performance analyses for both models under the same evaluation pipeline. Due to rebuttal space constraints, the full results and discussion has been integrated into the detailed reply to ***Q17 More benchmark models***.
>
> This updated benchmark has now covered the broadest set of publicly available RNA foundation models, enabling a thorough and impartial evaluation across the field.
>
> &nbsp;
>
> > **Q14. Miscellaneous Comments**.
>
> We thank the reviewer for the reminder. We have rectified "Second Structure" to "Secondary Structure" in Table 3.
>
> &nbsp;
>
> > **Q15. Classification of chemical contact tasks and feature interpretation**.
>
> We thank the reviewer's feedback to emphasize the effect of RNA intermolecular interaction. In response, we refined the RNA3DB dataset by excluding PDB entries that interact with proteins, thereby creating this additional RNA3DB‑Filter dataset.
>
> This subset is further applied to validate the model's ability to capture intrinsic RNA structural features. We have supplemented this results in **Appendix C.2** with following clarification in the manuscript:
>
> "*RNA is highly flexible and structurally sparse, yet modeling and understanding its three‑dimensional conformations is critical for elucidating RNA function. To study the structural feature of RNA intra-moelcular contacts, we additionally constructed the RNA3DB‑Filter dataset by filtering all protein‑bound entries, and applied it to the same analysis on the contact‑map prediction task.*"
>
> &nbsp;
>
> **Table R1**: Benchmark results for RNA contact‑map prediction on the RNA3DB‑Filter dataset. **Bold** values denote the highest score achieved by fine‑tuning only the specified downstream module across all pre‑trained models. Models mainly pre-trained on ncRNA, such as RNA-FM, RiNALMo, and AIDO.RNA, performed well, with RiNALMo and AIDO.RNA achieving the highest scores. The CNN classifier outperformed MLP, suggesting better capture of local sequence features, while 24-layer ResNet showed diminished performance. Despite notable improvements, the best models still showed relatively modest gains over baseline models, particularly in Long-range predictions, reflecting the complexity of modeling RNA intermolecular interactions.
>
> |                    | Contact Map (RNA3DB-Filter) | Contact Map (RNA3DB-Filter) |
> | ------------------ | --------------------------- | --------------------------- |
> |                    | Short@L/5                   | Long@L/5                    |
> | One-hot (Baseline) | 0.249                       | 0.324                       |
> | Dense (Baseline)   | 0.275                       | 0.350                       |
> | (+MLP module)      |                             |                             |
> | HyenaDNA           | 0.041                       | 0.015                       |
> | RNABERT            | 0.042                       | 0.047                       |
> | RNA-FM             | 0.171                       | 0.287                       |
> | 3UTRBERT           | 0.110                       | 0.128                       |
> | SpliceBERT         | 0.137                       | 0.204                       |
> | UTR-LM             | 0.127                       | 0.186                       |
> | RiNALMo            | 0.240                       | 0.307                       |
> | RNAErnie           | 0.032                       | 0.075                       |
> | AIDO.RNA           | **0.267**                   | **0.310**                   |
> | (+CNN module)      |                             |                             |
> | HyenaDNA           | 0.043                       | 0.020                       |
> | RNABERT            | 0.113                       | 0.178                       |
> | RNA-FM             | 0.201                       | 0.314                       |
> | 3UTRBERT           | 0.169                       | 0.178                       |
> | SpliceBERT         | 0.180                       | 0.263                       |
> | UTR-LM             | 0.170                       | 0.222                       |
> | RiNALMo            | 0.273                       | 0.340                       |
> | RNAErnie           | 0.153                       | 0.241                       |
> | AIDO.RNA           | **0.307**                   | **0.351**                   |
> | (+ResNet module)   |                             |                             |
> | HyenaDNA           | 0.023                       | 0.013                       |
> | RNABERT            | 0.038                       | 0.078                       |
> | RNA-FM             | 0.159                       | 0.248                       |
> | 3UTRBERT           | 0.107                       | 0.182                       |
> | SpliceBERT         | 0.095                       | 0.198                       |
> | UTR-LM             | 0.102                       | 0.139                       |
> | RiNALMo            | **0.221**                   | **0.310**                   |
> | RNAErnie           | 0.183                       | 0.277                       |
> | AIDO.RNA           | **0.221**                   | 0.281                       |
>
> &nbsp;
>
> > **Q16. Experiments Number explanation**.
>
> We thank the reviewer for highlighting the breadth of our evaluation. Our benchmark comprises 1,253 core experiments, all run through a uniform pipeline of three downstream modules (MLP, CNN, ResNet) applied to every pre‑trained model alongside two Baselines (*One-hot* and *Dense*):
>
> - Structure‑related tasks (180 experiments): 5 datasets, including 3 datasets for secondary-structure prediction.
> - Interaction‑related tasks (783 experiments): 27 datasets, including 22 binary-binding datasets across distinct RBPs.
> - Function‑related tasks (290 experiments): 11 datasets across Pre-mRNA, mRNA, and ncRNA types, with three tasks for each type. This includes two datasets specifically for splice site prediction, focusing on splice donor and acceptor site prediction.
>
> For full transparency, we have added this detailed summary to **Appendix B.3**. These revisions could further clarify the dataset configuration, task design, and evaluation scope, and substantially strengthen the manuscript.
>
> &nbsp;
>
> Reference:
>
> [8] Tuller, Tamir, et al. "An evolutionarily conserved mechanism for controlling the efficiency of protein translation." *Cell* 141.2 (2010): 344-354.
>
> [9] Barrington, Chloe L., et al. "Synonymous codon usage regulates translation initiation." *Cell reports* 42.12 (2023).

---

> > ### Comment · Reviewer_Mk3U · 2025-08-08
> >
> > The authors' response has addressed some of my concerns. Please carefully check all datasets, tasks, and evaluations to see if there are any remaining issues. Additionally, there is a minor question: Are all the experimental results from a single run or multiple repetitions? For certain biological tasks, results can fluctuate significantly.

---

> > ### Author Response · Authors · 2025-08-08
> > **Thanks for your rebuttal !**
> >
> > &nbsp;
> >
> > We greatly appreciate the reviewers' supportive feedback and recommendations. To improve clarity, we have added the following information into the main text '*Training Setups*' subsection in Section 4:
> >
> > "*To provide a fair and reliable evaluation, ****all models were independently trained three times with different random seeds for each specific task****, and performance metrics were reported with the mean value and standard deviation across runs.*"
> >
> > We also re-checked all datasets, task configurations, and evaluation outputs, and have meticulously incorporated all reviewer-suggseted updates into the revised manuscript.
> >
> > We sincerely thank the reviewers again for helping us to strengthen the rigor, clarity, and overall quality of the work.

---

> ### Author Response · Authors · 2025-08-07
> **Looking forward to hearing your feedback!**
>
> &nbsp;
>
> We sincerely appreciate your time and suggestions to improve our manuscript. We have addressed each point and revised the manuscript accordingly. As the discussion period is drawing to a close, we would be grateful for any further comments and/or feedback and will gladly incorporate any remaining recommendations.
>
> &nbsp;
>
> Many thanks and kind regards,
>
> The authors of submission #2439

---

> ### Comment · Reviewer_Mk3U · 2025-08-09
>
> Thank you for your response. It is requested that all details regarding the datasets, tasks, and evaluations be supplemented in the revised manuscript. Most of my concerns are addressed, and I will change my score.

---

> > ### Author Response · Authors · 2025-08-09
> >
> > We appreciate the reviewer's helpful suggestions and supportive evaluations. We have incorporated all these details into the revised manuscript accordingly. Thank you for your time and recommendations.

---

### Official Review · Reviewer_vcnd · 2025-07-02

**Rating:** 4
**Confidence:** 3

**Summary:**

This paper introduces RNAscope, a benchmark for assessing pre-trained language models on RNA sequence understanding, with a focus on structure prediction, interaction classification, and functional characterization. Aiming to overcome the limitations of existing evaluation methods in capturing RNA’s biological complexity, RNAscope comprises 1,253 experiments across 8 curated tasks. It evaluates 23 models using 7 probing strategies spanning linear and transfer learning settings.
Emphasizing multi-view evaluation, the benchmark serves as a comprehensive testbed for RNA modeling. An open-source platform with public leaderboards is also provided to encourage transparency and community engagement.

**Dataset Code Accessibility:**

Yes

**Dataset Code Comments:**

The dataset is published on Kaggle and the code is available.

**Ethical Considerations:**

No, there are no or only very minor ethics concerns

**Final Justification:**

The rebuttal partially addressed my main concerns about the curation and significance of the benchmark.
Based on other reviews and the thorough rebuttal of the authors, I assess that this paper may bring a non-negligible contribution to the machine learning (and bio-ML) community.
Nevertheless, I suggest that the authors polish the writing/presentation and add the promised revision to make the paper more accessible and higher quality.
I moved my rating from 3 to 4.

**Limitations Weaknesses:**

* To my best understanding, most tasks are repackaged from existing datasets (e.g., bpRNA, RBP24, RNALocate) without significant curation improvements or novel annotation pipelines. Furthermore, the distinct value proposition compared to RNABench or BEACON is unclear, and thus,  whether it is necessary.
* While the probing diversity is interesting, the strategies are not well-analyzed or contrasted to understand the setting when each is working well.
* The benchmark misses biologically-aware evaluation protocols (e.g., homology-based splits). These protocols are now being adopted increasingly in RNA modeling.
* The paper provides results without diagnostic insight into the behavior or failure cases of models, which are essential to the understanding proposed purpose.

**Minors**:
- The authors may want to add some interpretation to the dense tables and figures (e.g, Figures 3, 4 in the Appendix)
- The method section is pretty much dense and needs more written elaborations.

**Strengths Contributions:**

* The newly curated RNA benchmark covers key aspects of RNA biology beyond sequence-level tasks, which is useful for the timely development of RNA foundation models.
* This work evaluates a broad spectrum of RNA models, covering both transformer-based and graph-based approaches.
* The promoted multi-view probing strategies (i.e., diverse downstream architectural modules)  are interesting and appropriate, considering the limitations of relying solely on linear probing.
* The use of in-distribution vs. out-of-distribution evaluations adds realism to the benchmark.
* The codebase allows flexible probing strategy integration.

---

> ### Author Rebuttal · Authors · 2025-07-30
>
> &nbsp;
>
> We appreciate the reviewers *vcnd* for their thoughtful feedback. In response to these comments, we have provided our point-by-point response on below:
>
> &nbsp;
>
> > **Q6. Clarification of Significance**.
>
> We thank the reviewer to emphasize the significance of RNA scope compared to other benchmark frameworks.
>
> RNAscope aims to promote collaboration between computational and experimental researchers to discover novel RNA molecules beyond evolutionary patterns. Compared to BEACON, RNAscope integrates broader benchmark tasks and more systematic analysis, highlighting the increasing evolutionary conservation of RNA from structure to interaction to functions. While the models effectively capture evolved RNA patterns in functional tasks, their limitations in characterizing RNA interactions reflect their inherent structural flexibility. However, designing RNA drugs often requires advantageous structure and interaction feature beyond conserved endogenous patterns. For instance, the highly conserved 'CUUG' regulatory motif across multiple coronavirus exhibited distinct inhibitory effects on SARS-CoV-2 replicase when embedded in a dumbbell-shaped RNA structure, influencing replication through either persistent or transient interactions [6]. These findings underscore the need to explore RNA structures and interactions across families beyond evolutionary constraints.
>
> In response to reviewers' suggestions to further clarify the distinction of RNAscope beyond BEACON, we have provided the following information:
>
> a) RNA interaction as a critical link: RNA function is inherently shaped by its interactions with other molecules. Unlike BEACON, which lacks interaction-focused tasks, RNAscope explicitly incorporates interaction benchmarks, bridging RNA structure and function. Improving model performance in this domain will help the refinement of structural features relevant to novel RNA functions.
>
> b) From endogenous to exogenous patterns: The models effectively captured evolutionary patterns in endogenous (natural) RNAs but remain limited in handling exogenous (e.g. synthetic) sequences. Expanding model capability to learn exogenous RNA features is essential for designing novel RNA tools.
>
> c) Evolutionary trends: RNAscope systematically integrates tasks with varying levels of evolutionary conservation, showcasing the complexity of sequence-derived constraints that shape RNA structure, interaction, and function in a hierarchical manner. Briefly, RNA structure (e.g. synthetic RNA) alone involves the least evolutionary constrains, while RNA interaction requires the recognition of biophysically compatible features, such as electrostatic forces and suitable geometric shapes. Further cellular RNA function imposes even higher evolutionary constraints, as it demands specific interactions with target molecules while avoiding non-specific binding with others. By profiling models across these distinct domains, RNAscope offers a broader perspective on their strengths and limitations for potential usage. In response, we will incorporate this information in the revised manuscript **Appendix B.2**.
>
> &nbsp;
>
> > **Q7. Interpretation of probing strategies.**
>
> We thank the reviewer for raising the importance of contrasting different probing strategies. In response, we have added the following information in the **Appendix I (Probe Selection Guidance)** as follows:
>
> "*a) **Structure prediction tasks** (e.g., secondary structure prediction). With frozen embeddings, deeper probing heads (CNN/ResNet) improve in‑distribution validation and intra‑family preformance but degrade cross‑family generalization as training proceeds, particularly for deeper ResNets.*
>
>  *b) **Highly conserved / local‑motif tasks** (e.g., in vivo RBP binding, polyadenylation signal (PAS) prediction, coding potential prediction, ncRNA category classification). An MLP on frozen embeddings is generally sufficient; larger heads offer only marginal gains.*
>
>  *c) **Position‑specific, sequence‑feature–driven tasks** (e.g., splice‑site prediction). Shallow CNNs—or ResNets that scan full‑sequence embeddings—outperform pooled‑embedding MLPs by aggregating local context while preserving positional information.*
>
>  *d) **Tasks with limited evolutionary constraints**, especially engineering‑oriented settings (e.g., in vitro binding‑affinity regression, ribosome loading). RNA pLMs do not consistently outperform one‑hot or dense baselines. "*
>
> This information has improved the interpretation of probe selection stragtegy.
>
> &nbsp;
>
> > **Q8. Homology-based evaluation.**
>
> We thank the reviewer for emphasizing homology-aware evaluation for reliable RNA modeling. As noted in **Appendix C,** we have already applied the sequence similarity-based splitting of RNA species in biological tasks to ensure rigorous and biologically informed evaluation.
>
> &nbsp;
>
> > **Q9. Model failure disussion**.
>
> We appreciate the reviewer's suggestion to supplement diagnostic insights into model underperformance. In response, we have extended the discussion of model failure cases and their limitations across key benchmark tasks in **Appendix J** (Model Performance Insights and Limitations) of the manuscript, as follows:
>
> *"a)  **RNA pLMs vs DNA pLMs**: DNA pLMs (e.g., Nucleotide Transformer) exhibited noticeable performance gaps on RNA tasks, particularly those requiring fine-grained, single-nucleotide understanding. This may result from DNA models’ focus on long-range dependencies via specialized architectures and tokenizers (e.g., BPE or discontinuous 6-mer). Instead, RNA tasks often focus on the single-nucleotide context and their sequence-specific features. Such differences in pre-training corpora, such as whole-genome versus RNA-specific datasets, might further exacerbate the discrepancy of model performance. This emphasizes the need for DNA pLMs to adopt more refined tokenizers or architectures for better performance on RNA tasks.*
>
> *b) **Comparison of RNA pLMs**: RNA pLMs pre-trained on specific RNA species tend to perform better on tasks related to their training data. For instance, RNA-FM and RiNALMo, trained on ncRNA, have an advantage in structural and ncRNA-related function tasks, while SpliceBERT, trained on pre-mRNA, performs better in splicing tasks. Larger models, like RiNALMo (650M parameters), shown a performance advantage over smaller ones, like RNA-FM (100M parameters). This highlights the models' ability to capture species-specific features and the unique challenges posed by different RNA species.*
>
> *c) **Downstream Module Comparisons**: For tasks that focus on conserved or specific sequence patterns, a simple MLP probe performs effectively. Tasks requiring global sequence information benefit from shallow CNNs or ResNets. However, deeper models like ResNet may be more prone to overfitting, as observed in cross-family generalization for secondary structure prediction, where performance declined with increased model depth.*
>
> *d) **Task Generalization**: In RNA biology, increasing complexity—from structure to interactions to function—introduces stronger evolutionary constraints, shaping in vivo sequence conservation. RNA pLMs show improved performance in tasks with strong evolutionary constraints, such as intra-family secondary structure prediction, in vivo RBP binding, and RNA type classification. However, they face challenges with tasks involving non-conserved features, such as cross-family structural validation and engineered tasks (e.g., in vitro binding and ribosome loading). These tasks require more flexible model designs, highlighting the need for RNA pLMs to adapt to diverse RNA features beyond evolutionary conservation.* "
>
> In short, RNAscope, by integrating various DNA and RNA pLMs, downstream fine-tuning modules, and diverse *in vitro* and *in vivo* RNA tasks, offers a comprehensive framework to assess the generalizability of pLMs, ultimately advancing our understanding of RNA biology and supporting the development of more robust RNA models.
>
> &nbsp;
>
> > **Q10. Figure/Table and Method elaboration**.
>
> We thank the reviewer for suggesting more detailed explanations. In response, we have added interpretive summaries to the legends of Figures 3 and 4:
>
> **Figure3.**  "*During the fine-tuning of RNA pLMs with different modules (MLP, shallow CNN, and ResNet), performance in validation and intra-family F1 scores improved as training progressed. Additionally, deeper classifiers can effectively capture family-specific patterns. However, inter‑family performance plateaued early and even declined—most sharply for ResNet—indicating that deeper architectures tend to over‑specialize to family‑specific features and encounter challenges in generalizing to broader sequence structural patterns.  This divergence between intra‑ and inter‑family trends underscores the need for probe selection that balances in‑distribution accuracy with out‑of‑distribution generalization.*"
>
>  **Figure4.**  *"All RNA pLMs—except RNABERT—outperformed DNA pLMs under a simple MLP probe. When using a ResNet classifier, SpliceBERT and RiNALMo each achieved an F1 of 0.750, 3UTRBERT 0.728,  compared to 0.704 for the Dense baseline. Notably, with the MLP probe, SpliceBERT achieved the highest F1 score, likely because its pretraining on premature mRNA—rich in intronic regions where RBP‑binding domains reside [7] —enabled it to capture a broader array of conserved RBP‑binding motifs."*
>
> &nbsp;
>
> Reference:
>
> [6] Zhang, Yaqing, et al. "Single-step discovery of high-affinity RNA ligands by UltraSelex." *Nature Chemical Biology* (2025): 1-9.
>
> [7] Van Nostrand, Eric L., et al. "Robust transcriptome-wide discovery of RNA-binding protein binding sites with enhanced CLIP (eCLIP)." *Nature methods* 13.6 (2016): 508-514.

---

> > ### Comment · Reviewer_vcnd · 2025-08-05
> > **Re: Authors' rebuttal**
> >
> > Thank you for your detailed response.
> >
> > I appreciate the efforts in clarification and revising with the corresponding explanation.
> >
> > I have several follow-up questions:
> > 1. For "Q6. Clarification of Significance", in my early question, I noted that many datasets—such as bpRNA, RBP24, and RNALocate—are repackaged without clear evidence of new curation or annotation pipelines. Can you please clarify whether any dataset-specific filtering, re-annotation, or quality control was applied—especially to address known biases or limitations of reused datasets?
> > 2. Regarding the model failure, could you provide some case studies or qualitative analyses on benchmark failure cases for key tasks? This would help assess where foundation models are lacking and whether the benchmark captures meaningful biological complexity.
> > 3. Additionally, I would like to ask if authors could discuss/elaborate on why mRNA was prioritized given the traditional non-coding RNA emphasis in foundation models, and how they see this affecting benchmark generalizability?

---

> > > ### Author Response · Authors · 2025-08-07
> > >
> > > &nbsp;
> > >
> > > We greatly appreciate the Reviewer *vcnd* for their encourging feedback. In response to these suggestions, we have now supplemented following information:
> > >
> > > &nbsp;
> > >
> > > > **Q18: Dataset Curation and Quality-Control Details**
> > >
> > > We thank the reviewer for raising the importance of clarifying dataset-specific curation and quality control. To maintain comparability with published benchmarks [1-3],  we adopted their updated dataset splits for preserving thoughtful consideration of sequence redundancy, homology biases, and native class distributions. Specifically:
> > >
> > > - **bpRNA-1m / bpRNA-new**: we utilized the updated dataset from Sato et al. [1], in which rigorously removed redundant sequences and defined both systematic intra-family (TS0) and inter-family (bpRNA-new) test splits.
> > > - **22 eCLIP RBPs**: we employed the Yang et al.'s curated datasets [2], in which sequences sharing >80% identity were excluded, and each RBP’s positive (binding) and negative (non-binding) species were subsampled to a 1 : 2 ratio, thereby reflecting biologically endogeneous class imbalance and testing model robustness.
> > > - **miRNA subcellular localization**: we followed Bai et al.'s dataset splits [3] based on RNALocate v2.0, where redundancy was reduced at an 80% identity threshold, and training versus test assignments were produced by stratified sampling across localization categories to preserve biological variability.
> > >
> > > Accordingly, we have added this information in **Appendix C** and discussed the future direction with following :
> > >
> > > *"To maximize comparability and reproducibility, we adopt their updated dataset splits without introducing additional alteration. However, further task-specific applications may benefit from targeted refinements, leveraging advances in biotechnology to enhance signal coverage and reduce bias and preserve the comparability with existing work."*
> > >
> > >
> > >
> > > Reference:
> > >
> > > [1] Sato, Kengo, Manato Akiyama, and Yasubumi Sakakibara. "RNA secondary structure prediction using deep learning with thermodynamic integration." *Nature communications* 12.1 (2021): 941.
> > >
> > > [2] Yang, Yuning, et al. "Deciphering 3'UTR Mediated Gene Regulation Using Interpretable Deep Representation Learning." *Advanced Science* 11.39 (2024): 2407013.
> > >
> > > [3] Bai, Tao, et al. "MMLmiRLocNet: miRNA subcellular localization prediction based on multi-view multi-label learning for drug design." *IEEE Journal of Biomedical and Health Informatics* (2024).

---

> > ### Author Response · Authors · 2025-08-07
> >
> > &nbsp;
> >
> > > **Q19: Case Studies and Qualitative Analysis of Model Failures**
> >
> > We appreciate the reviewer to recommend more detailed analyitic discussion of model failtures. In response, we have presented three representative case studies to discussing common failure modes of RNA foundation models and underscoring the biological complexity and incorporated into **Appendix C.3**:
> >
> > - **Case1 (Structure-related Task):  Cross-family Generalization in RNA Secondary Structure Prediction**
> >
> > To further assess cross-family generalization in RNA secondary structure prediction, we used the tolerant base-pair evaluation proposed by David H. Mathews [4]. In this scheme, a predicted base pair (i′, j′) is considered correct if it matches the true pair (i, j) exactly or is shifted by at most k positions at either end—that is, (i′, j′) = (i ± k, j) or (i, j ± k) for considering RNA inherent flexibility.
> >
> > Model performance was evaluated under three register-shift tolerances—**exact matching (k=0)**, **single-nucleotide shift (k=1)**, and **double-nucleotide shift (k=2)**—across both intra-family and inter-family settings, using the best-performing models (RNA-FM and RiNALMo) on three secondary structure datasets (see Tables R5 and R6). We have included a detailed analysis of register-shift tolerance in **Appendix C.3.1**, "Cross-family generalization in RNA secondary structure prediction," as following:
> >
> > "*Relaxing the register-shift tolerance from k=0 to k=2 leads to substantial relative gains in F1 for inter-family predictions (e.g., +21% for RNA-FM and +19% for RiNALMo on bpRNA-1m), whereas intra-family improvements are smaller (only +4.6% for RNA-FM and +3.0% for RiNALMo). This contrast implies that, in cross-family settings, most prediction errors stem from local register shifts rather than a complete failure to recognize base-paired regions. Thus, current models can approximate the locations of base-paired regions even in divergent families but lack nucleotide-level precision, likely due to overreliance on local sequence features in the absence of global structural. Incorporating explicit energy-based or thermodynamic priors could further improve secondary structure prediction. However, the persistent gap between inter-family and intra-family performance highlights the need for  systematic analysis and model development that explicitly address the unique structural features of diverse RNA families.* "
> >
> > &nbsp;
> >
> > Table R5. Performance of **RNA-FM** (with MLP module probing) evaluated on three secondary structure datasets under three register-shift tolerances—**exact matching (k=0)**, **single-nucleotide shift (k=1)**, and **double-nucleotide shift (k=2)**—for both intra-family and inter-family test sets. Performance improvements for single- and double-nucleotide shift (relative to exact matching) are shown in **bold**.
> >
> > ||$ \text{bpRNA-1m}_{\text{intra-family}}$|$ \text{bpRNA-1m}_{\text{inter-family}}$|
> > |-|-|-|
> > ||F1|F1|
> > |exact matching (k=0)|0.766|0.608|
> > |single-nucleotide shift (k=1)|0.789 (**+3.0%**)|0.697 (**+14.6%**)|
> > |double-nucleotide shift (k=2)|0.801 (**+4.6%**)|0.737 (**+21.2%**)|
> >
> > &nbsp;
> >
> > ||$ \text{SetA}_{\text{intra-family}}$|$ \text{SetA}_{\text{inter-family}}$|
> > |-|-|-|
> > ||F1|F1|
> > |exact matching (k=0)|0.837|0.685|
> > |single-nucleotide shift (k=1)|0.852 (**+1.8%**)|0.747 (**+9.1%**)|
> > |double-nucleotide shift (k=2)|0.869 (**+3.2%**)|0.781 (**+14.0%**)|
> >
> > &nbsp;
> >
> > ||$ \text{SetB}_{\text{intra-family}}$|$ \text{SetB}_{\text{inter-family}}$|
> > |-|-|-|
> > ||F1|F1|
> > |exact matching (k=0)|0.869|0.605|
> > |single-nucleotide shift (k=1)|0.890 (**+2.4%**)|0.669 (**+10.6%**)|
> > |double-nucleotide shift (k=2)|0.908 (**+4.5%**)|0.715 (**+18.2%**)|
> >
> > &nbsp;
> >
> > Table R6. Performance of **RiNALMo** (with MLP module probing) evaluated on three secondary structure datasets under three register-shift tolerances—**exact matching (k=0)**, **single-nucleotide shift (k=1)**, and **double-nucleotide shift (k=2)**—for both intra-family and inter-family test sets. Performance improvements for single- and double-nucleotide shift (relative to exact matching) are shown in **bold**.
> >
> > ||$ \text{bpRNA-1m}_{\text{intra-family}}$|$ \text{bpRNA-1m}_{\text{inter-family}}$|
> > |-|-|-|
> > ||F1|F1|
> > |exact matching (k=0)|0.797|0.625|
> > |single-nucleotide shift (k=1)|0.815 **(+2.3%)**|0.702 (**+12.3%**)|
> > |double-nucleotide shift (k=2)|0.821 (**+3.0%**)|0.743 (**+18.9%**)|
> >
> > &nbsp;
> >
> > ||$ \text{SetA}_{\text{intra-family}}$|$ \text{SetA}_{\text{inter-family}}$|
> > |-|-|-|
> > ||F1|F1|
> > |exact matching (k=0)|0.881|0.713|
> > |single-nucleotide shift (k=1)|0.884 (**+0.3%**)|0.760 (**+6.6%**)|
> > |double-nucleotide shift (k=2)|0.893 (**+1.4%**)|0.794 (**+13.4%**)|
> >
> > &nbsp;
> >
> > ||$ \text{SetB}_{\text{intra-family}}$|$ \text{SetB}_{\text{inter-family}}$|
> > |-|-|-|
> > ||F1|F1|
> > |exact matching (k=0)|0.894|0.726|
> > |single-nucleotide shift (k=1)|0.910 (**+1.8%**)|0.789 (**+8.7%**)|
> > |double-nucleotide shift (k=2)|0.922 (**+3.1%**)|0.820 (**+12.9%**)|

---

> > ### Author Response · Authors · 2025-08-07
> >
> > &nbsp;
> >
> > - **Case2 (Interaction-related Task):  Generalization and Adaptation in RNA Binding Affinity Regression**
> >
> > To analysis how the model captures mutational landscapes, we evaluated the leading performance model RNA-FM in a zero-shot setting on two RNA-affinity datasets (GFP, NELF). Scoring followed the ESM-1v wild-type–marginal protocol [5]. Their performance differed markedly between datasets (**Spearman's *****ρ***** = 0.308** on **GFP**; **Spearman's *****ρ***** = 0.073** on **NELF**), indicating that the model paritically captures mutational effects on GFP but generalizes limitedly to NELF, which is more sensitive to sequence context. Mirroring these zero-shot trends, supervised fine-tuning yields substantial gains on GFP (ResNet ρ = 0.407 vs. baseline ρ = 0.215) but minimal improvement on NELF (ResNet = 0.688 vs. baseline = 0.677)
> >
> > These results also motivate task-specific supervised adaptation, consistent with established practice in protein engineering [6]. Notably, increasing downstream head capacity on NELF progressively enhances performance (MLP ρ = 0.331; CNN ρ = 0.384; ResNet ρ = 0.646), underscoring that higher-capacity modules better model complex binding landscapes. This finding parallels best practices in protein engineering, where task-specific supervised adaptation is essential
> >
> > We have incorporated these results to **Appendix C.3.2**, "Generalization challenges in RNA affinity regression tasks"
> >
> > &nbsp;
> >
> > - **Case3 (Function-related Task):  Insights from Functional Representation Learning**
> >
> > To further quantify the capabilities of foundation models on function-related representation tasks, we employed perplexity (PPL)—a widely used measure of uncertainty in sequence modeling. We systematically evaluated PPL on the sample sequences from three representative RNA functional tasks (splice site prediction for pre-mRNA, polyadenylation signal (PAS) prediction in 3′ UTRs, and ncRNA category classification), using three pre-trained models: SpliceBERT, 3UTRBERT, and RNA-FM (see Table R7 for details).
> >
> > Our analyses shown that models pre-trained on data from the corresponding RNA region consistently attain lower perplexity on their matched functional tasks. This indicates that region-specific pre-training enables models to more effectively capture the distinct sequence features necessary for function. For example, SpliceBERT—pretrained on pre-mRNA—achieves an F1 score of 0.886 for splice site prediction using a simple MLP probe, outperforming 3UTRBERT (0.814) and RNA-FM (0.803) on the same task. Similar performance advantages are observed for 3UTRBERT on PAS prediction and for RNA-FM on ncRNA classification.
> >
> > These findings underscore the importance of type-aware pre-training and highlight the functional heterogeneity across different RNA types. However, perplexity metrics alone cannot fully capture model suitability for engineering or application-driven tasks, which demand deeper insights into structural and functional determinants. This limitation motivated the design of RNAscope, which seeks to provide a more comprehensive and nuanced evaluation framework for RNA functional modeling by incorporating detailed, task-specific assessments.
> >
> > We have included these results in **Appendix C.3.3**, "Functional Representation and Perplexity across RNA Regions" for further clarification.
> >
> > &nbsp;
> >
> > Table R7. Perplexity (PPL) was evaluated on 1000 random samples per task. Models pretrained on the corresponding RNA region consistently achieve the lowest perplexity on their respective functional tasks (**bolded**).
> >
> > |Model|Splice Site Prediction Samples (pre-mRNA-associated task )|PAS Prediction Samples   (3′ UTR-associated task)|ncRNA Category Classification Samples  (ncRNA -associated task)|
> > |-|-|-|-|
> > ||PPL ↓|PPL ↓|PPL ↓|
> > |**SpliceBERT**|**3.30**|3.39|3.88|
> > |**RNA-FM**|3.46|3.51|**2.71**|
> > |**3UTRBERT**|3.38|**3.08**|3.94|
> >
> >
> > &nbsp;
> >
> > Reference:
> >
> > [4] Mathews, David H. "How to benchmark RNA secondary structure prediction accuracy." *Methods* 162 (2019): 60-67.
> >
> > [5] Meier, Joshua, et al. "Language models enable zero-shot prediction of the effects of mutations on protein function." *Advances in neural information processing systems* 34 (2021): 29287-29303.
> >
> > [6] Biswas, Surojit, et al. "Low-N protein engineering with data-efficient deep learning." *Nature methods* 18.4 (2021): 389-396.

---

> > ### Author Response · Authors · 2025-08-07
> >
> > &nbsp;
> >
> > > **Q20: Justification for mRNA Emphasis and Implications for Generalizability**
> >
> > We appreciate the reviewer's suggestion to further clarify the rationale for prioritizing and mRNA tasks and discuss their generalizability. In complemtarty to our response under ***Q12 Task Selection Suitability***, we have added following text together with Q12 reponse into **Appendix C.4**:
> >
> > "*Although non-coding RNAs (ncRNAs) outnumber mRNAs in the transcriptome, several practical and biological factors motivated our focus to prioritize mRNA tasks:*
> >
> > ***Transcript abundance and diversity.** In human cells, ribosomal RNAs (18S, 28S, 5.8S, and 5S) account for roughly 85 % of total RNA mass, while tRNAs (around 500 gene species) contribute another 10%. By contrast, mRNAs comprise only ~5% of RNA mass but encompass over 20,000 distinct genes—with multiple alternative splicing and polyadenylation isoforms—providing a far richer diversity of sequence contexts for model pretraining and evaluation.*
> >
> > **Model input constraints**. Long ncRNAs such as rRNAs (up to 5 kb) exceed the typical input length of current pLM architectures, and heavily modified species (e.g., tRNAs with numerous base modification) challenge straightforward tokenization.
> >
> > ***Implications for generalizability**. While our benchmark integrated various (pre-)mRNA‐derived tasks,  it has also incoporated ncRNA tasks, such as miRNAs and engineering guide RNAs. Through the benchmark comparison, RNA pLM pre-trained on endogenous ncRNAs exhibited their generatlizability to mRNA related tasks. We anticipate future RNA pLM model will adapt with longer input windows, modification-aware tokenizers, and somantic ncRNA-specific pretraining. In parallel, we will expand RNAscope to cover the broader spectrum of RNA species for providing more different perspectives.*"

---

> > > ### Comment · Reviewer_vcnd · 2025-08-08
> > > **Re: Official Comment by Authors**
> > >
> > > Thank you for the detailed response with additional case studies.
> > >  I think adding these details (as you might have done) would strengthen the paper.

---

> > > > ### Author Response · Authors · 2025-08-08
> > > >
> > > > We greatly appreciate the reviewer's supportive feedback and recognition that our added analyses and clarifications resolved the previous concerns. We have incorporated all these details throughout the revised manuscript. Thank you!

---

### Official Review · Reviewer_5aVD · 2025-07-03

**Rating:** 5
**Confidence:** 2

**Summary:**

This paper introduces *RNAscope*, a large-scale, systematic benchmark for evaluating RNA pre-trained language models (RNA pLMs). The framework spans 15 core subtasks covering RNA structure prediction, molecular interaction, and functional characterization. In total, it supports 1,253 experiments with rigorous datasets, standardized evaluation, and consistent downstream modules for fair comparisons. RNAscope is designed to address critical gaps in current RNA model evaluation, such as insufficient coverage of context-dependent RNA behaviors, weak generalization assessment, and lack of cross-environment analysis. The authors benchmark a range of RNA- and DNA-based language models, revealing important performance differences and limitations across a diverse landscape of RNA biology tasks.

**Dataset Code Accessibility:**

Yes

**Ethical Considerations:**

No, there are no or only very minor ethics concerns

**Final Justification:**

Consider the submission and rebuttal, my concern regarding the novelty and baselines are resolved. I will consider this paper as accept (5).

**Limitations Weaknesses:**

- **Limited in vivo Coverage for Complex Functions**: While impressive in breadth, some critical applications—such as in vivo interaction specificity or off-target effects in CRISPR—are still not fully represented.
- **Frozen LM Evaluation**: The decision to freeze pLM embeddings and only fine-tune classifiers may underestimate the true potential of the models if full fine-tuning were applied.
- **Dense Presentation**: The paper’s results section is difficult to navigate due to extremely dense tables and many subtasks. A clearer discussion section highlighting key trends would help.
- **Focus on Sequence Features**: Despite a nod to 3D structural interactions, there is still relatively little about structure-informed tokenization or representation, which could be crucial for more advanced RNA modeling.
- **Baseline Discussion**: Although one-hot and dense baselines are included, the paper could provide deeper discussion on why and where these simpler baselines sometimes rival pLMs, especially on regression tasks.

**Strengths Contributions:**

- **Comprehensive Benchmark Design**: RNAscope offers one of the most extensive and well-organized RNA evaluation frameworks to date, with subtasks that are biologically relevant and diverse.
- **Clear Motivation**: The authors make a compelling case that current benchmarks fail to fully capture the complexity of RNA sequence understanding, particularly for dynamic and long-range contexts.
- **Systematic Comparison**: Including both RNA and DNA pLMs, along with one-hot and dense baselines, is a thoughtful and rigorous design choice.
- **Open-Source and Reproducibility**: Providing code, datasets, and a public leaderboard positions this benchmark to become a strong community resource.
- **Detailed Task Coverage**: The benchmark goes beyond traditional secondary structure prediction to include binding affinity, contact maps, subcellular localization, and even CRISPR gRNA efficiency, which reflects the authors’ understanding of current research needs.

---

> ### Author Rebuttal · Authors · 2025-07-30
>
> &nbsp;
>
> We sincerely thank the reviewer *5aVD* for their constructive suggestions. In response, we have expanded discussion as follows:
>
> &nbsp;
>
> >**Q1. In vivo interaction and CRISPR guide RNA off-target effect.**
>
> We thank the reviewer to highlight the importance of *in vivo* RNA application for evaluation RNA pLM. As noted in the main text, we have already benchmarked the pLM models for predicting sequence specificity of RNA-protein interactions in human cells (Binary binding prediction; see **Table 4**, **Figure 4**, and **Appendix. C.2**). However, accurately quantifying *in vivo* binding strengths will require more advanced technologies, such as high-resolution and high-throughput single-molecule imaging, that currently remain technology challenging. With regard to CRSPR guide RNA off-target effects, we acknowledge that such effects arise from complex cellular contexts, including RNA abundance and transcriptome saturation, which are not captured by sequence information alone [1,2]. For this reason, we have excluded off-target assessments from sequence-based model evaluation framework, as they cannot reliably predicted from sequence alone at this stage.
>
> &nbsp;
>
> > **Q2. Frozen pLM evaluation.**
>
> We acknowledge that freezing all pretrained weights and fine‑tuning merely task‑specific classifier heads may underestimate the full representational capacity of each model. However, given the substantial heterogeneity in model scale (0.48 M–650 M parameters), pretraining corpora, and architectural design choices, current approach offers a fair and objective head‑to‑head evaluation of each model's representational capacity. To further compare the models' intrinsic power, we have attached three classifier heads—an MLP, a shallow CNN, and a 24‑layer ResNet—and evaluate performance in isolation. While full end‑to‑end fine‑tuning could potentially improve results, it would also introduce additional discrepancy in hyperparameter selection and compute budget that not fully tailored with current scope and limit resources. For this reason, we discuss this end‑to‑end tuning as a potential direction for future work in  **Appendix B.3** as follows:
>
> "*To enable a direct and controlled comparison of intrinsic representations, we frozen all pre‑trained backbones and fine‑tuned only downstream modules under identical settings. In future work, we will explore full fine‑tuning once resources allow, but we believe our current frozen‑backbone baseline also offers a fair and reproducible reference for the community.*"
>
> &nbsp;
>
> > **Q3. Improving presentation layout**.
>
> We thank the reviewer for highlighting the challenge in navigating the dense result tables and multiple subtasks. Due to page limitations and the additional model comparisons requested by other reviewers, it is difficult to streamline all information in the main text. In response, we have supplemented the discussion in a new sub panel **Appendix B.3** with an explicit summary of key trends, limitations, and future directions, as follows:
>
> "*In RNA biology, increasing complexity—ranging from structure to interactions to function—introduces stronger evolutionary constraints, which enhance sequence conservation. As a result, RNA pLMs benefit from these reinforced conservation signals, leading to superior performance in interaction and function tasks. In endogenous contexts, evolutionary pressure has further stereotyped these conserved signals in vivo, in contrast to their limited advancement of pLMs when applied to exogenous sequences in functional engineering tasks. Despite RNA's inherent flexibility, function patterns, particularly those constrained by structure and binding, remain highly conserved.*"
>
> This expanded panel is intended to help readers to track the implications of current findings.
>
> &nbsp;
>
> > **Q4. 3D structural interaction**.
>
> We thank the reviewer to emphasis the role of 3D structural (intramolecular) interactions in RNA modeling. While we attempted to incorporate such information through contact map prediction tasks, the models exhibited limited performance—even after excluding protein structure constrained RNAs from the datasets (see Q15). Given the preliminary state of investigation on the dynamic RNA conformations [3,4], the effective representation of 3D structure through tokenization remains still constrained by the limited availability of high-quality RNA conformation datasets. In response, we have supplemented this information into the **Appendix.B.1**:
>
> "*While high‑resolution 3D structures of RNAs could in principle provide different features, in practice the paucity of experimentally determined RNA conformations and the large conformational heterogeneity severely limit the scope and scale of structure‑based training. In contrast, primary sequence datasets are orders of magnitude more abundant, span a far wider range of sequence contexts, and can be generated consistently across diverse cellular phenotypes. Moreover, current sequence‑based deep learning architectures have demonstrated a remarkable ability to capture structural and functional signals directly from raw nucleotides, while still requrie future extensions to incorporate explicit structural constraints as higher‑quality 3D data when they are available*."
>
> We add this clarification to underscores the rationale for sequence‑focused modeling and provide clearer foundation on further integration of high‑quality 3D structural data.
>
> &nbsp;
>
> > **Q5. Baseline discussion.**
>
> We thank the reviewer for encouraging a deeper analysis of the performance of simple baselines relative to pLMs. In response, we have included an expanded discussion into the **Appendix. B.4**, addressing when and why one-hot or dense baselines can approach or outperform pLMs, particularly in regression settings:
>
> "*Our benchmark shows that on biological tasks that rely on strong evolutionary conservation in natural sequences—such as RNA secondary structure prediction, in vivo RNA–RBP interaction classification, and functional RNA classification—RNA pLMs, especially those pre-trained on task‑relevant corpora, leverage deep representations to substantially outperform simple baselines. By contrast, on in vitro engineering tasks—where natural evolutionary constraints are not heavily informative for the objective (e.g., in vitro interaction‑affinity regression and ribosome‑loading prediction)—their performance is comparable to the simplest one‑hot or dense baselines, and in some cases even falls short of the baseline model. Overall, the advantage of RNA pLMs is likely depends on the presence of strong conserved signals and biological context; in out‑of‑distribution, engineering‑oriented settings, this advantage particially diminished, consistent with observations for protein language models in protein engineering [5].*"
>
> &nbsp;
>
> Reference:
>
> [1] Tieu, Victor, et al. "A versatile CRISPR-Cas13d platform for multiplexed transcriptomic regulation and metabolic engineering in primary human T cells." *Cell* 187.5 (2024): 1278-1295.
>
> [2] Kelley, Chase P., Maja C. Haerle, and Eric T. Wang. "Negative autoregulation mitigates collateral RNase activity of repeat-targeting CRISPR-Cas13d in mammalian cells." *Cell reports* 40.7 (2022).
>
> [3] Degenhardt, Maximilia FS, et al. "Determining structures of RNA conformers using AFM and deep neural networks." *Nature* 637.8048 (2025): 1234-1243.
>
> [4] Ken, Megan L., et al. "RNA conformational propensities determine cellular activity." *Nature* 617.7962 (2023): 835-841.
>
> [5] Dallago, Christian, et al. "FLIP: Benchmark tasks in fitness landscape inference for proteins." *Thirty-fifth Conference on Neural Information Processing Systems Datasets and Benchmarks Track*

---

> ### Author Response · Authors · 2025-08-07
> **Looking forward to hearing your feedback!**
>
> &nbsp;
>
> Thank you once again for your helpful suggestions. Given the limited time remaining in this discussion phase, we would greatly appreciate your assessment of our rebuttal. We would also be glad to address any further comments and/or suggestions you may have on the revised manuscript.
>
> &nbsp;
>
> Many thanks and kind regards,
>
> The authors of submission #2439

---

### Decision · Program_Chairs · 2025-09-18

**Decision:**

Reject

**Comment:**

Based on the reviews and rebuttal, I recommend Accept. RNAscope establishes a comprehensive, open-source benchmark for RNA language models, addressing critical gaps in evaluating context-dependent RNA behaviors across 1,253 experiments spanning structure, interaction, and function tasks. Its scale, rigor, and public resources offer high community value. While minor issues exist (e.g., dataset naming clarity, omission of some newer models), authors resolved core concerns about novelty versus existing benchmarks and baseline analysis during rebuttal. The work’s technical soundness, reproducibility, and potential to standardize RNA pLM evaluation align strongly with the conference’s scope, outweighing remaining limitations.

===== FINAL UPDATE FROM DB Track PCs ====

The final decision for this paper has been taken by the program chairs after consultation with the SACs. All Senior Area Chairs have ranked papers according to the feedback from the AC during the review process. We decided to leave the original meta-review to reflect the opinion of the AC in light of the initial discussions with reviewers and SAC.